



## 2 Seasonal cycles and trends of water budget components in 18

## 3 river basins across Tibetan Plateau: a multiple datasets

## 4 perspective

Wenbin Liu[a], Fubao Sun[a*], Yanzhong Li[a], Guoqing Zhang[b,c],
Yan-Fang Sang[a], Jiahong Liu[d], Hong Wang[a] ,Peng Bai[a]
[a]Key Laboratory of Water Cycle and Related Land Surface Processes, Institute of Geographic
Sciences and Natural Resources Research, Chinese Academy of Sciences, Beijing 100101, China
[b]Key Laboratory of Tibetan Environmental Changes and Land Surface Processes, Institute of
Tibetan Plateau Research, Chinese Academy of Sciences, Beijing 100101, China
[c]CAS Center for Excellent in Tibetan Plateau Earth Sciences, Beijing 100101, China
[d]Key Laboratory of Simulation and Regulation of Water Cycle in River Basin, China Institute of
Water Resources and Hydropower Research, Beijing 100038, China
**Submitted to**: Hydrology and Earth System Sciences
**Corresponding Author**: Dr. Fubao Sun (Sunfb@igsnrr.ac.cn), from the Key Laboratory of Water
Cycle and Related Land Surface Processes, Institute of Geographic Sciences and Natural
Resources Research, Chinese Academy of Sciences (No. A11, Datun Road, Chaoyang District,
Beijing 100101, China)
**Email Addresses for other authors**: Wenbin Liu (liuwb@igsnrr.ac.cn), Yanzhong Li
(liyz.14b@igsnrr.ac.cn), Guoqing Zhang (guoqing.zhang@itpcas.ac.cn), Yan-fang Sang
(sangyf@igsnrr.ac.cn), Jiahong Liu (liujh@iwhr.com) , Hong Wang (wanghong@igsnrr.ac.cn) ,
Peng Bai (baip.11b@igsnrr.ac.cn)

27                                                                                    2016/11/25





**Highlights**

• Monthly basin-wide ET was calculated through water balance considering the
impacts of glacier and water storage change
• Water budget components and trends for 18 river basins over TP were evaluated
• Uncertainties were discussed from multiple dataset perspective





**Abstract.** The insights of water budgets over Tibetan Plateau (TP) are not fully
understood so far due to the lack of quantitative observations of the land surface
processes. Here, we investigated the seasonal cycles and trends of water budget
components in 18 TP basins through the use of multi-source datasets during the period
1982-2011. A two-step bias correction procedure was applied to calculate the
basin-wide evapotranspiration (ET) through the water balance considering the
influences of glacier and water storage change. The results indicated that precipitation,
which mainly concentrated during June-October (varied among different monsoons
impacted basins), is the major contributor to the runoff in TP basins. The basin-wide
snow water equivalent (SWE) was relatively higher from mid-autumn to spring for
most TP basins. The water cycles intensified under a global warming in most basins
except for the upper Yellow and Yalong Rivers, which were significantly influenced
by the weakening East Asian monsoon. Corresponded to the climate warming and
moistening in the TP and western China, the aridity index (PET/P) in most basins
decreased. The general hydrological regimes could be inferred from the perspective of
multi-source datasets although there are considerable uncertainties from different
datasets, which are comparable to some existing studies using the field observations
and complex modeling approaches. The results highlighted the usefulness of
integrating the multi-source data (e.g., in situ observations, remote sensing products,
reanalysis, land surface model simulations and climate model outputs) for
hydrological applications in the data-sparse environments and could be benefit for





understanding the water and energy budgets, sustainable management of water
resources under a warming climate in the harsh and data-sparse Tibetan Plateau.

## 1 Introduction

As the highest plateau in the globe (the average elevation is higher than 4000 meters
above the sea level), Tibetan Plateau (TP, also called "the roof of the world" or "the
third Pole") is one of the most vulnerable region under a warming climate and is
subjected to strong interactions among atmosphere, hydrosphere, biosphere and
cryosphere in the earth system (Duan and Wu, 2006; Yao et al., 2012; Liu W. et al.,
2016b). It also serves as the "Asian water tower" with many major Asian rivers such
as Yellow river, Yangtze river, Brahmaputra river, Mekong river, Indus river, etc.,
originate from, which provides a vital water resource to support hundreds of millions
of people in China and the surrounding countries (Immerzell et al., 2010; Zhang et al.,
2013). Knowledge about the water budgets and their responses to the changing
environment is thus crucial for understanding the hydrological regimes and for
sustainable water resources management as well as environmental protection in this
special region (Yang et al., 2014; Chen et al., 2015).

TP is also known as a typical data-sparse mountain region which brings great
challenges to hydrological and related land surface studies (Zhang et al., 2007; Li F. et
al., 2013; Liu X. et al., 2016). For example, since the 1950s, totally 750 stations have
been established over China by the Chinese Meteorological Administration (CMA),
among which only less than 80 stations are distributed over the plateau (Wang and
Zeng, 2012). They are primary sparse and unevenly located at relatively low elevation
regions, focus only on the meteorological variables and lack of other land surface



observations such as evapotranspiration, snow water equivalent and latent heat fluxes,
etc.. In addition, long-term consecutive observations of river discharge, snow depth,
lake depth and glacier melts in TP are also absent (Akhta et al., 2009; Ma et al., 2016).
Therefore, the insights of water balance over various TP river basins locates at
different monsoon-dominant regions are, to some extent, still unclear so far due to the
lack of quantitative observations of the land surface processes (Cuo et al., 2014; Xu et
al., 2016). One way to break this limitation is to install more instruments to measure
the point scale water budgets (Yang et al., 2013; Zhou et al., 2013; Ma et al., 2015),
but it is extremely expensive to maintain long-term observations at the harsh
environment and is often difficult to be applied to basin or regional scales. Another
more popular way is to simulate basin-wide water budgets through physical-based
land surface models at several large river basins forced with remote sensing data and
large-scale gridded meteorological forcing datasets (Bookhagen and Burbank, 2010;
Xue et al., 2013; Zhang et al., 2013; Cuo et al., 2015; Zhou et al., 2015; Wang et al.,
2016). However, it is also limited by the lack of adequate data for model
calibration/validation and is hard to be used to multiple basins especially to relatively
smaller basins under the complex terrains (Li F. et al., 2014).

In recent years, a number of global (or regional) datasets for water budget components
have been released including remote sensing-based retrievals (Tapley et al., 2004;
Zhang et al., 2010; Long et al., 2014; Zhang Y. et al., 2016), land surface model (LSM)
simulations (Rui, 2011), reanalysis outputs (Berrisford et al., 2011; Kobayashi et al.,
2015) and gridded forcing data interpolated from the in situ observations (Harries et
al., 2014). For example, there are considerable products for terrestrial
evapotranspiration (ET) such as GLEAM_E (Global Land surface Evaporation: the



Amsterdam Methodology, Miralles et al., 2011a), MTE_E (a product integrated the
point-wise ET observation at FLUXNET sites with geospatial information extracted
from surface meteorological observations and remote sensing in a machine-leaning
algorithm, Jung et al., 2010 ), LSM-simulated ETs from Global Land Data
Assimilation System version 2 (GLDAS-2) with different land surface schemes
(Rodell et al., 2004), ETs from Japanese 55-year reanalysis (JRA55_E), the
ERA-Interim global atmospheric reanalysis dataset (ERAI_E) and the National
Aeronautic and Space Administration (NASA) Modern Era Retrospchective-analysis
for Research and Application (MERRA) reanalysis data (Lucchesi, 2012). Moreover,
there are also several global or regional LSM-based runoff simulations from GLDAS
and the Variable Infiltration Capacity (VIC) model (Zhang et al., 2014). A few
attempts have been made to validate multiple datasets for certain water budget
component and to explore their possible hydrological implications, for example, Li X.
et al. (2014) and Liu W. et al. (2016a) evaluated multiple ET estimates against the
water balance method at annual and monthly time scales. Bai et al. (2016) assessed
streamflow simulations of GLDAS LSMs in five major rivers over TP based on the
discharge observations. Although there are certain uncertainties among different
datasets with various spatial and temporal resolutions and calculated though different
algorithms (Xia et al., 2012), they do provide a great chance for us to quantify the
general basin-wide water budgets and their uncertainties in gauge-sparse regions such
as TP considered in this study.

The objectives of this study are (1) to investigate the general water budgets in 18 river
basins across Tibetan Plateau from the perspective of multiple datasets, and (2) to
evaluate the seasonal cycles and annual trends of water budget components for 18 TP



basins. The paper is organized as follows: the datasets and methods applied in this
study are described in Sect.2. The results of season cycles and annual trends of water
budget components for 18 TP basins are presented and discussed in Sect.3. The
uncertainties inherited from multiple datasets are also discussed. In the Sect.4, we
summarized the general results which would helpful for understanding the water
balances of TP Rivers located at westerlies-dominated, Indian monsoon-dominated
and East Asian monsoon-dominated regions.

**2  Data and Method**
**2.1 Multiple datasets used**
**2.1.1 Study basins**
Eighteen river basins over TP (Fig.1) with the drainage area ranging from 2832 to
191235 km$^2$ (Table 1) are chosen in this study due to the availability of runoff data
during the period 1982-2011. They mainly locate at the northwestern, southeastern
and eastern parts of the plateau with multiyear-mean and basin-averaged temperature
and precipitation ranging from -5.68 to 0.97 $^{o}$C and 128 to 717 mm, which are solely
or combined controlled by the westerlies, the Indian Summer monsoon and the Easter
Asian monsoon (Yao et al., 2012). The altitudes of the lowest and highest
hydrological gauging stations are 1650 m and 4982 m above the sea level. The glacier
and snow covers are relatively more for the westerlies-dominant basins such as
Yerqiang, Yulongkashi and Keliya (10.86~23.27% and 29.16~35.95%, respectively)
whereas are less for the East Asian monsoon-dominated basins such as Yellow,
Yangtze and Bayin (0~0.96% and 9.42~20.05%, respectively) (Table 1).
<Figure 1, here please, thanks>
<Table 1, here please, thanks>





### 2.1.2 Runoff, Precipitation and Terrestrial storage change


Observed daily runoff (Q) during the period 1982-2011 used for water balance
calculation for 18 TP basins was obtained from the National Hydrology Almanac of
China (Table 2). There are < 30% missing data in some gauging stations such as
Yajiang, Tongren, Gandatan and Zelingou. Therefore, the VIC Retrospective Land
Surface Dataset over China (1952~2012, VIC_IGSNRR simulated) with a spatial
resolution of 0.25 degree and a daily temporal resolution from the Geographic
Sciences and Natural Resources Research (IGSNRR), Chinese Academy of Sciences,
is also used, which is derived from the VIC model forced by the gridded daily
observed forcing (IGSNRR_forcing) (Zhang et al., 2014). A degree-day scheme was
used in the model to consider the influences of snow and glacier on hydrological
processes. In this study, we first assess the VIC_IGSNRR simulated runoff against the
observations for each basin (for example, at Tangnaihai and Pangduo stations in
Fig.2). The VIC_IGSNRR simulated runoff is acceptable and could be used to replace
the missing values for a given basin, if the Nash Efficiency coefficient (NSE) between
the observation and simulation is above 0.65.

< Figure 2, here please, thanks>

Monthly gridded precipitation dataset (0.5 degree, 1961-2011) form CMA, which was
interpolated from observations of 2472 national meteorological stations using the
Thin Plate Spline method, was used in this study (Table 2). Considering the
uncertainty of CMA precipitation over TP due to the relatively sparse stations used
and the complex terrain conditions, two other precipitation datasets (IGSNRR_forcing
and TRMM (Tropical Rainfall Measuring Mission) 3B43 V7, Huffman et al., 2012)
were also applied. The precipitation from IGSNRR forcing datasets (0.25 degree) was
derived by interpolating gauged daily precipitation from 756 CMA stations based on





the synergraphic mapping system algorithm (Shepard, 1984; Zhang et al., 2014) and
was further bias-corrected using the CMA gridded precipitation. The CMA
precipitation is perfectly consistent with TRMM (Corr = 0.86, RMSE = 8.34
mm/month) and IGSNRR forcing (Corr = 0.94, RMSE = 7.15mm/month)
precipitation for multiple basins (and also for the smallest basin above Tongren station,
Fig.2), which reveals the applicably of CMA precipitation under the TP conditions.

<Table 2, here please, thanks>

Three latest global terrestrial water storage anomaly and water storage change (ΔS)
datasets (available on the GRACE Tellus website: http://grace.jpl.nasa.gov/) retrieved
from the Gravity Recovery and Climate Experiment (GRACE, Tapley et al., 2004;
Landerer and Swenson, 2012; Long et al., 2014), which were processed separately at
the Jet Propulsion Laboratory (JPL), the GeoForschungsZentrum (GFZ) and the
Center for Space Research at the University of Texas (CSR), were used. The GRACE
retrievals (2002-2013) from three processing centers were averaged and a glacier
isostatic adjustment correction as well a destriping filter were applied to minimize the
errors and uncertainties of extracted ΔS.

**2.1.3 Temperature, potential evaporation and ET**
The CMA monthly gridded temperature (0.5 degree) and potential evaporation (PET)
dataset (0.5 degree, Harris et al., 2013) from Climatic Research Unit (CRU) in the
University of East Anglia were used in this study. Moreover, six published
global/regional ET products (four diagnostic products and two LSMs simulations,
Table 2), namely (1) GLEAM_E (Miralles et al., 2010, 2011), which estimated three
sources of ET (transpiration, soil evaporation and interception) separately through





bare soil, short vegetation and vegetation with a tall canopy through a set of algorithm
(www.gleam.eu), (2) GNoah_E simulated by GLDAS-2 with the Catchment Noah
scheme (http://disc.sci.gsfc.nasa.gove/hydrology/data-holdings) (Rodell et al., 2004),
(3) Zhang_E (Zhang et al., 2010) estimated using the modified Penman-Monteith
approach forced with MODIS data, satellite-based vegetation parameters and
meteorological observations (http://www.ntsg.umt.edu/project/et), (4) MET_E (Jung
et al., 2010) (https://www.bgc-jena.mpg.de/geodb/projects/Home.phs), (5) VIC_E
(Zhang    et    al.,    2014)    from    VIC_IGSNRR    simulations
(http://hydro.igsnrr.ac.cn/public/vic_outputs.html) and (6) PML_E (Zhang Y. et al.,
2016) computed from global observation-driven Penman-Monteith-Leuning (PML)
model (https://data.csiro.au/dap/landingpage?pid=csiro:17375&v=2&d=true).

**2.1.4 Vegetation and snow/glacier parameters**
Two vegetation parameter datasets, the Normalized Difference Vegetation Index
(NDVI) and the Leaf Area Index (LAI) were used to quantify the dynamics of
vegetation for 18 TP basins (Table 2). The NDVI data was obtained from the Global
Inventory Modeling and Mapping Studies (GIMMS) (Turker et al., 2005)
(https://nex.nasa.gov/nex/projects/1349/wiki/general_data_description_and_access/)
while the LAI data was collected from the Global Land Surface Satellite (GLASS)
products (http://www.glcf.umd.edu/data/lai/) (Liang and Xiao, 2012). Seasonal snow
and glacier are widespread over the plateau which significantly influences the water
and energy budgets in TP, but their observations are difficult due to the harsh
environment, especially at the basin scale. However, there are currently a few
satellite-based or LSM-simulated products which could provide general information



about the variations of snow and glacier. The daily cloud free snow composite product
from MODIS Terra-Aqua and the Interactive Multisensor Snow and Ice Mapping
System for the Tibetan Plateau was applied to quantify the snow cover changes for
each basin (Zhang et al., 2012; Yu et al., 2015). The snow water equivalent (SWE)
retrieved from Global Snow Monitoring for Climate Research product (GlobSnow-2,
http://www.globsnow.info/) and the VIC_IGSNRR simulations were also used in this
study (Takala et al., 2011; Zhang et al., 2014). Moreover, the Second Glacier
Inventory Dataset of China was used to extract the general distribution of glacier
(Guo et al., 2014). All gridded datasets used were first uniformly interpolated to a
spatial resolution of 0.5 degree to make their inter-comparison possible. The datasets
were then extracted for each of TP basins.

**2.1.5 Monsoon indices**
The TP climate is generally influenced by the westerlies, Indian summer monsoon and
East Asian summer monsoon (Yao et al., 2012). To investigate the changes of
monsoon systems and their potential influences on the water budget in TP basins,
three monsoon indices, namely Asian Zonal Circulation Index (AZCI), Indian Ocean
Dipole Mode Index (IODMI) and East Asian Summer Monsoon Index (EASMI), are
also used in this study. The IODMI is an indicator of the east-west temperature
gradient across the tropical Indian Ocean defined by Saji et al. (1999), which can be
downloaded from the following website:
http://www.jamstec.go.jp/frcgc/research/d1/iod/HTML/Dipole%20Mode%20Index.ht
ml. The EASMI and AZCI ($60^{o}$-$150^{o}$E) reflect the dynamics of East Asian summer
monsoon (Li and Zeng, 2002) and the westerlies, which can be obtained from the
http://ljp.gcess.cn/dct/page/65577 and the National Climate Center of China





(http://ncc.cma.gov.cn/Website/index.php?ChannelID=43WCHID=5), respectively.
**2.2 Methods**
**2.2.1 Water balance-based ET estimation**
The basin-wide water balance at the monthly and annual timescales could
traditionally be written as the principle of mass conservation (also known as the
continuity equation, Oliverira et al., 2014) of basin-wide precipitation (P, mm),
evapotranspiration ($ET_{wb}$, mm), runoff (Q, mm) as well as terrestrial water storage
change (ΔS, mm),
$$ET_{wb} = P - Q - \Delta S \qquad (1)$$
In most TP basins, glacier melt ($M_G$) contributes to river discharge together with
precipitation (liquid precipitation and snow). The monthly and annual water balance
in these basins can thus be revised as,
$$ET_{wb} = P + M_G - Q - \Delta S \qquad (2)$$
Several attempts have been made for separating glacier contributions to river
discharge through site-scale isotopic observations, remote sensing as well as
land-surface hydrological modeling for some individual TP basins (Zhang et al., 2013;
Zhou et al., 2014; Neckel et al., 2014). However, accurate quantification of $M_G$ is
difficult in data-sparse TP, especially for multiple basins. In this study, we simply use
the percentages of glacier melt to river discharge for some TP basins concluded from
the existing studies (Chen, 1988; Mansur and Ajnis, 2005; Zhang et al., 2013; Liu J. et
al., 2016) and the empirical relations between the glacier area ratio (%) and glacier
melt in basins mentioned above (Table 3).
<Table 3, here please, thanks>
The terrestrial water storage (ΔS) in Eq.(2), which includes the surface, subsurface
and ground water changes, cannot be neglected in water balance calculation at a





monthly or annual timescale due to snow accumulation and some anthropogenic
interferences such as reservoir regulation and agriculture irrigation (Liu W. et al.,
2016a). The water balance-based ET ($ET_{wb}$) during 2002-2011 can be calculated
through Eq. (2) using the GRACE-derived mass anomaly as $\Delta S$. For $ET_{wb}$
calculation before 2002 when the GRACE data is unavailable, we use a two-step bias
correction procedure (Li X. et al., 2014) to close the water balance for 18 basins at
monthly timescale considering the $\Delta S$. We define $P + M_G - Q$ as biased ET
($ET_{biased}$, available from 1982-2011) relative to the $ET_{wb}$ (available from 2002-2011
when the GRACE data is available) calculated from Eq. (2). Firstly, the $ET_{biased}$ and
$ET_{wb}$ series over the period 2002-2011 were separately fitted using a gamma
distribution, which has been evidenced as an proper method for modeling the
probability distribution of ET (Bouraoui et al., 1999). The value in monthly $ET_{biased}$
series (2002-2011) can be bias-corrected through the inverse function ($F^{-1}$) of the
gamma cumulative distribution function (CDF, F) of $ET_{wb}$ by matching the
cumulative probabilities between two CDFs as follow (Liu W. et al., 2016a),

$$ET_{wb}(m) = F^{-1}(F(ET_{biased}(m)|\alpha_{biased}, \beta_{biased})|\alpha_{wb}, \beta_{wb}) \qquad (3)$$

Here $\alpha_{biased}, \beta_{biased}, \alpha_{wb}$ and $\beta_{wb}$ are the shape and scale parameters of gamma
distribution for $ET_{biased}$ and $ET_{wb}$. The second step is to eliminate the annual bias
through the ratio of annual $ET_{biased}$ to annual $ET_{wb}$ calculated in the first step using
the following method,

$$ET_{wb}(m) = \frac{ET_{biased}(a)}{ET_{wb}(a)} \times ET_{wb}(m) \qquad (4)$$

The procedure was then applied to correct the monthly $ET_{biased}$ series and calculated
the monthly $ET_{wb}$ during the period 1982-2001 for all TP basins. The $ET_{wb}$ obtained
was seemed as the "true" ET for evaluating multiple ET products and further for the
trend analysis.





### 2.2.2 Modified Mann-Kendall test method


The Mann-Kendall (MK) test is a rank-based nonparametric approach and is less
sensitive to outlier relative to other parametric statistics. However, it is sometimes
impacted by the serial correlation of time series. In this study, we use a modified
version of MK test (MMK, Hamed and Rao, 1998) to quantify the trends of water
budget components in18 TP basins. The MMK considers the lag-$i$ autocorrelation and
related robustness of the autocorrelation, which has been widely used in previous
studies during the last five decades (McVicar et al., 2012; Liu and Sun, 2016).

### 3   Results and Discussion


### 3.1 ET evaluation and General hydrological characteristics of 18 TP basins


We first evaluated monthly performances of six ET products in 18 TP basins against
the $ET_{wb}$, which was calculated through water balance considering the impacts of
glacier and water storage change (Fig. 3). The ranges of monthly averaged ET among
different basins (approximately 4−39 mm month$^{-1}$) are very close for all products
compare with that calculated from the $ET_{wb}$ (6−42 mm month$^{-1}$). However,
GLEAM_E (correlation coefficient: Corr = 0.85 and root-mean-square-error: RMSE =
5.69 mm month$^{-1}$) and VIC_E (Corr = 0.82 and RMSE = 6.16 mm month$^{-1}$) perform
relatively better than others. Although Zhang_E and GNoah_E were found closely
correlated to monthly $ET_{wb}$ in the upper Yellow River, the upper Yangtze River,
Qiangtang and Qaidam basins (Li X. et al., 2014), they did not exhibit overall good
performances (Corr = 0.61, RMSE = 7.97 mm month$^{-1}$ for Zhang_E and Corr = 0.42,
RMSE = 10.16 mm month$^{-1}$ for GNoah_E) for 18 TP basin used in this study. We thus
use GLEAM_E and VIC_E together with $ET_{wb}$ to calculate the seasonal cycles and
trends of ET in 18 TP basins in the following sections.




< Figure 3, here please, thanks>

To investigate the general hydroclimatic characteristics of rivers over TP, we classify
18 basins into three categories, namely westerlies-dominated basins (Yerqiang,
Yulongkashi and Kelia), Indian monsoon-dominated basins (Brahmaputra and
Salween), and East Asian monsoon-dominated basins (Yellow, Yalong and Yangtze)
referred to Tian et al. (2007) and Yao et al. (2012, 2013). Interestingly, they are
clustered into three groups under the perspective of Budyko framework (Budyko,
1974; Zhang D. et al., 2016) with relatively lower evaporative index for Indian
monsoon-dominant basins and higher aridity index for westerlies-dominant basins,
which reveal various long-term hydroclimatologic conditions (Fig. 4). Overall, the
annual mean air temperature increases (-5.68 ~0.97 $^{\circ}$C) while multiyear mean glacier
area (and thus the glacier melt normalized by precipitation) decreases (23.27 ~ 0%)
gradually from the westerlies-dominant, Indian monsoon-dominant to East Asian
monsoon-dominant basins. The vegetation status (NDVI range: 0.05~0.43; LAI range:
0.03~0.83) tends to be better and ET increases (and thus runoff coefficient gradually
decreases) from cold to warm basins (Fig. 4 and Table 1). It is a general picture of
hydrological regime in high-altitude and cold regions (Zhang et al., 2013; Cuo et al.,
2014), which could be interpreted from the perspective of multi-source datasets in
data-sparse TP.

< Figure 4, here please, thanks>

**3.2 Seasonal cycles of basin-wide water budget components for TP basins**
The multi-year means of water budget components (i.e., P, Q, ET, snow cover and
SWE) and vegetation parameters (i.e., NDVI and LAI) were calculated for each
calendar month and for 18 TP river basins over using multi-source datasets available
from 1982 to 2011. Overall, the seasonal variations of P, Q, ET, air temperature and





vegetation parameters are similar in all TP basins with peak values occurred in May to
September (Fig.5 and Fig.6). The seasonal cycles of snow cover and SWE are
generally time consistent as well for 18 TP basins (the peak values mainly occur from
October to next April, Fig.7). With the ascending air temperature from cold to warm
months, the basin-wide precipitation increases and vegetation turns green gradually
(the basin-wide ET also increase). Meanwhile, glacier and snow melt or vanish
gradually with the melt water supply the river discharge together with precipitation.
The inter-basin variations of hydrological regime are to a large extent linked to the
climate systems that prevail over the TP.

< Figure 5, here please, thanks>

Although the temporal patterns of hydrological components are general analogous,
they varied among parameters, climate zones and even basins (Zhou et al., 2005). For
example, relative to air temperature, the seasonal variation of runoff is more similar to
precipitation which reveals that runoff is mainly controlled by precipitation in the TP
basins. It is in agreement with that summarized by Cuo et al. (2014). In the
westerlies-dominated basins, the peak values of precipitation and runoff mainly
concentrate in June-August, which contribute approximately 68-82% and 67-78% of
annual totals, respectively. During this period, the runoff always exceeds precipitation
which indicates large contributions of melt water to streamflow. It is consistent with
the existing findings in Tarim River (Yerqiang, Yulongkashi and Keliya rivers are the
major tributaries of Tarim River), which indicated that the melt water accounted for
about half of the annual total streamflow (Fu et al., 2008). The ET (vegetation cover)
in three westerlies-dominated basins are relatively less (scarcer) than that in other TP
basins while the percentages of glacier and seasonal snow cover are higher in these
basins which contribute more melt water to river discharge (Fig.6 and Fig.7). Overall,





the SWE in Yerqiang, Yulongkashi and Keliya rivers are relatively higher in winter
than other seasons, but they vary with basins and products which reveal considerable
uncertainties in SWE estimations.

< Figure 6, here please, thanks>

In the Indian monsoon and East Asian monsoon-dominated basins, the runoff
concentrates during June-September or June-October with precipitation being the
dominant contributor of annual total runoff. For example, the peak values of
precipitation and runoff occur during June-September at Zhimenda station
(contributing about 80% and 74% of the annual totals) while those occur during
June-October at Tangnaihai station (contributing about 78% and 71% of the annual
totals, respectively). The results are quite similar to the related studies in eastern and
southern TP such as Liu (1999), Dong et al. (2007), Zhu et al. (2011), Zhang et al.
(2013), Cuo et al. (2014). The vegetation cover (ET) in most basins is relatively better
(higher) than that in the westerlies-dominant basins. Moreover, the seasonal snow
mainly covers from mid-autumn to spring and correspondingly the SWE is relatively
higher in these months in all basins except for Yellow River above Xining station,
Salwee River above Jiayuqiao station and Brahmaputra River above Nuxia and
Yangcun stations.

< Figure 7, here please, thanks>

**3.3 Trends of basin-wide water budget components for TP basins**
Trends in water budget components for 18 TP basins during the period 1982-2011
were also examined through the modified Mann-Kendall test (MMK) in this study.
The hydrological cycles intensified in the westerlies-dominated basins with Q, P and
$ET_{wb}$  all ascended with regional warming (Fig.8), especially in the Keliya River
basin (Numaitilangan station). The aridity index (PET/P), which is an indicator for the



degree of dryness, declined in all basins in northwestern TP. The results were in line
with the overall climate warming and moistening reported in northwest China (Shi et
al., 2003), at which these basins located. The increase in streamflow was also found in
most tributaries of the Tarim River (Sun et al., 2006; Fu et al., 2010; Mamat et al.,
2010). Moreover, the westerlies, revealed by the Asian Zonal Circulation Index
($60^{o}$-$150^{o}$ E), enhanced (linear trend: 0.21) over the period of 1982-2011 (Fig.9).
More water vapor was transported and fell as precipitation or snow in northwestern
TP (e.g., the eastern Pamir region) with the strengthening westerlies. The SWE
showed increase for all basins and for both products (VIC_IGSNRR simulated and
GlobaSnow-2 product) with the incremental seasonal snow cover and advanced
glaciers (Yao et al., 2012). More precipitation was transformed into snow or glacier
and the runoff coefficient (Q/P) exhibited decrease although precipitation obviously
increased (Fig.8). In addition, the transpiration in these basins may decrease with
vegetation degradation revealed by the NDVI and LAI (Yin et al., 2016) but the
atmospheric evaporative demand indicated by CRU PET increased (significantly
increase in the Yulongkashi and Keliya rivers) during the period 1982-2011.

< Figure 8, here please, thanks>

< Figure 9, here please, thanks>

In the East Asian monsoon-dominated basins, there are two types of change for
basin-wide water budget components. For example, P and Q decreased in the upper
Yellow River (Tangnihai, Huangheyan and Jimai stations) and Yalong River (Yajiang
station) but increased in other basins (Zelingou, Gandatan, Xining, Tongren and
Zhimenda stations) over the period of 1982-2011 (Fig.10). The decline in Q and P for
the upper Yellow and Yalong Rivers (locate at eastern Tibetan Plateau) were
consistent with that found by Cuo et al. (2013, 2014) as well as Yang et al. (2014), and





were in line with the weakening (linear slope: -0.01) of the East Asian Summer
Monsoon (Fig.9). The vegetation turned green while $ET_{wb}$ and PET increased in all
nine basins with the ascending air temperature during the period 1982-2011. The
aridity index (PET/P) was found decrease in all basins except for the upper Yellow
River basin above Jimai station and the upper Yalong River basin above Yajiang
station. Moreover, the runoff coefficients (SWE) were decrease (decrease except for
the Bayin River above Zelingou station and the upper Yellow River above Tongren
station) in the East Asian monsoon dominated basins.
< Figure 10, here please, thanks>
The hydrological cycles were also found intensified in the Indian monsoon-dominated
basins such as Salween River and Brahmaputra River (Fig.11), which were in line
with the strengthen (linear trend: 0.0006) of the Indian Summer monsoon (revealed by
the Indian Ocean Dipole Mode Index) during the specific period 1982-2011 (Fig.9).
In the six basins, trends in P, Q and $ET_{wb}$ were all upward. For example, at
Jiayuqiao station, the annual streamflow showed increasing trend which was
consistent with that examined during 1980-2000 by Yao et al. (2012). The vegetation
status, revealed by NDVI and LAI, turned better with the ascending air temperature.
The aridity index (PET/P) decreased in all basins except for the Brahmaputra River
above Tangjia station, which indicated that most basins in the Indian
monsoon-dominated regions turn wet over the period of 1982-2011. The runoff
coefficient (Q/P) increased at Gongbujiangda and Nuxia while decreased at Jiayuqiao,
Pangduo, Tangji and Yangcun stations. Moreover, the basin-wide SWE declined in the
upper Salween River and Brahmaputra River above Pangduo, Tangjia and
Gongbujiangda stations while increased in Brahmaputra River above Nuxia and
Yangcun stations.



< Figure 11, here please, thanks>

**3.4 Uncertainties**
The results may unavoidably associate with several aspects of uncertainties which
mainly inherited from the multi-source datasets used. For example, although the
seasonal cycles of $ET_{wb}$ can be captured by GLEAM_E and VIC_E, they still have
considerable uncertainties such as at Numaitilangan, Gongbujiangda and Nuxia
stations (Fig.5). With respect to the annual trend of $ET_{wb}$ (Table 4), most ET products
(including the well-performed GLEAM_E and VIC_E in some basins) cannot detect
the decreasing trends in 7 out of 18 basins (at Kulukelangan, Tongguziluoke, Xining,
Tongren, Jimai, Nuxia and Gongbujiangda stations). We thus only used $ET_{wb}$ in the
trend detection of water budget components in Fig.8, Fig.10 and Fig.11 in this study.
The two SWE products also showed large uncertainty, with respect to both their
seasonal cycles and trends due to their different forcing data; different algorithms
applied as well as varied spatial-temporal resolutions. Moreover, the interpolation of
missing values of runoff with VIC_IGSNRR simulated runoff and the gridded
precipitation data (which interpolated from limited gauged precipitation over the
plateau) involved some uncertainties as well as. However, with these caveats, we can
interpret the general hydrological regimes and their responses to the changing climate
in TP basins from solely the perspective of multi-source datasets, which are
comparable to the existing studies based on the in situ observations and complex
hydrological modeling.

<Table 4, here please, thanks>

**4    Summary**
In this study, we investigated the seasonal cycles and trends of water budget
components in 18 TP basins during the period 1982-2011, which is not well



understood so far due to the lack of adequate observations in the harsh environment,
through integrating the multi-source global/regional datasets such as gauge data,
satellite remote sensing and land surface model simulations. By using a two-step bias
correction procedure, annual basin-wide $ET_{wb}$ was calculated through the water
balance considering the impacts of glacier and water storage change. The GLEAM_E
and VIC_E were found perform better relative to other products against the
calculated $ET_{wb}$.

The general water and energy budgets were different in the westerlies-dominated
(with higher aridity index, runoff coefficient and glacier cover), the Indian
monsoon-dominated and the East Asian monsoon-dominated (with higher air
temperature, vegetation cover and evapotranspiration) basins under the perspective of
Budyko framework. In 18 TP basins, precipitation is the major contributor to the river
runoff, which concentrates mainly during June-October (June-August for the
westerlies-dominated basins, June-September or June to October for the Indian
monsoon-dominated and the East Asian monsoon-dominated basins). The basin-wide
SWE is relatively higher from mid-autumn to spring for all 18 TP basins except for
Keliya River and Brahmaputra River above the Nuxia and Yangcun stations. The
vegetation cover is relatively less whereas snow/glacier cover is more in the
westerlies-dominant basins compared with other basins. The hydrological cycles were
found intensified under the regional warming in most TP basins except for most
tributaries of the upper Yellow River and the Yalong River, which were significantly
influenced by the weakening East Asian monsoon during the period 1982-2011. The
aridity index (PET/P) exhibited decrease in most TP basins which corresponded to the
warming and moistening climate in the TP and western China. Moreover, the runoff





coefficient (Q/P) declined in most basins which may be, to some extent, due to ET
increase induced by vegetation greening and the influences of snow and glacier
changes. Although there are considerable uncertainties inherited from multi-source
data used, the general hydrological regimes in TP basins could be revealed, which are
consistent to the existing results obtained from in situ observations and complex land
surface modeling. It indicated the usefulness of integrating the multiple datasets
available such as in situ observations, remote sensing-based products, reanalysis
outputs, land surface model simulations and climate model outputs for hydrological
applications. The results obtained could be helpful for understanding the hydrological
cycles, and further for the water resources management and eco-environment
protection under a warming climate in the vulnerable Tibetan Plateau.

*Author contributions*. Wenbin Liu and Fubao Sun developed the idea to see the
general water budgets in TP basins from the perspective of multisource datasets.
Wenbin Liu collected and processed the multiple datasets with the help of Yanzhong
Li, Guoqing Zhang, Hong Wang as well as Peng Bai, and prepared the manuscript.
The results were extensively commented and discussed by Fubao Sun, Jiahong Liu
and Yan-Fang Sang.

*Acknowledgements*. This study was supported by the National Key Research and
Development Program of China (2016YFC0401401 and 2016YFA0602402),National
Natural Science Foundation of China (41401037 and 41330529), the Open Research
Fund of State Key Laboratory of Desert and Oasis Ecology in Xinjiang Institute of
Ecology and Geography, Chinese Academy of Sciences (CAS), the CAS Pioneer
Hundred Talents Program (Fubao Sun), the Initial Founding of Scientific Research
(Y5V50019YE) and the program for the "Bingwei" Excellent Talents from the



Institute of Geographic Sciences and Natural Resources Research, CAS. We are
grateful to the NASA MEaSUREs Program (Sean Swenson) for providing the
GRACE land data processing algorithm. The basin-wide water budget series in TP
Rivers used in this study are available from the authors upon request
(liuwb@igsnrr.ac.cn).

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






**Table 1**: Main features of the 18 used TP river basins. GA% and SC% represent the percentages of multiyear-mean glacier cover and snow cover in each basin. The glacier and snow cover data are extracted, respectively, from the Second Glacier Inventory Dataset of China and the daily TP snow cover dataset (2005-2013)

| No. | Station | Altitude (m) | River name | Drainage area (km²) | Multiyear-mean (1982-2011) and basin-averaged parameters | | | | | | | |
| | | | | | Q (mm/yr) | Prec. (mm/yr) | Temp.(°C/yr) | NDVI | LAI | GA% | SC% |
|---|---|---|---|---|---|---|---|---|---|---|---|
| 01 | Kulukelangan | 2000 | Yerqiang | 32880.00 | 158.60 | 128.34 | -5.68 | 0.05 | 0.03 | 10.97 | 35.03 |
| 02 | Tongguziluoke | 1650 | Yulongkashi | 14575.00 | 151.56 | 134.04 | -4.07 | 0.06 | 0.04 | 23.27 | 35.95 |
| 03 | Numaitilangan | 1880 | Keliya | 7358.00 | 103.18 | 137.14 | -4.78 | 0.06 | 0.03 | 10.86 | 29.16 |
| 04 | Zelingou | 4282 | Bayin | 5544.00 | 41.42 | 340.68 | -4.98 | 0.13 | 0.09 | 0.09 | 21.22 |
| 05 | Gadatan | 3823 | Yellow | 7893.00 | 200.95 | 566.01 | -4.60 | 0.34 | 0.54 | 0.13 | 14.94 |
| 06 | Xining | 3225 | Yellow | 9022.00 | 99.90 | 503.74 | 0.97 | 0.36 | 0.70 | 0.00 | 10.06 |
| 07 | Tongren | 3697 | Yellow | 2832.00 | 149.36 | 533.25 | -1.37 | 0.39 | 0.83 | 0.00 | 9.42 |
| 08 | Tainaihai | 2632 | Yellow | 121972.00 | 159.48 | 540.32 | -2.40 | 0.34 | 0.72 | 0.09 | 15.89 |
| 09 | Huangheyan | 4491 | Yellow | 20930.00 | 31.18 | 386.42 | -4.81 | 0.23 | 0.61 | 0.00 | 17.25 |
| 10 | Jimai | 4450 | Yellow | 45015.00 | 85.50 | 441.48 | -4.16 | 0.26 | 0.52 | 0.00 | 20.05 |
| 11 | Yajiang | 2599 | Yalong | 67514.00 | 237.66 | 717.05 | -0.23 | 0.43 | 0.80 | 0.15 | 18.36 |
| 12 | Zhimenda | 3540 | Yangtze | 137704.00 | 96.23 | 405.66 | -4.83 | 0.20 | 0.26 | 0.96 | 17.87 |
| 13 | Jiaoyuqiao | 3000 | Salween | 72844.00 | 364.26 | 620.88 | -1.89 | 0.29 | 0.44 | 2.02 | 23.73 |
| 14 | Pangduo | 5015 | Brahmaputra | 16459.00 | 348.31 | 544.59 | -1.53 | 0.27 | 0.33 | 1.66 | 23.33 |
| 15 | Tangjia | 4982 | Brahmaputra | 20143.00 | 350.61 | 555.17 | -1.89 | 0.27 | 0.34 | 1.39 | 21.83 |
| 16 | Gongbujiangda | 4927 | Brahmaputra | 6417.00 | 586.96 | 692.06 | -4.24 | 0.27 | 0.36 | 4.12 | 25.99 |
| 17 | Nuxia | 2910 | Brahmaputra | 191235.00 | 307.38 | 401.35 | -0.73 | 0.22 | 0.25 | 1.90 | 13.50 |
| 18 | Yangcun | 3600 | Brahmaputra | 152701.00 | 163.25 | 349.91 | -0.87 | 0.19 | 0.18 | 1.28 | 10.52 |







**Table 2**: Overview of multi-source datasets applied in this study

| Data category | Data source | Spatial resolution | Temporal resolution | Available period used | Reference |
|---|---|---|---|---|---|
| Runoff (Q) | Observed, National Hydrology Almanac of China | — | Daily | 1982-2011 | — |
| | VIC_IGSNRR simulated | 0.25° | Daily | 1982-2011 | Zhang et al. (2014) |
| Precipitation (P) | Observed, CMA | 0.5° | Monthly | 1982-2011 | — |
| | TRMM 3B43 V7 | 0.25° | Monthly | 2000-2011 | Huffman et al. (2012) |
| | IGSNRR forcing | 0.25° | Daily | 1982-2011 | Zhang et al. (2014) |
| Temperature (Temp.) | Observed, CMA | 0.5° | Monthly | 2000-2011 | — |
| Terrestrial storage change | GRACE-CSR | Approx.300-400 km | Monthly | 2002-2011 | Tapley et al. (2004) |
| (ΔS) | GRACE-GFZ | Approx.300-400 km | Monthly | 2002-2011 | Tapley et al. (2004) |
| | GRACE-JPL | Approx.300-400 km | Monthly | 2002-2011 | Tapley et al. (2004) |
| Potential evaporation (PET) | CRU | 0.5° | Monthly | 1982-2011 | Harris et al. (2013) |
| Actual evaporation (ET) | MTE_E | 0.5° | Monthly | 1982-2011 | Jung et al. (2010) |
| | VIC_E | 0.25° | Daily | 1982-2011 | Zhang et al. (2014) |
| | GLEAM_E | 0.25° | Daily | 1982-2011 | Miralles et al. (2011) |
| | PML_E | 0.5° | Monthly | 1982-2011 | Zhang Y et al. (2016) |
| | Zhang_E | 8 km | Monthly | 1983-2006 | Zhang et al. (2010) |
| | GNoah_E | 1.0° | 3 hourly | 1982-2011 | Rui (2011) |
| NDVI | GIMMS NDVI dataset | 8 km | 15 daily | 1982-2011 | Tucker et al. (2005) |
| LAI | GLASS LAI Product | 0.05° | 8 daily | 1982-2011 | Liang and Xiao (2012) |
| Snow Cover | TP Snow composite Products | 500 m | Daily | 2005-2013 | Zhang et al. (2012) |
| SWE | VIC_IGSNRR simulated | 0.25° | Daily | 1982-2011 | Zhang et al. (2014) |
| | GlobSnow-2 Product | 25 km | Daily | 1982-2011 | Takala et al. (2011) |







**Table 3**: Contribution of glacier-melt to discharge in eighteen basins ("——" shows no glacier influences, "——*" shows the percentage is empirically estimated through the relation between lacier area ratio and glacier melt for basins in which the glacier melt contribution has been reported in existing studies)

| Basin | Contributions of glacier-melt to discharge (%) | Reference |
|---|---|---|
| Kulukelangan | 62.73 | Mansur and Ajnisa (2005) |
| Tongguziluoke | 64.90 | Liu J et al. (2016) |
| Numaitilangan | 71 | Chen (1988) |
| Zelingou | — | — |
| Gadatan | — | — |
| Xining | — | — |
| Tongren | — | — |
| Tainaihai | 0.80 | Zhang et al. (2013) |
| Huangheyan | — | — |
| Jimai | — | — |
| Yajiang | 1.40 | —* |
| Zhimenda | 6.50 | Zhang et al. (2013) |
| Jiaoyuqiao | 4.80 | Zhang et al. (2013) |
| Nuxia | 11.60 | Zhang et al. (2013) |
| Pangduo | 10.13 | —* |
| Tangjia | 8.49 | —* |
| Gongbujiangda | 25.15 | —* |
| Yangcun | 7.81 | —* |






**Table 4:** Nonparametric trends for different ET estimates during the period 1982–2006 detected by modified Mann-Kendall test, the bold number showed the detected trend is statistically significant at the 0.05 level

| Basin | $ET_{wb}$ | GLEAM_E | VIC_E | Zhang_E | PML_E | MET_E | GNoah_E |
|---|---|---|---|---|---|---|---|
| Kulukelangan | **-0.09** | 0.09 | **0.18** | — | 0.03 | -0.01 | 0.07 |
| Tongguziluoke | -0.02 | 0.10 | **0.13** | — | 0.03 | **-0.08** | 0.19 |
| Numaitilangan | 0.04 | **0.10** | 0.14 | — | 0.14 | **-0.10** | 0.22 |
| Zelingou | **0.13** | **0.23** | 0.11 | **0.09** | 0.04 | **0.06** | 0.02 |
| Gadatan | -0.09 | 0.25 | 0.070 | -0.10 | -0.01 | **0.06** | -0.07 |
| Xining | -0.06 | **0.54** | 0.01 | -0.08 | 0.01 | 0.02 | -0.06 |
| Tongren | -0.06 | **0.34** | -0.15 | **-0.17** | 0.07 | 0.02 | 0.13 |
| Tainaihai | 0.06 | **0.28** | -0.03 | **-0.11** | 0.04 | **0.05** | 0.04 |
| Huangheyan | 0.08 | **0.19** | -0.01 | **-0.10** | **0.08** | **0.05** | **0.10** |
| Jimai | -0.07 | **0.23** | -0.01 | -0.08 | 0.03 | **0.05** | 0.10 |
| Yajiang | 0.17 | **0.26** | **0.06** | **-0.21** | -0.01 | 0.03 | -0.02 |
| Zhimenda | 0.11 | **0.28** | 0.10 | 0.01 | 0.07 | **0.04** | 0.07 |
| Jiaoyuqiao | **0.18** | **0.28** | 0.10 | **-0.11** | 0.05 | **0.05** | 0.07 |
| Nuxia | **-0.09** | **0.25** | 0.09 | **-0.10** | **0.12** | **0.04** | 0.10 |
| Pangduo | 0.05 | **0.28** | **0.17** | **-0.07** | 0.07 | **0.07** | **0.11** |
| Tangjia | 0.09 | **0.26** | **0.17** | **-0.09** | **0.20** | **0.06** | **0.12** |
| Gongbujiangda | -0.26 | 0.12 | 0.13 | **-0.16** | **0.19** | 0.01 | **0.15** |
| Yangcun | 0.03 | **0.28** | 0.08 | **-0.06** | 0.10 | 0.04 | 0.09 |




**Figure captions:**

**Figure1.** Map of river basins and hydrological gauging stations (green dots) over the Tibetan Plateau (TP) used in this study. The grey shading shows the topography of TP in meters above the sea level and the blue shading exhibits the glaciers distribution in TP extracted from the Second Glacier Inventory Dataset of China.

**Figure 2.** Comparison of VIC_IGSNRR simulated and observed monthly runoff for Tangnaihai and Panduo stations (a and b) as well as (c) basin-averaged monthly TRMM, CMA gridded and IGSNRR forcing precipitations for the smallest basin (Tongren station) over the period 1982-2011. (d) shows the comparison of TRMM (blue) and IGSNRR forcing (red) precipitations against CMA gridded precipitation for 18 river basins over TP during the period 2000-2011.

**Figure 3.** Comparison of different ET products against the calculated ET through the water balance method ($ET_{wb}$) for 18 TP basins. The boxplot of annual estimates of different ET products for 18 TP basins are shown in (a) while the correlation coefficients and root-mean-square-errors (RMSEs, mm/month) for each ET product relatively to $ET_{wb}$ are exhibited in (b).

**Figure 4**. General water and energy status (a. the perspective of Budyko framework) and their relationships with glacier (b) and vegetation (c and d) for eighteen TP river basins (1983-2006). The ET used in this figure is calculated from the bias-corrected water balance method.

**Figure 5**. Seasonal cycles (1982-2011) of water budget components in westerlies-dominated (column 1), East Asian monsoon-dominated (columns 2-4) and Indian monsoon-dominated (columns 5-6) TP basins.

**Figure 6**. Seasonal cycles (1982-2011) of air temperature and vegetation parameters in westerlies-dominated (column 1), East Asian monsoon-dominated (columns 2-4) and Indian monsoon-dominated (columns 5-6) TP basins.

**Figure 7**. Seasonal cycles (1982-2011) of snow cover and snow water equivalent (SWE) in westerlies-dominated (column 1), East Asian monsoon-dominated (columns 2-4) and Indian monsoon-dominated (columns 5-6) TP basins. The snow cover was





extracted from cloud free snow composite product during the period 2005-2013. It
should also be noted that the GlobSnow data are not available for some basins.
**Figure 8**. Sen's slopes of water budget components and vegetation parameters in
westerlies-dominated TP basins during the period of 1982-2011. The double red stars
showed that the trend was statistically significant at the 0.05 level.
**Figure 9**. Linear trends of westerly, Indian monsoon and East Asian summer monsoon
during the period 1982-2011 revealed prospectively by the Asian Zonal Circulation
Index, Indian Ocean Dipole Mode Index and East Asian Summer Monsoon Index.
**Figure 10**. Similar to Figure 8 but for East Asian monsoon-dominated TP basins. It
should be noted that the GlobSnow data are not available for some basins. The double
red stars showed that the trend was statistically significant at the 0.05 level.
**Figure 11**. Similar to Figure 8 but for Indian monsoon-dominated TP basins. It should
be noted that the GlobSnow data are not available for some basins.The double red
stars showed that the trend was statistically significant at the 0.05 level.





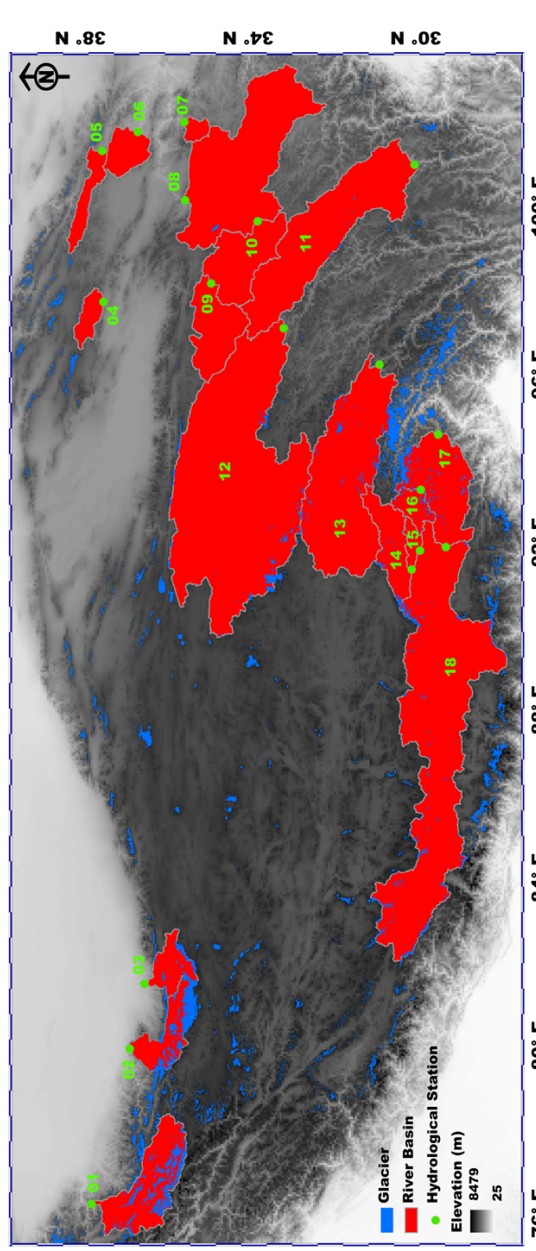

**Figure 1.** Map of river basins and hydrological gauging stations (green dots) over the Tibetan Plateau (TP) used in this study. The grey shading shows the topography of TP in meters above the sea level and the blue shading exhibits the glaciers distribution in TP extracted from the Second Glacier Inventory Dataset of China.






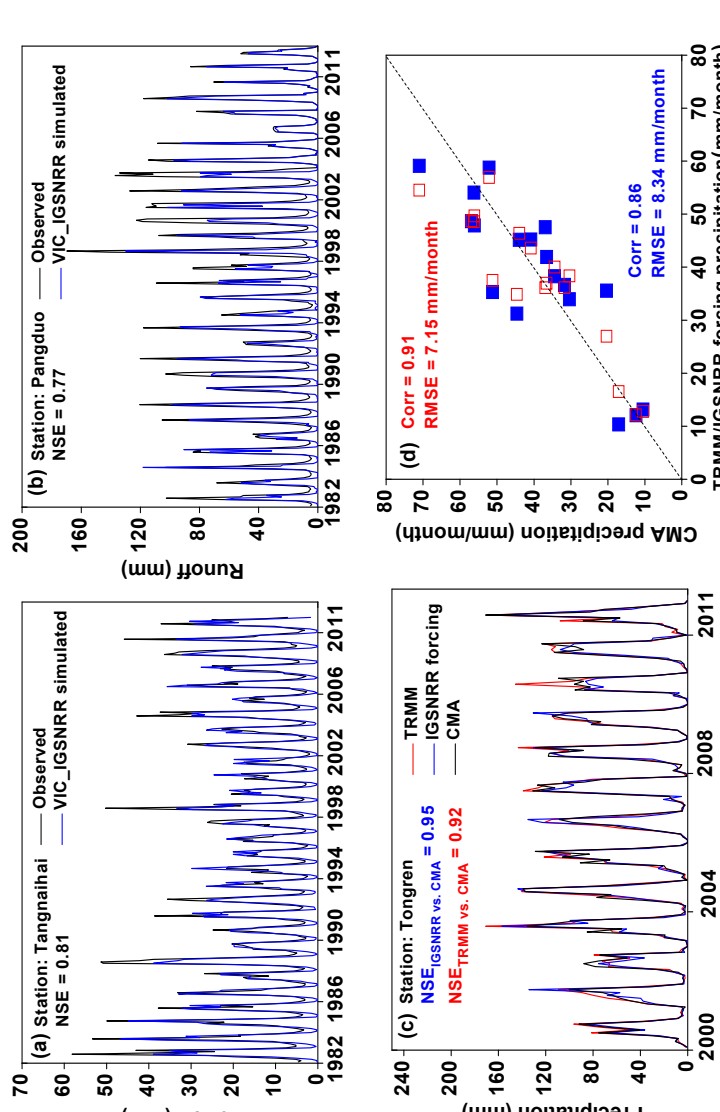

**Figure 2.** Comparison of VIC_IGSNRR simulated and observed monthly runoff for Tangnaihai and Panduo stations (a and b) as well as (c) basin-averaged monthly TRMM, CMA gridded and IGSNRR forcing precipitations for the smallest basin (Tongren station) over the period 1982-2011. (d) shows the comparison of TRMM (blue) and IGSNRR forcing (red) precipitations against CMA gridded precipitation for 18 river basins over TP during the period 2000-2011.





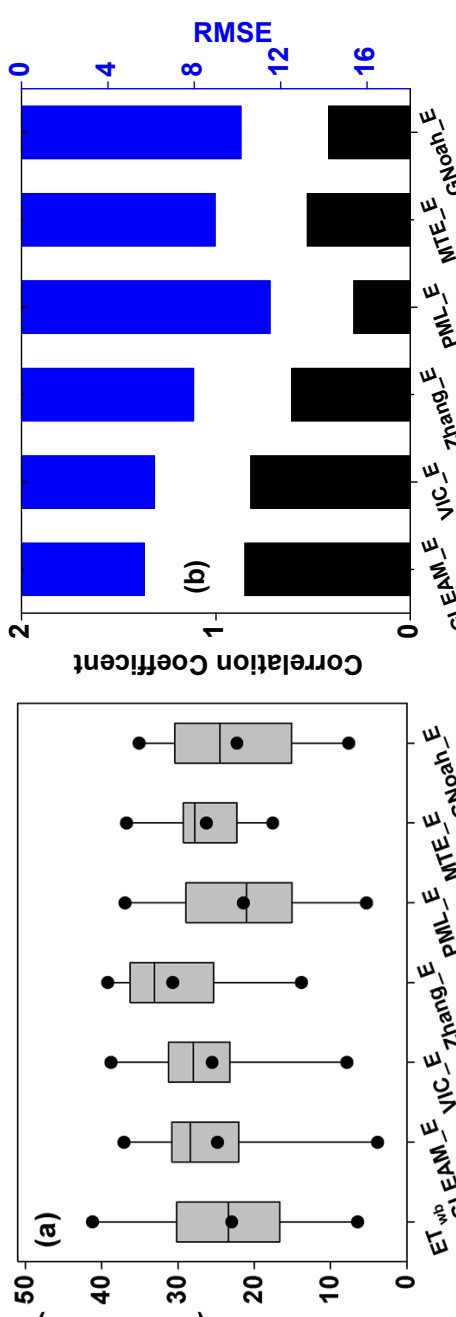

**Figure 3.** Comparison of different ET products against the calculated ET through the water balance ($ET_{wb}$) for 18 river basins over the Tibetan Plateau. The
boxplot of annual estimates of different ET products for 18 TP basins are shown in (a) while the correlation coefficients and root-mean-square-errors (RMSEs,
mm/month) for each ET product relatively to $ET_{wb}$ are exhibited in (b).



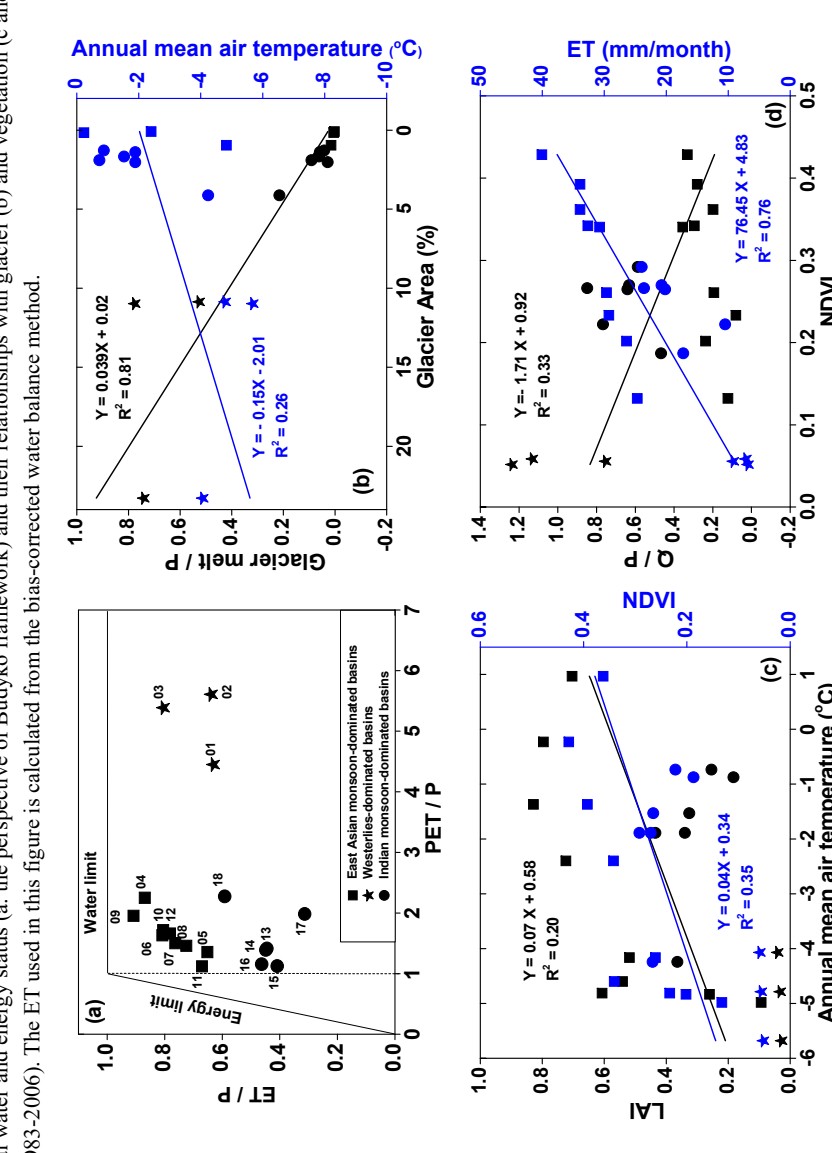

**Figure 4.** General water and energy status (a. the perspective of Budyko framework) and their relationships with glacier (b) and vegetation (c and d) for eighteen TP river basins (1983–2006). The ET used in this figure is calculated from the bias-corrected water balance method.









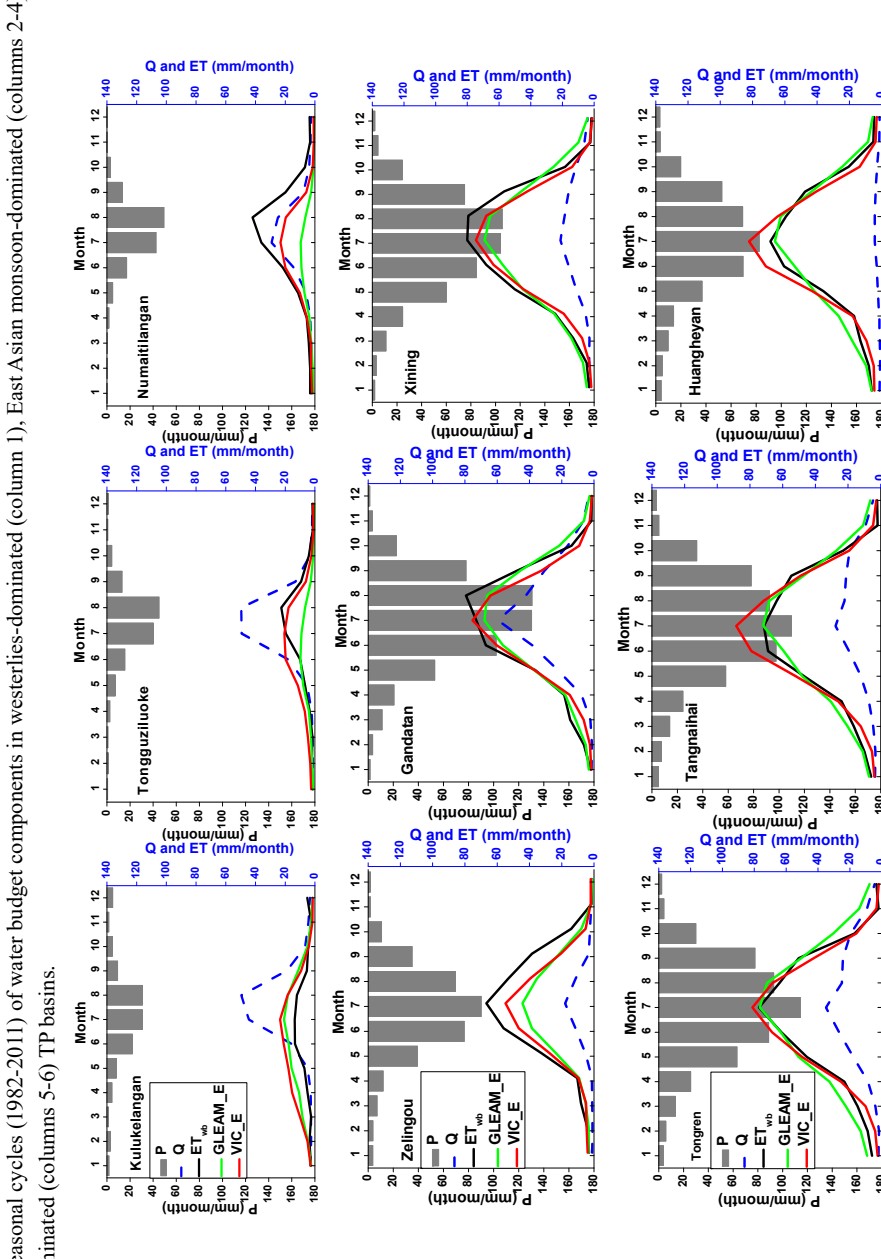

**Figure 5.** Seasonal cycles (1982–2011) of water budget components in westerlies-dominated (column 1), East Asian monsoon-dominated (columns 2–4) and Indian monsoon-dominated (columns 5–6) TP basins.





**Figure 5:** (continued)

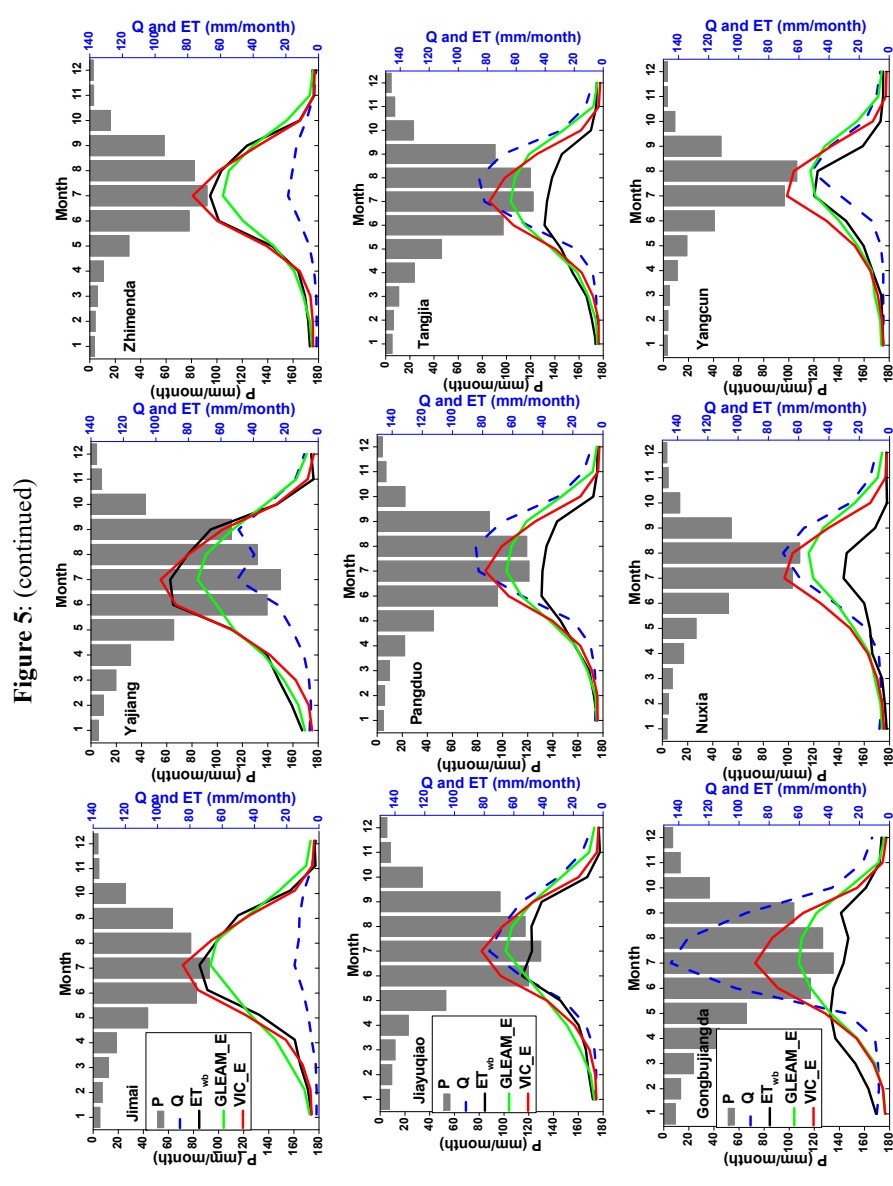





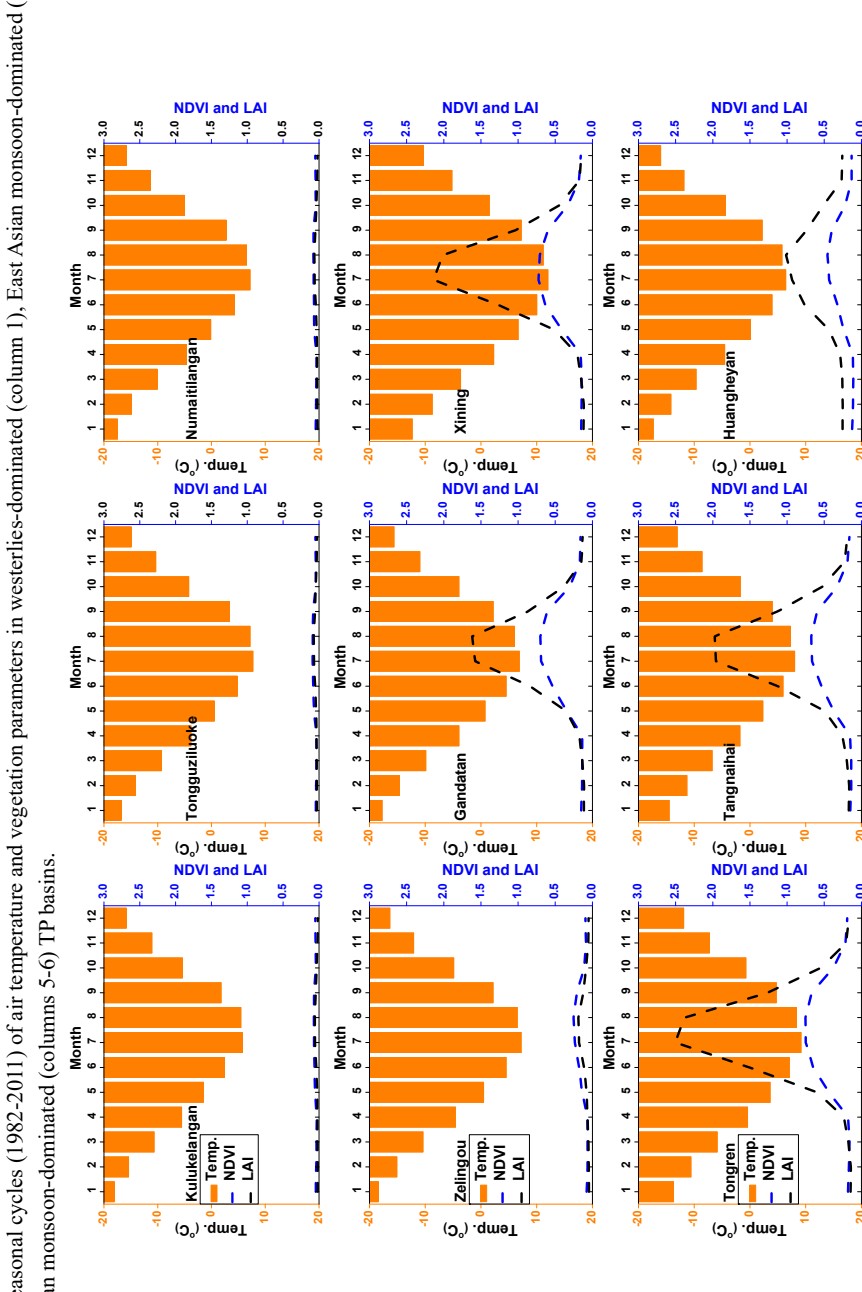

**Figure 6.** Seasonal cycles (1982–2011) of air temperature and vegetation parameters in westerlies-dominated (column 1), East Asian monsoon-dominated (columns 2–4) and Indian monsoon-dominated (columns 5–6) TP basins.



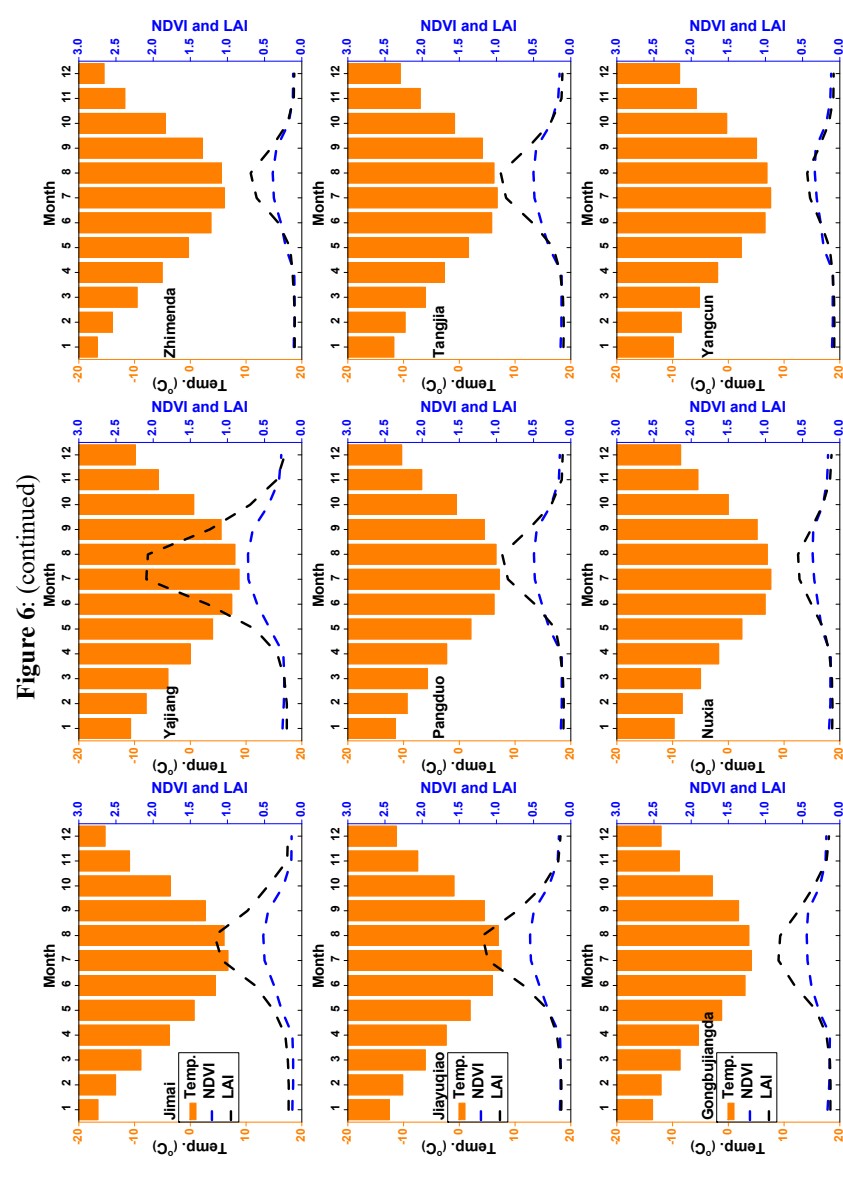

**Figure 6:** (continued)





**Figure 7.** Seasonal cycles (1982–2011) of snow cover and snow water equivalent (SWE) in westerlies-dominated (column 1), East Asian monsoon- dominated (columns 2-4) and Indian monsoon-dominated (columns 5-6) TP basins. The snow cover was extracted from cloud free snow composite product during the period 2005-2013. It should also be noted that the GlobSnow data are not available for some basins.




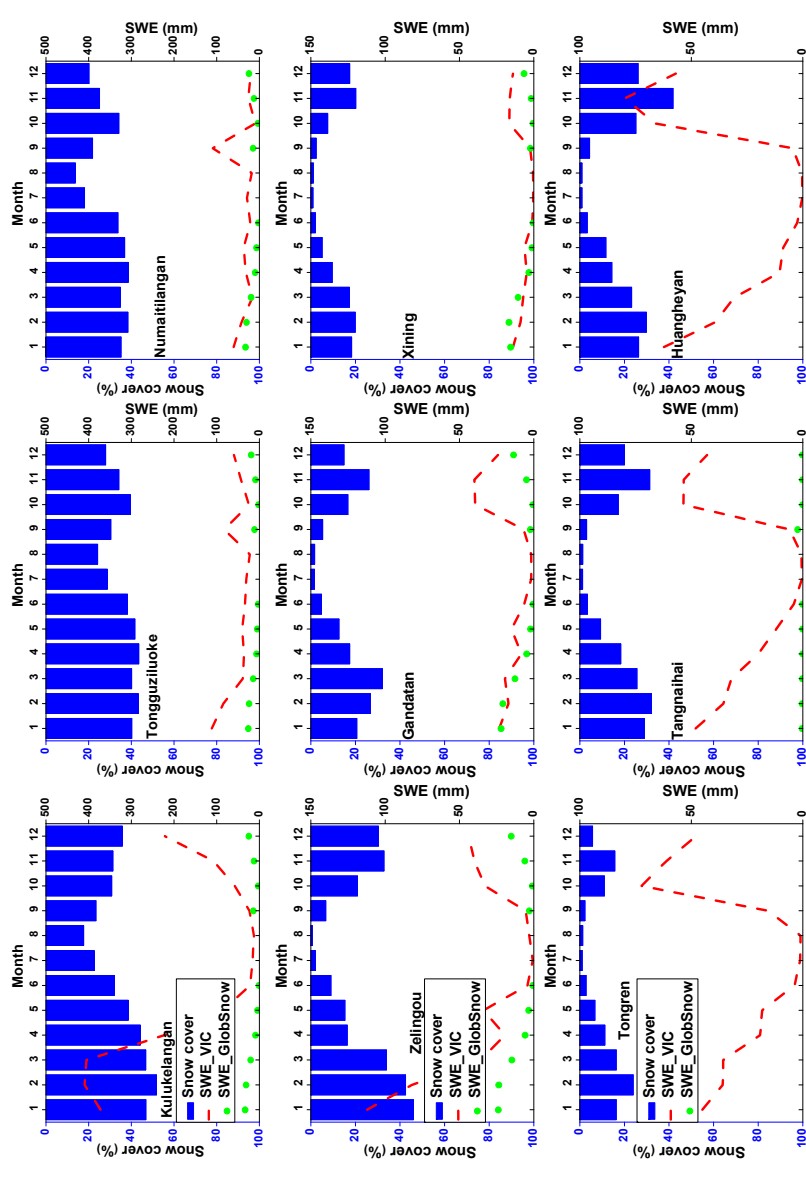





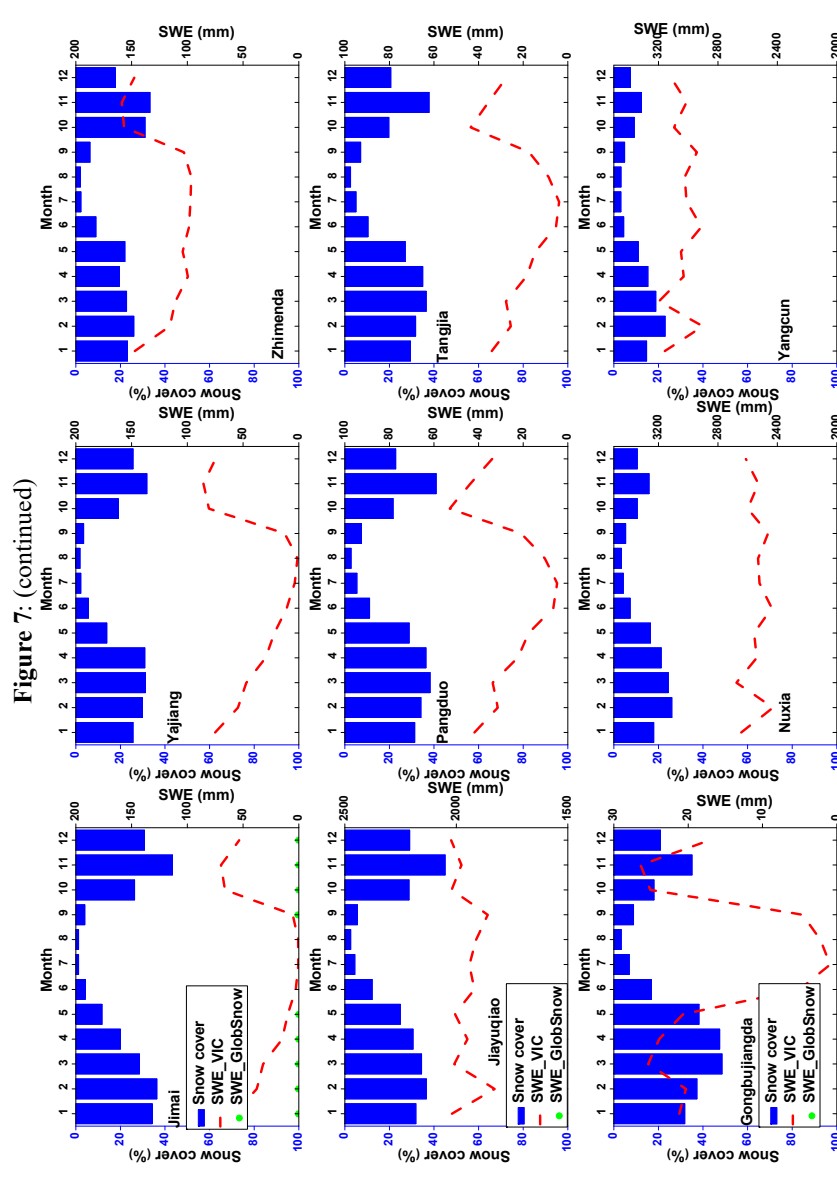

**Figure 7:** (continued)



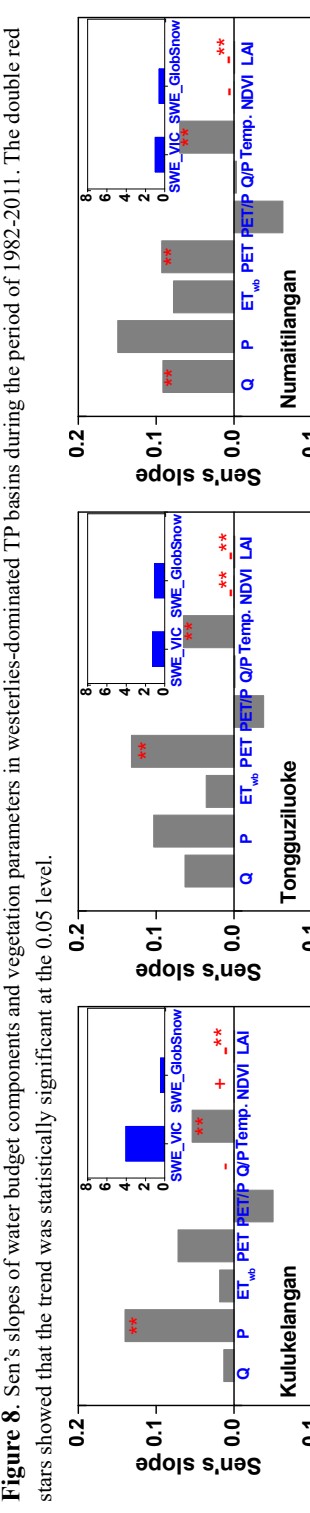

**Figure 8.** Sen's slopes of water budget components and vegetation parameters in westerlies-dominated TP basins during the period of 1982-2011. The double red stars showed that the trend was statistically significant at the 0.05 level.







**Figure 9**. Linear trends of westerly, Indian monsoon and East Asian summer monsoon during the
period 1982-2011 revealed prospectively by the Asian Zonal Circulation Index, Indian Ocean
Dipole Mode Index and East Asian Summer Monsoon Index.




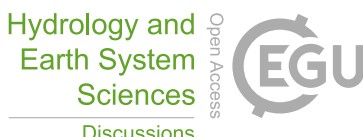



**Figure 10.** Similar to Figure 8 but for East Asian monsoon-dominated TP basins. It should be noted that the GlobSnow data are not available for some basins. The double red stars showed that the trend was statistically significant at the 0.05 level.





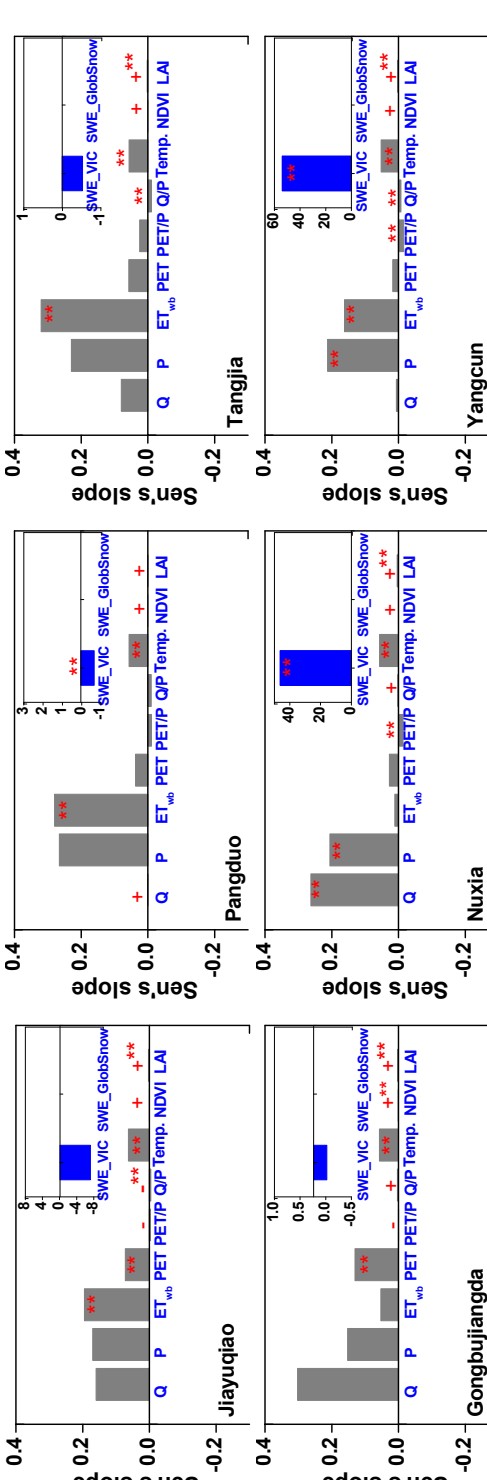

**Figure 11.** Similar to Figure 8 but for Indian monsoon-dominated TP basins. It should be noted that the GlobSnow data are not available for some basins. The double red stars showed that the trend was statistically significant at the 0.05 level.