# Peer review of "Seasonal cycles and trends of water budget components in"

_Hydrology and Earth System Sciences, 2016_

## Referee Comment (RC1) · Anonymous Referee #1 · 9 Dec 2016

Short summary

Liu et al., presented an excellent study that investigates water cycle at Tibetan Plateau. Comparing multiple datasets (model derived product, satellite derived product, in situ observations) is challenging, because of scale mismatch, methodology inconsistency, data quality. Sometime, it's hard to extract consistent and insightful information from multiple datasets. But this study is doing quite good to this end. The major finding is that Tibetan Plateau is becoming wetter as the climate warming up, indicated by both model and data products. This is a high impact finding, which will potentially foster lots of discussion in terms of e.g., ecological consequences, soil biogeochemistry alteration, and green house gases (particularly CH4 in TP region) emissions in response

to the regional wetting.

Strengths

First of all, the paper is well written. Data, method, results, evidences and literature supports are clearly presented. Secondly, the storyline is logistically consistent. Finally, the results are scientifically significant. I like this study very much. Below, I listed a few comments that might help to improve the presentation and results interpretation. But, overall, this study is a well done.

Weaknesses & Major comments

The trend of westerly, Indian monsoon and East Asian summer monsoon (Figure 9) seems not significant. At least, by visually checking, there is no positive trend at all. Is there a rigorous way (or statistical test) to show that the positive trend is detectable and statistically significant?

As climate warming, we expect to see more water coming from glacier-melt. As a result, the contribution of glacier-melt to discharge might go up. While in the Table 3, they are fixed numbers, which may bias the estimate of ET (eqn. 2). I guess to estimate the change of "contribution of glacier-melt to discharge" is technically difficult. But at least, the paper should discuss the uncertainty associated with this particular issue.

Specific comments

L35. Insights -> dynamics

L37 land surface water cycle

L38 list the components, e.g., precipitation, runoff . . . . . .

L38 through the use of -> using

L38 through the water balance -> remove

L47 corresponded to -> consisten with

L47 The general hydrological regimes . . . complex modeling approaches. Not sure need this sentence or not.

L55 environments -> regions

L55 benefit -> beneficial

L65 with ->, from which

L67 , which -> It

L85 to some extend -> remove

L86 due to the lack of quantitative observations of the land surface processes -> remove

L87 break -> overcome

L88 point scale -> in situ

L90 the harsh environment and is often difficult to be applied to -> remove

L91 more popular way -> workaround

L95-97 it is also limited . . . complex terrains. -> sentence change the active voice

L99 In recent years, remove

L100 have been released recently

L148 Is it necessary to mention gauging station here? What's the purpose?

L156 used for water balance calculation for 18 TB basins, remove

L164, which is -> This dataset is

L169 The VIC_IGSNRR . . . .. . is above 0.65. belongs to result section.

L176 used, remove

L179 applied -> used

L182 The CMA precipitation . . . TP conditions. Belongs to result section

L218 Two vegetation parameter datasets, remove

L247 also, remove

L258 traditionally, remove

L272 concluded -> derived

L273 existing studies -> literatures

L284 for 18 basin, remove

L308 Did you use the same method (MM) to quantify the the trend of westerly, Indian monsoon and East Asian summer monsoon (Figure 9)?

L315 monthly performances of, remove

L315 in 18 TP basins against our calculated ET at a monthly basis.

L316 which was calculate through . . . water storage change, remove

L335 the perspective of, remove

L344 The figure also shows a clear vegetation control on ET. higher ENVI -> higher ET. The R2 is highest among those linear regressions.

L404 dryness declined in al basins. This is one of the most significant findings of this study, I guess it warrant more discussion about implications?

L408. Linear trend 0.21. Is this trend statistically significant? What's the p value? L440. Linear thrend is 0.0006, which is tiny.

---

## Author Comment (AC1) · 10 Dec 2016

Review Comments (Anonymous Referee 1): Short summary Liu et al., presented an excellent study that investigates water cycle at Tibetan Plateau. Comparing multiple datasets (model derived product, satellite derived product, in situ observations) is challenging, because of scale mismatch, methodology inconsistency, data quality. Sometime, it's hard to extract consistent and insightful information from multiple datasets. But this study is doing quite good to this end. The major finding is that Tibetan Plateau is becoming wetter as the climate warming up, indicated by both model and data products. This is a high impact finding, which will potentially foster lots of discussion in terms of e.g., ecological consequences, soil biogeochemistry alteration, and green

house gases (particularly CH4 in TP region) emissions in response to the regional wetting.

Strengths First of all, the paper is well written. Data, method, results, evidences and literature supports are clearly presented. Secondly, the storyline is logistically consistent. Finally, the results are scientifically significant. I like this study very much. Below, I listed a few comments that might help to improve the presentation and results interpretation. But, overall, this study is a well done.

Thanks for your invaluable comments. We have revised the manuscript accordingly (please see the point-to-point responses below) based on your suggestions. The following sentence was also added in the acknowledgement section: "We wish to thank the editors and reviewers for their invaluable comments and constructive suggestions to improve the quality of the manuscript". [Line 540-541 in the new version].

Weaknesses Major comments The trend of westerly, Indian monsoon and East Asian summer monsoon (Figure 9) seems not significant. At least, by visually checking, there is no positive trend at all. Is there a rigorous way (or statistical test) to show that the positive trend is detectable and statistically significant?

Thank you very much. We have also detected the non-parametric trends for the indices of westerly, Indian monsoon and East Asian summer monsoon by modified Mann-Kendall test in the revised version based on the reviewer's suggestion (Figure 9 in the new version and Figure R1 in this file). However, the trends for the indices of Asian Zonal Circulation Index and East Asian Summer Monsoon Index were still insignificant. The true trends may be very small (or unchanged) during only the 30 years. Actually, by using these indices, we only want to see the variability of the Indian summer monsoon (not focus on if the trend is significant or not), the westerly and the East Asian summer monsoon and then try to relate to the trends in water balance components in 18 TP rivers. The insignificant trends are also true just like the changes of some water balance components are insignificant as well in such a relatively short period.

[Figure]

As climate warming, we expect to see more water coming from glacier-melt. As a result, the contribution of glacier-melt to discharge might go up. While in the Table 3, they are fixed numbers, which may bias the estimate of ET (eqn. 2). I guess to estimate the change of "contribution of glacier-melt to discharge" is technically difficult. But at least, the paper should discuss the uncertainty associated with this particular issue.

We totally agree with you. Accurate quantification of the contribution of glacier-melt to discharge is very difficult, especially for multi-basins. We have discussed the uncertainty associated with this particular issue in the new version, as suggested by the reviewer, as follows [Line 470-476 in the new version], "...we obtained the contributions of glacier-melt to discharge in some basins from the literatures and took them as fixed numbers. It may inherit considerable uncertainty from varied studies using different approaches such as glacier mass-balance observation, isotope observation and hydrological modeling, and the contribution rates would also change under a warming climate. However, accurate quantification of the contribution of glacier-melt to discharge is technically difficult nowadays, especially for the data-sparse basins...".

Specific comments L35. Insights -> dynamics L37 land surface water cycle L38 list the components, e.g., precipitation, runoff : : :: : : L38 through the use of -> using L38 through the water balance -> remove L47 corresponded to -> consisten with

Revised, thanks!

L47 The general hydrological regimes : : : complex modeling approaches. Not sure need this sentence or not.

We have removed this sentence in the new version.

L55 environments -> regions L55 benefit -> beneficial L65 with ->, from which L67 , which -> It L85 to some extend -> remove L86 due to the lack of quantitative observations of the land surface processes -> remove L87 break -> overcome L88 point scale -> in situ L90 the harsh environment and is often difficult to be applied to -> remove

L91 more popular way -> workaround

Done!

L95-97 it is also limited : : : complex terrains. -> sentence change the active voice

We have revised this sentence as active voice as follows [Line 90-92 in the new version], ". . . it is still difficult to use land surface models to multiple basins especially to the relatively smaller ones under complex terrains due to the lack of adequate data for model calibration and validation. . ."

L99 In recent years, remove L100 have been released recently L156 used for water balance calculation for 18 TB basins, remove L164, which is -> This dataset is L176 used, remove L179 applied -> used L218 Two vegetation parameter datasets, remove L247 also, remove L258 traditionally, remove L272 concluded -> derived L273 existing studies -> literatures L284 for 18 basin, remove L315 monthly performances of, remove L315 in 18 TP basins against our calculated ET at a monthly basis. L316 which was calculate through : : : water storage change, remove L335 the perspective of, remove

Revised, thanks!

L148 Is it necessary to mention gauging station here? What's the purpose?

We totally agree with the reviewer. This sentence has been removed in the new version.

L169 The VIC$_{IGSNRR}$ :::::: $is above 0.65. belongs to result section. L182 The CMA precipitation$ ::: $TP conditions. Belongs to result section$

We have removed them to the results section as suggested by the reviewer [Line 299-308 in the new version], thanks.

L344 The figure also shows a clear vegetation control on ET. higher ENVI -> higher ET. The R2 is highest among those linear regressions.

We totally agree with you. We have detailed explained this phenomenon in the revised version as follows [Line 337-340 in the new version]: "...The R2 between basin-averaged NDVI and ET is 0.76 which shows a clear vegetation control on ET in 18 TP basins. The result is in line with Shen et al. (2015), which indicated that the spatial pattern of ET trend was significantly and positively correlated with NDVI trend over TP..."

L404 dryness declined in all basins. This is one of the most significant findings of this study, I guess it warrant more discussion about implications?

We have added more discussions for this finding in the new version as follows [Line 400-405 in the new version]: "...Although P and PET were found both increase since the 1980s (Shi et al., 2003; Yao et al., 2014), the declined PET/P is, to some extent, attributed to the ascending P exceed the increase in PET for these basins (except for the Yulongkashi basin). The climate moistening in the headwaters of these inland rivers would be beneficial to the water resources and oasis agro-ecosystems in the middle and lower basins....".

L308 Did you use the same method (MM) to quantify the the trend of westerly, Indian monsoon and East Asian summer monsoon (Figure 9)? L408. Linear trend 0.21. Is this trend statistically significant? What's the p value?

Thanks, we have also detected the non-parametric trends for the indices of westerly, Indian monsoon and East Asian summer monsoon by modified Mann-Kendall test in the revised version. Also, we have revised this Figure (Figure R1 in this file) by adding both the linear and non-parametric trends as well as their corresponding P-values. However, the trends for the indices of Asian zonal circulation index and East Asian summer monsoon index were still insignificant. The true trends may be very small (or unchanged) during only the 30 years. Actually, by using these indices, we only want to see the variability of the Indian summer monsoon (not focus on if the trend is significant or not), the westerly and the East Asian summer monsoon and then try to relate to the trends in water balance components in 18 TP rivers. The insignificant trends are also

true just like the changes of some water balance components are insignificant as well in such a relatively short period.

Linear thrend is 0.0006, which is tiny.

We have recalculated the linear trend at the annual time scale in the revised version and also redrawn the figures (Figure R1 in this file). The linear and non-parametric trends are both 0.01. The linear trend is insignificant while the non-parametric trend is significant at the 0.05 level.  
* * *
[Figure]
[Figure]

*Figure R1. Linear and non-parametric trends of westerly, Indian monsoon and East Asian summer monsoon during the period 1982-2011 revealed prospectively by the Asian Zonal Circulation Index, Indian Ocean Dipole Mode Index and East Asian Summer Monsoon Index.*

[Figure]

**Fig. 1.** Figure R1

[Figure]

---

## Referee Comment (RC2) · Anonymous Referee #2 · 4 Jan 2017

The paper presents a very interesting study, by which the authors investigated general water budgets and trends in water balance components in 18 river basins in the data-sparse Tibetan Plateau from the perspective of multi-sources datasets. In my opinion, it is a good attempt to understand the hydrological regimes in TP basins in the big data era. The manuscript is overall well-organized and should fall into the aims and scopes of HESS. I do not find major problems with this manuscript and recommend accepting it only after a few minor revisions.

Major Comments: (1) In the methodology, I suggest to present more detail about "the modified MK method". You may think to add it as an appendix. (2) Though an exhaustive literature review has been done. However, some recently published ones are

missing, for example, Xiang et al. (2016); Dong et al. (2016).

Xiang, L., Wang, H., Steffen, H., Wu, P., Jia, L., Jiang, L., and Shen, Q.: Groundwater storage changes in the Tibetan Plateau and adjacent areas revealed from GRACE satellite gravity data, Earth Planet Sc Lett, 449, 228-239, 2016. Dong, W., Lin, Y., Wright, J. S., Ming, Y., Xie, Y., Wang, B., Luo, Y., Huang, W., Huang, J., Wang, L., Tian, L., Peng, Y., and Xu, F.: Summer rainfall over the southwestern Tibetan Plateau controlled by deep convection over the Indian subcontinent, Nature Communications, 7, 10925, 2016.

Details points: (1) Language: The language is clear. Throughout the manuscript, I suggest to add "the" before TP or Tibetan Plateau or other proper nouns. Please do check them. (2) line 84: "locates" to "located" (3) Line 131-137: Instead of describing the structure of this paper, you should indicate the aim/objective of this study. (4) Line 182: As the mean annual precipitation ranges from 128-717mm, the RMSE (8.34mm/month=100mm/year) seems not so "perfect". Please tone down this sentence. (5) line 238, which interpolation method was used? Linear interpolation or others? (6) line 263, what is the unit for MG? It should also be specified as P and Q. (7) line 330, how did you classify the basins into westerlies-dominated, Indian monsoon-dominated and East Asian monsoon-dominated basins? (8) line 836, Unit of RMSE in Figure 3b should be added. (9) Figure 10, in the subplot for Xining station, the Q/P declined/increased or closed to zero?

---

## Referee Comment (RC3) · Anonymous Referee #3 · 4 Jan 2017

The authors exploited different sources of data to look at the variability and trend of water budget of the Tibetan Plateau. I find the paper generally well written, but language editing is required throughout the paper to fix the typos and grammar before the paper can be published. I will not give specific comments but the authors need to make good efforts to fix the language.

The paper is logically clear and gives some invaluable insights about the hydrology in the TP. However, while working with multiple datasets, the authors did not fully describe the advantage and disadvantages of each dataset in applying to the TP region, provided that these global data sets from either models or satellites have their own weakness when applied to the TP area. In particular, it's well known that land surface

models have some difficulties when applying to TP (e.g., parameter tuning in boundary layer schemes), even though they have good performances in different regimes.

I think the paper is not doing well on uncertainty analysis in the water balance estimation and trend detection. In fact, no uncertainty assessment is done at all. The authors acknowledged that the multi-source data sets have their own uncertainties biases, but failed to address the implications in their analysis. In the trend analysis, it is unclear whether the self correlation is removed, and what uncertainties are associated with the derived trends.

The tables and figures are high quality.

Figure 5, 6 and 7 show very similar seasonal behaviors in the hydrology and meteorology between the basins. So why divide the regions to these basins?

Figure 9, what is $R^2$ here? Do you need to remove low frequency in the indices before calculating trends?

---

## Author Comment (AC2) · 4 Jan 2017

Review Comments (Anonymous Referee #2): Short summary The paper presents a very interesting study, by which the authors investigated general water budgets and trends in water balance components in 18 river basins in the data-sparse Tibetan Plateau from the perspective of multi-sources datasets. In my opinion, it is a good attempt to understand the hydrological regimes in TP basins in the big data era. The manuscript is overall well-organized and should fall into the aims and scopes of HESS. I do not find major problems with this manuscript and recommend accepting it only after a few minor revisions.

Thank a lot for your invaluable comments/suggestions. We have revised the manuscript

accordingly (please see the point-to-point response below). Also, the following sentence was added in the acknowledgement section [Line 549-550 in the new version]: "We wish to thank the editors and reviewers for their invaluable comments and constructive suggestions to improve the quality of the manuscript".

Major Comments: In the methodology, I suggest to present more detail about "the modified MK method". You may think to add it as an appendix.

We totally agree with you. In the revised version, we have presented more details for the MMK approach as follows [Line 291-306 in the new version], "...Pre-whitening is often used to eliminate the influence of lag-1 autocorrelation before the use of MK test, for example, in pre-whitening, the analyzed time series $(X_1, X_2, ..., X_n)$ will be replaced by $(X_2-cX_1, X_3-cX_2, ..., X_{(n+1)}-cX_n)$ if the lag-1 autocorrelation coefficient (c) is larger than 0.1 (von Storch, 1995). However, significant lag-i autocorrelation may still be detected after pre-whitening because only the lag-1 autocorrelation is considered in pre-whitening (Zhang et al., 2013). Moreover, it sometimes underestimate the trend for a given time series (Yue et al., 2002). Hamed and Rao (1998) proposed a modified version of MK test (MMK) to consider the lag-i autocorrelation and related robustness of the autocorrelation through the use of equivalent sample size, which has been widely used in previous studies during the last five decades (McVicar et al., 2012; Zhang et al., 2013; Liu and Sun, 2016). In the MMK approach, if the lag-i autocorrelation coefficients are significantly distinct from zero, the original variance of MK statistics will be replaced by the modified one. In this study, we used the MMK approach to quantify the trends of water budget components in18 TP basins and the significance of trend was tested at the >95% confidence level..."

Though an exhaustive literature review has been done. However, some recently published ones are missing, for example, Xiang et al. (2016); Dong et al. (2016).

Xiang, L., Wang, H., Steffen, H., Wu, P., Jia, L., Jiang, L., and Shen, Q.: Groundwater storage changes in the Tibetan Plateau and adjacent areas revealed from GRACE

satellite gravity data, Earth Planet Sc Lett, 449, 228-239, 2016.

Dong, W., Lin, Y., Wright, J. S., Ming, Y., Xie, Y., Wang, B., Luo, Y., Huang, W., Huang, J., Wang, L., Tian, L., Peng, Y., and Xu, F.: Summer rainfall over the southwestern Tibetan Plateau controlled by deep convection over the Indian subcontinent, Nature Communications, 7, 10925, 2016.

Thanks, we have downloaded/read/cited them at the proper places in the new version.

Details points: (1) Language: The language is clear. Throughout the manuscript, I suggest to add "the" before TP or Tibetan Plateau or other proper nouns. Please do check them.

We have double-checked and added "the" before TP, Tibetan Plateau or other proper nouns throughout the manuscript. Thank you very much.

(2) line 84: "locates" to "located"

Done!

(3) Line 131-137: Instead of describing the structure of this paper, you should indicate the aim/objective of this study.

Actually, we have indicated the objective of this study in Line 123-126 in the manuscript.

(4) Line 182: As the mean annual precipitation ranges from 128-717mm, the RMSE (8.34mm/month=100mm/year) seems not so "perfect". Please tone down this sentence.

We have revised this sentence as follows [Line 313-317 in the new version], thank you very much. "Moreover, the CMA precipitation is consistent with TRMM (Corr = 0.86, RMSE = 8.34 mm/month) and IGSNRR forcing (Corr = 0.94, RMSE = 7.15mm/month) precipitation for multiple basins (and also for the smallest basin above Tongren station, Fig.2), which reveals the applicably of CMA precipitation under the TP conditions."

(5) line 238, which interpolation method was used? Linear interpolation or others?

We used the bilinear interpolation. We have described the method in the revised version as follows to make it clearer [Line 222-224 in the new version]. "All gridded datasets used were first uniformly interpolated to a spatial resolution of 0.5 degree based on the bilinear interpolation to make their inter-comparison possible."

(6) line 263, what is the unit for MG? It should also be specified as P and Q.

The unit of MG is mm. We have specified it as P and Q in the new version.

(7) line 330, how did you classify the basins into westerlies-dominated, Indian monsoondominated and East Asian monsoon-dominated basins?

The basins were classified through the locations of the categorized climate zones (Tian et al. 2007; Yao et al., 2012) in which the basin rested. Actually, the boundaries of climate zones are not fixed with the monsoon strengthening/weakening and the extent of certain basin may sometimes not entirely rest in one climate zone. We thus just approximately classified them in order to generally link the basin-scale results to different climate regimes in this study.

References: Tian, L., Yao, T., MacClune, K., White, J.W.C.., Schilla, A., Vaughn, B., Vachon, R., and Ichiyanagi, K.: Stable isotopic variations in west China: a consideration of moisture sources, J. Geophys. Res. Atmos., 112, D10112, 2007. Yao, T.D., Thompson, L., Yang, W., Yu, W.S., Gao, Y., Guo, X.J., Yang, X.X., Duan, K.Q., Zhao, H.B., Xu, B.Q., Pu, J.C., Lu, A.X., Xiang, Y., Kattel, D.B., and Joswiak, D.: Different glacier status with atmospheric circulations in Tibetan Plateau and surroundings, Nat. Clim. Change, 2, 1-5, 2012.

(8) line 836, Unit of RMSE in Figure 3b should be added.

We have added the unit of RMSE (please see the Figure R2 in this file) in the revised version. Thank you very much.

(9) Figure 10, in the subplot for Xining station, the Q/P declined/increased or closed to zero?

Yes, the calculated trend equaled to zero. Say, the Q/P showed no trend at Xining station in Figure 10.
* * *
[Figure]

*Figure R1. Comparison of different ET products against the calculated ET through the water balance (ETwb) for 18 river basins over the Tibetan Plateau. The boxplot of annual estimates of different ET products for 18 TP basins are shown in (a) while the correlation coefficients and root-mean-square-errors (RMSEs, mm/month) for each ET product relatively to ETwb are exhibited in (b).*

[Figure]

**Fig. 1.**

---

## Author Comment (AC3) · 4 Jan 2017

**Seasonal cycles and trends of water budget components in 18 river basins across the Tibetan Plateau: a multiple datasets perspective**

Wenbin Liu[a], Fubao Sun[a*], Yanzhong Li[a], Guoqing Zhang[b,c],

Yan-Fang Sang[a], Jiahong Liu[d], Hong Wang[a] ,Peng Bai[a]

[revised manuscript text omitted]

2016). Therefore, the insights of water balance over various TP river basins located at different monsoon-dominant regions are still unclear so far (Cuo et al.,

2014; Xu et al., 2016). One way to  overcome this limitation is to install more instruments to measure the  in situ water budgets (Yang et al., 2013; Zhou et al., 2013; Ma et al., 2015), but it is extremely expensive to maintain long-term observations at  basin or regional scales. Another  workaround is to simulate basin-wide water budgets through physical-based land surface models at several large river basins forced with remote sensing data and large-scale gridded meteorological forcing datasets (Bookhagen and Burbank, 2010; Xue et al., 2013; Zhang et al., 2013; Cuo et al., 2015; Zhou et al., 2015; Wang et al., 2016). However, it is still difficult to use land surface models to multiple basins especially to the relatively smaller ones under complex terrains due to 
[revised manuscript text omitted]

Pre-whitening is often used to eliminate the influence of lag-1 autocorrelation before the use of MK test, for example, in pre-whitening, the analyzed time series

$(X_1, X_2, \ldots, X_n)$ will be replaced by $(X_2 - cX_1, X_3 - cX_2, \ldots, X_{n+1} - cX_n)$ if the lag-1

autocorrelation coefficient (c) is larger than 0.1 (von Storch, 1995). However, significant lag-i autocorrelation may still be detected after pre-whitening because only the lag-1 autocorrelation is considered in pre-whitening (Zhang et al., 2013).

Moreover, it sometimes underestimate the trend for a given time series (Yue et al.,

2002). Hamed and Rao (1998) proposed a modified version of MK test (MMK) to consider the lag-i autocorrelation and related robustness of the autocorrelation through the use of equivalent sample size, which has been widely used in previous studies during the last five decades (McVicar et al., 2012; Zhang et al., 2013; Liu and Sun, 2016).

In the MMK approach, if the lag-i autocorrelation coefficients are significantly distinct from zero, the original variance of MK statistics will be replaced by the modified one. ~In this study, we used a modified version of MK test (MMK, Hamed and Rao, 1998)the MMK approach to quantify the trends of water budget components in18 TP basins-- and the significance of trend was tested at the >95% confidence level. The MMK considers the lag-*i* autocorrelation and related robustness of the autocorrelation, which has been widely used in previous studies during the last five decades (McVicar et al., 2012; Liu and Sun, 2016).

**3  Results and Discussion**

**3.1 ET evaluation and General hydrological characteristics of 18 TP basins**

[revised manuscript text omitted]

$ET_{wb}$ all ascended with regional warming (Fig.8), especially in the Keliya River basin (Numaitilangan station). The aridity index (PET/P), which is an indicator for the degree of dryness, slightly declined in all basins in northwestern TP. Although P and

PET were found both increase since the 1980s

 (Shi et al., 2003; Yao et al., 2014), the declined PET/P is, to some extent, attributed to the ascending P exceed the increase in PET for these basins (except for the Yulongkashi basin). The climate moistening in the headwaters of these inland rivers would be beneficial to the water resources and oasis agro-ecosystems in the middle and lower basins. –The increase in streamflow was also found in most tributaries of the Tarim River (Sun et al., 2006; Fu et al., 2010; Mamat et al., 2010).

Moreover, the westerlies, revealed by the Asian Zonal Circulation Index ($60^{o}$-$150^{o}$ E), slightly enhanced (linear trend: 0.21) over the period of 1982-2011 (Fig.9). More water vapor was transported and fell as precipitation or snow in northwestern TP (e.g., the eastern Pamir region) with the strengthening westerlies. Both SWE products (VIC_IGSNRR simulated and GlobaSnow-2 product) showed slightly increase for all basinsThe SWE showed increase for all basins and for both products (VIC_IGSNRR

[revised manuscript text omitted]

1-5, 2012.

Yao, Y.J., Zhao, S.H., Zhang, Y.H., Jia, K., and Liu, M.: Spatial and decadal variations in potential evapotranspiration of China based on reanalysis datasets during 1982-2010, Atmosphere, 5,

737-754, 2014.

Yin, G., Hu, Z.Y., Chen, X., and Tiyip, T.: Vegetation dynamics and its response to climate change in Central Asia, J. Arid Land, 8, 375, 2016.

Yu, J., Zhang, G., Yao, T., Xie, H., Zhang, H., Ke, C., and Yao, R.: Developing daily cloud-free snow composite products from MODIS Terra-Aqua and IMS for the Tibetan Plateau, IEEE

Trans. Geosci. Remote Sens., 54(4), 2171-2180, 2015.

Yue, S., Pilon, P., Phinney, B., Cavadias, G.: The influence of autocorrelation on the ability to detect trend in hydrological series, Hydrol. Process., 16(9), 1807-1829, 2002.

Zhang, D., Liu, X., Zhang, Q., Liang, K., and Liu, C.: Investigation of factors affecting intea-annual variability of evapotranspiration and streamflow under different climate conditions.

J. Hydrol., doi:10.1016/j.jhydrol.2016.10.047, 2016.

Zhang, G., Xie, H., Yao, T., Liang, T., and Kang, S.: Snow cover dynamics of four lake basins over Tibetan Plateau using time series MODIS data (2001-2100), Water Resour. Res., 48(10),

W10529, 2012.

Zhang, K., Kimball, J.S., Nemani, R.R., and Running, S.W.: A continuous satellite-derived global record of land surface evapotranspiration from 1983 to 2006, Water Resour. Res., 46(9),

W09522, 2010.

Zhang, L., Su, F., Yang, D., Hao, Z., and Tong, K.: Discharge regime and simulation for the upstream of major rivers over Tibetan Plateau, J. Geophys. Res. Atmos., 118(15), 8500-8518,

2013.

Zhang, Q., Li, J., Singh, V., and Xu, C.: Copula-based spatial-temporal patterns of precipitation extremes in China, Int. J. Climatol., 33, 1140-1152, 2013.

Zhang, X., Tang, Q., Pan, M., and Tang, Y.: A long-term land surface hydrologic fluxes and states dataset for China, J. Hydrometeorol., 15, 2067-2084, 2014.

Zhang, Y., Peña-Arancibia, J.L., McVicar, T.R., Chiew, F.H.S., Vaze, J., Liu, C.M., Lu, X.J.,

Zheng, H.X., Wang, Y.P., Liu, Y.Y., Miralles, D.G., and Pan, M.: Multi-decadal trends in global terrestrial evapotranspiration and its components, Scientific Reports, 6, 19124, 2016.

Zhang, Y., Liu, C., Tang, Y., and Yang, Y.: Trend in pan evaporation and reference and actual evapotranspiration across the Tibetan Plateau, J. Geophys. Res., 112, D12110, 2007.

Zhou, C., Jia, S., Yan, H., and Yang, G.: Changing trend of water resources in Qinghai Province from 1956 to 2000, J. Glaciol. Geocryol., 27(3), 432-437, 2005 (in Chinese).

Zhou, J., Wang, L., Zhang, Y.S., Guo, Y.H., Li, X.P., and Liu, W.B.: Exploring the water storage changes in the largest lake (Selin Co) over the Tibetan Plateau during 2003-2012 from a basin-wide hydrological modeling,. Water Resour. Res., 51, 8060-8086, 2015.

Zhou, S.Q., Kang, S., Chen, F., and Joswiak, D.R.: Water balance observations reveal significant subsurface water seepage from Lake Nam Co., south-central Tibetan Plateau,. J. Hydrol., 491,

89-99, 2013.

Zhou, S.Q., Wang, Z., and Joswiak, D.R.:From precipitation to runoff: stable isotopic fractionation effect of glacier melting on a catchment scale, Hydrol. Process., 28(8), 3341-3349, 2014.

Zhu, Y., Chen, J., Chen, G.: Runoff variation and its impacting factors in the headwaters of the

Yangtze River in recent 32 years, J.Yangtze River Sci. Res. Inst., 28(6), 1-4, 2011 (in Chinese ).

**Table 1**: Main features of the 18 used TP river basins. GA% and SC% represent the percentages of multiyear-mean glacier cover and snow cover in each basin. The glacier and snow cover data are extracted, respectively, from the Second Glacier Inventory Dataset of China and the daily TP snow cover dataset (2005-2013)

[revised manuscript text omitted]

---

## Author Comment (AC4) · 4 Jan 2017

**Seasonal cycles and trends of water budget components in 18 river basins across the Tibetan Plateau: a multiple datasets perspective**

Wenbin Liu[a], Fubao Sun[a*], Yanzhong Li[a], Guoqing Zhang[b,c],

Yan-Fang Sang[a], Jiahong Liu[d], Hong Wang[a] ,Peng Bai[a]

[revised manuscript text omitted]

$(X_2 - cX_1, X_3 - cX_2, ..., X_{n+1} - cX_n)$ if the lag-1 autocorrelation coefficient (c) is larger than 0.1 (von Storch, 1995). However, significant lag-i autocorrelation may still be detected after pre-whitening because only the lag-1 autocorrelation is considered in pre-whitening (Zhang et al., 2013). Moreover, it sometimes underestimate the trend for a given time series (Yue et al., 2002). Hamed and Rao (1998) proposed a modified version of MK test (MMK) to consider the lag-i autocorrelation and related robustness of the autocorrelation through the use of equivalent sample size, which has been widely used in previous studies during the last five decades (McVicar et al., 2012;

Zhang et al., 2013; Liu and Sun, 2016). In the MMK approach, if the lag-i autocorrelation coefficients are significantly distinct from zero, the original variance of MK statistics will be replaced by the modified one. In this study, we used the

MMK approach to quantify the trends of water budget components in18 TP basins and the significance of trend was tested at the >95% confidence level.

**3    Results and Discussion**

**3.1 ET evaluation and General hydrological characteristics of 18 TP basins**

In this study, we first assess the VIC_IGSNRR simulated runoff against the observations for each basin (for example, at Tangnaihai and Pangduo stations in

Fig.2). The VIC_IGSNRR simulated runoff is acceptable and could be used to replace the missing values for a given basin, if the Nash Efficiency coefficient (NSE) between the observation and simulation is above 0.65. Moreover, the CMA precipitation is consistent with TRMM (Corr = 0.86, RMSE = 8.34 mm/month) and IGSNRR forcing (Corr = 0.94, RMSE = 7.15mm/month) precipitation for multiple basins (and also for the smallest basin above Tongren station, Fig.2), which reveals the applicably of CMA

precipitation under the TP conditions.

                  < Figure 2, here please, thanks>

We then evaluated six ET products in 18 TP basins against our calculated $ET_{wb}$  at a monthly basis (Fig. 3). The ranges of monthly averaged ET among different basins (approximately 4−39 mm month$^{-1}$) are very close for all products compare with that calculated from the  $ET_{wb}$(6−42 mm month$^{-1}$). However, GLEAM_E (correlation coefficient: Corr = 0.85 and root-mean-square-error: RMSE = 5.69 mm month$^{-1}$) and

VIC_E (Corr = 0.82 and RMSE = 6.16 mm month$^{-1}$) perform relatively better than others. Although Zhang_E and GNoah_E were found closely correlated to monthly $ET_{wb}$  in the upper Yellow River, the upper Yangtze River, Qiangtang and

Qaidam basins (Li X. et al., 2014), they did not exhibit overall good performances (Corr = 0.61, RMSE = 7.97 mm month$^{-1}$ for Zhang_E and Corr = 0.42, RMSE =

10.16 mm month$^{-1}$ for GNoah_E) for 18 TP basin used in this study. We thus use

GLEAM_E and VIC_E together with  $ET_{wb}$  to calculate the seasonal cycles and trends of ET in 18 TP basins in the following sections.

       **14 / 52**

To investigate the general hydroclimatic characteristics of rivers over the TP, we classify 18 basins into three categories, namely westerlies-dominated basins (Yerqiang, Yulongkashi and Kelia), Indian monsoon-dominated basins (Brahmaputra and Salween), and East Asian monsoon-dominated basins (Yellow, Yalong and

Yangtze) referred to Tian et al. (2007), Yao et al. (2012) and Dong et al. (2016).

Interestingly, they are clustered into three groups under Budyko framework (Budyko,

1974; Zhang D. et al., 2016) with relatively lower evaporative index for Indian monsoon-dominant basins and higher aridity index for westerlies-dominant basins, which reveal various long-term hydroclimatologic conditions (Fig. 4). Overall, the annual mean air temperature increases (-5.68 ~0.97 $^{o}$C) while multiyear mean glacier area (and thus the glacier melt normalized by precipitation) decreases (23.27 ~ 0%)

gradually from the westerlies-dominant, Indian monsoon-dominant to East Asian monsoon-dominant basins. The vegetation status (NDVI range: 0.05~0.43; LAI range:

0.03~0.83) tends to be better and ET increases (and thus runoff coefficient gradually decreases) from cold to warm basins (Fig. 4 and Table 1). The $R^2$ between basin-averaged NDVI and ET is 0.76 which shows a clear vegetation control on ET in

18 TP basins. The result is in line with Shen et al. (2015), which indicated that the spatial pattern of ET trend was significantly and positively correlated with NDVI

[revised manuscript text omitted]

$ET_{wb}$ all ascended with regional warming (Fig.8), especially in the Keliya River basin (Numaitilangan station). The aridity index (PET/P), which is an indicator for the degree of dryness, slightly declined in all basins in northwestern TP. Although P and

PET were found both increase since the 1980s (Shi et al., 2003; Yao et al., 2014), the declined PET/P is, to some extent, attributed to the ascending P exceed the increase in

PET for these basins (except for the Yulongkashi basin). The climate moistening in the headwaters of these inland rivers would be beneficial to the water resources and oasis agro-ecosystems in the middle and lower basins. The increase in streamflow was also found in most tributaries of the Tarim River (Sun et al., 2006; Fu et al., 2010;

Mamat et al., 2010). Moreover, the westerlies, revealed by the Asian Zonal

[revised manuscript text omitted]

1-5, 2012.

Yao, Y.J., Zhao, S.H., Zhang, Y.H., Jia, K., and Liu, M.: Spatial and decadal variations in potential evapotranspiration of China based on reanalysis datasets during 1982-2010, Atmosphere, 5,

737-754, 2014.

Yin, G., Hu, Z.Y., Chen, X., and Tiyip, T.: Vegetation dynamics and its response to climate change in Central Asia, J. Arid Land, 8, 375, 2016.

Yu, J., Zhang, G., Yao, T., Xie, H., Zhang, H., Ke, C., and Yao, R.: Developing daily cloud-free snow composite products from MODIS Terra-Aqua and IMS for the Tibetan Plateau, IEEE

Trans. Geosci. Remote Sens., 54(4), 2171-2180, 2015.

Yue, S., Pilon, P., Phinney, B., Cavadias, G.: The influence of autocorrelation on the ability to detect trend in hydrological series, Hydrol. Process., 16(9), 1807-1829, 2002.

[revised manuscript text omitted]

---

## Author Comment (AC5) · 5 Jan 2017

Review Comments (Anonymous Referee #3): The authors exploited different sources of data to look at the variability and trend of water budget of the Tibetan Plateau. I find the paper generally well written, but language editing is required throughout the paper to fix the typos and grammar before the paper can be published. I will not give specific comments but the authors need to make good efforts to fix the language.

Thank you very much for your invaluable comments/suggestions. We have revised the manuscript accordingly (please see the point-to-point response below). The typos and grammar were double-checked and revised throughout the new version. Also, the following sentence was added in the acknowledgement section [Line 558-559 in the

new version]: "We wish to thank the editors and reviewers for their invaluable comments and constructive suggestions to improve the quality of the manuscript".

The paper is logically clear and gives some invaluable insights about the hydrology in the TP. However, while working with multiple datasets, the authors did not fully describe the advantage and disadvantages of each dataset in applying to the TP region, provided that these global data sets from either models or satellites have their own weakness when applied to the TP area. In particular, it's well known that land surface models have some difficulties when applying to TP (e.g., parameter tuning in boundary layer schemes), even though they have good performances in different regimes.

Thanks. We totally agree with you that the advantage/disadvantages of each global dataset in applying to the TP regions should be described. However, nowadays these global datasets (e.g., different ET products, SWE products, NDVI and LAI) has rarely been comprehensively assessed in the TP due to the lack of in situ observations. The advantages/disadvantages of different datasets are thus difficult to be fully described. Moreover, detailed validations of these products in the TP are beyond the scope of this study, which need be further investigated in the future works.

We have also tried to add more details in the uncertainty section through summarizing the limited studies available as follows [Line 473-485 in the new version]: "...due to their uncertainties inherited from different forcing data, algorithm used and varied spatial-temporal resolutions (Xue et al., 2013; Li et al., 2014; Liu W et al., 2016a). In particular, it is well known that land surface models have some difficulties (e.g., parameter tuning in boundary layer schemes) when applying to the TP (Xia et al., 2012; Bai et al., 2016). For example, Xue et al. (2013) indicated that GNoah_E underestimated theãĂŰ ETãĂŮ_wb in the upper Yellow River and Yangtze River basins on the Tibetan Plateau mainly due to its negative-biased precipitation forcing.The VIC_IGSNRR simulated and GlobaSnow-2 snow water equivalents have also not been validated in the TP due to the lack of in situ observations. However, they showed similar seasonal cycles and annual trends in some basins such as Zelinggou and Numaitilangan, which revealed the applicability of the SWE products for these basins. . ."

I think the paper is not doing well on uncertainty analysis in the water balance estimation and trend detection. In fact, no uncertainty assessment is done at all. The authors acknowledged that the multi-source data sets have their own uncertainties biases, but failed to address the implications in their analysis. In the trend analysis, it is unclear whether the self correlation is removed, and what uncertainties are associated with the derived trends..

This study may unavoidably associate with some uncertainty due to the use of multi-source datasets. We totally agree with the reviewer that the uncertainty should be quantified as well in the analysis. We have actually tried to consider the uncertainty in the analysis, for example, we compared the observed CMA precipitation with TRMM and IGSNRR_forcing data during 2000-2011. The water balance-based ETwb was also compared with other six global/regional ET products (including the mean and annual trends) during the period 1982-2006. Moreover, to minimize the uncertainty in the analysis we only analyzed the well-performed ET products together with the observed runoff, precipitation and ETwb of basic water balance components during 1982-2011.

However, adequately quantification of uncertainty for each water budget component is difficult. When we focused on the analysis of one variable during the period 1982-2011, few datasets can be used together in the TP to quantify its uncertainty due to the data availability. For example, we have observed CMA precipitation from 1982-2011, but the TRMM precipitation is only available since 2000. We can also calculate ETwb for the period 1982-2011, but Zhang_E is only available from 1983-2006. Moreover, the global datasets for NDVI, LAI, SWE and water storage changes are also limited which, to some extent, restricted our attempts to quantify the uncertainties in the analysis using multi-source datasets.

(2) For the trend analysis, we used the modified Mann-Kendall test which can consider (remove) the lag-i autocorrelation and related robustness of the autocorrelation through the use of equivalent sample size. We have added the following details to describe the method to make it more clearly, especially for its consideration on the self correlation [Line 289-306 in the new version],

"...Pre-whitening is often used to eliminate the influence of lag-1 autocorrelation before the use of MK test, for example, in pre-whitening, the analyzed time series $(X_1, X_2, ..., X_n)$ will be replaced by $(X_2$-ãĂŰcXãĂŮ$_1, X_3$-ãĂŰcXãĂŮ$_2, ..., X_{(n+1)}$-ãĂŰcXãĂŮ$_n)$ if the lag-1 autocorrelation coefficient ($c$) is larger than 0.1 (von Storch, 1995). However, significant lag-$i$ autocorrelation may still be detected after pre-whitening because only the lag-1 autocorrelation is considered in pre-whitening (Zhang et al., 2013). Moreover, it sometimes underestimate the trend for a given time series (Yue et al., 2002). Hamed and Rao (1998) proposed a modified version of MK test (MMK) to consider the lag-$i$ autocorrelation and related robustness of the autocorrelation through the use of equivalent sample size, which has been widely used in previous studies during the last five decades (McVicar et al., 2012; Zhang et al., 2013; Liu and Sun, 2016). In the MMK approach, if the lag-$i$ autocorrelation coefficients are significantly distinct from zero, the original variance of MK statistics will be replaced by the modified one. In this study, we used the MMK approach to quantify the trends of water budget components in18 TP basins and the significance of trend was tested at the >95% confidence level..."

The tables and figures are high quality. Figure 5, 6 and 7 show very similar seasonal behaviors in the hydrology and meteorology between the basins. So why divide the regions to these basins?

The TP climate is influenced by the westerly, East Asian summer monsoon and Indian summer monsoon (Tian et al. 2007; Yao et al., 2012), thus the basin-scale hydrological regimes may also different in different climate zones. Actually, in Figure 5 (first row), the hydrological regimes (e.g., precipitation amount and distribution, snow cover) in the westerly-dominated basins are different from those rest in other climate zones. We classified the basins in order to generally link the basin-scale results to different climate regimes in this study.

References: Tian, L., Yao, T., MacClune, K., White, J.W.C.., Schilla, A., Vaughn, B., Vachon, R., and Ichiyanagi, K.: Stable isotopic variations in west China: a consideration of moisture sources, J. Geophys. Res. Atmos., 112, D10112, 2007. Yao, T.D., Thompson, L., Yang, W., Yu, W.S., Gao, Y., Guo, X.J., Yang, X.X., Duan, K.Q., Zhao, H.B., Xu, B.Q., Pu, J.C., Lu, A.X., Xiang, Y., Kattel, D.B., and Joswiak, D.: Different glacier status with atmospheric circulations in Tibetan Plateau and surroundings, Nat. Clim. Change, 2, 1-5, 2012.

Figure 9, what is RЁĘ2 here? Do you need to remove low frequency in the indices before calculating trends?

We have deleted the $R^2$ and added the P-values for the trends detected in the revised version (figure R3 in this file). Moreover, we have also added the results of non-parametric trend detected by the modified Mann-Kendall test, which can consider (remove) the lag-i autocorrelation and related robustness of the autocorrelation through the use of equivalent sample size.

Please also note the supplement to this comment:
http://www.hydrol-earth-syst-sci-discuss.net/hess-2016-624/hess-2016-624-AC5-supplement.pdf
* * *
[Figure]

*Figure R3. Linear and non-parametric trends of westerly, Indian monsoon and East Asian summer monsoon during the period 1982-2011 revealed prospectively by the Asian Zonal Circulation Index, Indian Ocean Dipole Mode Index and East Asian Summer Monsoon Index.*

**Fig. 1.**

**Supplement:**

**Seasonal cycles and trends of water budget components in 18 river basins across the Tibetan Plateau: a multiple datasets perspective**

Wenbin Liu[a], Fubao Sun[a*], Yanzhong Li[a], Guoqing Zhang[b,c],

Yan-Fang Sang[a], Jiahong Liu[d], Hong Wang[a] ,Peng Bai[a]

[a]Key Laboratory of Water Cycle and Related Land Surface Processes, Institute of Geographic

Sciences and Natural Resources Research, Chinese Academy of Sciences, Beijing 100101, China

[b]Key Laboratory of Tibetan Environmental Changes and Land Surface Processes, Institute of

Tibetan Plateau Research, Chinese Academy of Sciences, Beijing 100101, China

[c]CAS Center for Excellent in Tibetan Plateau Earth Sciences, Beijing 100101, China

[d]Key Laboratory of Simulation and Regulation of Water Cycle in River Basin, China Institute of

Water Resources and Hydropower Research, Beijing 100038, China

**Submitted to**: Hydrology and Earth System Sciences

**Corresponding Author**: Dr. Fubao Sun (Sunfb@igsnrr.ac.cn), from the Key Laboratory of Water

Cycle and Related Land Surface Processes, Institute of Geographic Sciences and Natural

Resources Research, Chinese Academy of Sciences (No. A11, Datun Road, Chaoyang District,

Beijing 100101, China)

**Email Addresses for other authors**: Wenbin Liu (liuwb@igsnrr.ac.cn), Yanzhong Li (liyz.14b@igsnrr.ac.cn), Guoqing Zhang (guoqing.zhang@itpcas.ac.cn), Yan-fang Sang (sangyf@igsnrr.ac.cn), Jiahong Liu (liujh@iwhr.com) , Hong Wang (wanghong@igsnrr.ac.cn) ,

Peng Bai (baip.11b@igsnrr.ac.cn)

2016/11/25

**Highlights**

- Monthly basin-wide ET was calculated through water balance considering the impacts of glacier and water storage change

- Water budget components and trends for 18 river basins over the TP were evaluated

- Uncertainties were discussed from multiple dataset perspective

**Abstract.** The  dynamics of water budget over the Tibetan Plateau (TP) are not fully understood so far due to the lack of quantitative observations of the land surface water cycle. Here, we investigated the seasonal cycles and trends of water budget components, e.g., precipitation, runoff and evapotranspiration (ET), in

TP basins using multi-source datasets during the period

1982-2011. A two-step bias correction procedure was applied to calculate the basin-wide  considering the influences of glacier and water storage change. The results indicated that precipitation, which mainly concentrated during June-October (varied among different monsoons impacted basins), is the major contributor to the runoff in the TP basins. The basin-wide snow water equivalent (SWE) was relatively higher from mid-autumn to spring for most TP basins. The water cycles intensified under a global warming in most basins except for the upper Yellow and Yalong Rivers, which were significantly influenced by the weakening East Asian monsoon. Consistent with the climate warming and moistening in the TP and western China, the aridity index (PET/P) in most basins decreased.

 The results highlighted the usefulness of integrating the multi-source data (e.g., in situ observations, remote sensing products, reanalysis, land surface model simulations and climate model outputs) for hydrological applications in the data-sparse regions and could be  beneficial for understanding the water and energy budgets, sustainable management of water resources under a warming climate in the harsh and the data-sparse Tibetan Plateau.

**1 Introduction**

As the highest plateau in the globe (the average elevation is higher than 4000 meters above the sea level), the Tibetan Plateau (TP, also called "the roof of the world" or "the third Pole") is one of the most vulnerable region under a warming climate and is subjected to strong interactions among atmosphere, hydrosphere, biosphere and cryosphere in the earth system (Duan and Wu, 2006; Yao et al., 2012; Liu W. et al., 2016b). It also serves as the "Asian water tower"  from which many major Asian rivers such as Yellow River, Yangtze River, Brahmaputra River, Mekong River, Indus River, etc., originate   It provides a vital water resource to support hundreds of millions of people in China and the surrounding countries (Immerzell et al., 2010; Zhang et al., 2013). Knowledge about the water budgets and their responses to the changing environment is thus crucial for understanding the hydrological regimes and for sustainable water resources management as well as environmental protection in this special region (Yang et al., 2014; Chen et al., 2015).

The TP is also known as a typical data-sparse mountain region which brings great challenges to hydrological and related land surface studies (Zhang et al., 2007; Li F. et al., 2013; Liu X. et al., 2016). For example, since the 1950s, totally 750 stations have been established over China by the Chinese Meteorological Administration (CMA), among which only less than 80 stations are distributed over the plateau (Wang and

Zeng, 2012). They are primary sparse and unevenly located at relatively low elevation regions, focus only on the meteorological variables and lack of other land surface observations such as evapotranspiration, snow water equivalent and latent heat fluxes, etc.. In addition, long-term consecutive observations of river discharge, snow depth, lake depth and glacier melts in the TP are also absent (Akhta et al., 2009; Ma et al.,

2016). Therefore, the insights of water balance over various TP river basins locates located at different monsoon-dominant regions are, to some extent, still unclear so far due to the lack of quantitative observations of the land surface processes (Cuo et al.,

2014; Xu et al., 2016). One way to break  overcome this limitation is to install more instruments to measure the point scalein situ water budgets (Yang et al., 2013; Zhou et al., 2013; Ma et al., 2015), but it is extremely expensive to maintain long-term observations at the harsh environment and is often difficult to be applied to basin or regional scales. Another more popular wayworkaround is to simulate basin-wide water budgets through physical-based land surface models at several large river basins forced with remote sensing data and large-scale gridded meteorological forcing datasets (Bookhagen and Burbank, 2010; Xue et al., 2013; Zhang et al., 2013; Cuo et al., 2015; Zhou et al., 2015; Wang et al., 2016). However, it is still difficult to use land surface models to multiple basins especially to the relatively smaller ones under complex terrains due to 
[revised manuscript text omitted]
. , but  it is sometimes  influenced by the serial correlation of time series. Pre-whitening is often used to eliminate the influence of lag-1 autocorrelation before the use of MK test, for example, in pre-whitening, the analyzed time series $(X_1, X_2, ..., X_n)$ will be replaced by $(X_2 - cX_1, X_3 - cX_2, ..., X_{n+1} - cX_n)$ if the lag-1 autocorrelation coefficient (c) is larger than 0.1 (von Storch, 1995). However, significant lag-i autocorrelation may still be detected after pre-whitening because only the lag-1 autocorrelation is considered in pre-whitening (Zhang et al., 2013). Moreover, it sometimes underestimate the trend for a given time series (Yue et al.,

2002). Hamed and Rao (1998) proposed a modified version of MK test (MMK) to consider the lag-$i$ autocorrelation and related robustness of the autocorrelation through the use of equivalent sample size, which has been widely used in previous studies during the last five decades (McVicar et al., 2012; Zhang et al., 2013; Liu and Sun, 2016).

In the MMK approach, if the lag-$i$ autocorrelation coefficients are significantly distinct from zero, the original variance of MK statistics will be replaced by the modified one. In this study, we used  the MMK approach to quantify the trends of water budget components in18 TP basins and the significance of trend was tested at the >95% confidence level.

**3 Results and Discussion**

**3.1 ET evaluation and General hydrological characteristics of 18 TP basins**

In this study, we first assess the VIC_IGSNRR simulated runoff against the observations for each basin (for example, at Tangnaihai and Pangduo stations in Fig.2). The VIC_IGSNRR simulated runoff is acceptable and could be used to replace the missing values for a given basin, if the Nash Efficiency coefficient (NSE) between the observation and simulation is above 0.65. Moreover, the CMA precipitation is consistent with TRMM (Corr = 0.86, RMSE = 8.34 mm/month) and IGSNRR forcing (Corr = 0.94, RMSE = 7.15mm/month) precipitation for multiple basins (and also for the smallest basin above Tongren station, Fig.2), which reveals the applicably of CMA precipitation under the TP conditions.

We  then evaluated  six ET products in 18 TP basins against our calculated $ET_{wb}$ at a monthly basis, (Fig. 3).

[revised manuscript text omitted]

$ET_{wb}$ all ascended with regional warming (Fig.8), especially in the Keliya River basin (Numaitilangan station). The aridity index (PET/P), which is an indicator for the degree of dryness, slightly declined in all basins in northwestern TP. Although P and

PET were found both increase since the 1980sThe results were in line with the overall climate warming and moistening reported in northwest China (Shi et al., 2003; Yao et al., 2014), at which these basins located., the declined PET/P is, to some extent, attributed to the ascending P exceed the increase in PET for these basins (except for the Yulongkashi basin). The climate moistening in the headwaters of these inland rivers would be beneficial to the water resources and oasis agro-ecosystems in the middle and lower basins. –The increase in streamflow was also found in most tributaries of the Tarim River (Sun et al., 2006; Fu et al., 2010; Mamat et al., 2010). Moreover, the westerlies, revealed by the Asian Zonal Circulation Index ($60^o$-$150^o$ E), slightly enhanced (linear trend: 0.21) over the period of 1982-2011 (Fig.9). More water vapor was transported and fell as precipitation or snow in northwestern TP (e.g., the eastern Pamir region) with the strengthening westerlies. Both SWE products (VIC_IGSNRR simulated and GlobaSnow-2 product) showed slightly increase for all basinsThe SWE showed increase for all basins and for both products (VIC_IGSNRR simulated and GlobaSnow-2 product) with the incremental seasonal snow cover and advanced glaciers (Yao et al., 2012). More precipitation was transformed into snow or glacier and the runoff coefficient (Q/P) exhibited decrease although precipitation obviously increased (Fig.8). In addition, the transpiration in these basins may decrease with vegetation degradation revealed by the NDVI and LAI (Yin et al., 2016) but the atmospheric evaporative demand indicated by CRU PET increased (significantly increase in the Yulongkashi and Keliya rivers) during the period 1982-2011.

< Figure 8, here please, thanks>

< Figure 9, here please, thanks>

In the East Asian monsoon-dominated basins, there are two types of change for basin-wide water budget components. For example, P and Q slightly decreased in the upper Yellow River (Tangnihai, Huangheyan and Jimai stations) and Yalong River (Yajiang station) but increased in other basins (Zelingou, Gandatan, Xining, Tongren and Zhimenda stations) over the period of 1982-2011 (Fig.10). The decline in Q and P for the upper Yellow and Yalong Rivers (locate at the eastern Tibetan Plateau) were consistent with that found by Cuo et al. (2013, 2014) as well as Yang et al. (2014), and were in line with the weakening (linear slope: -0.01) of the East Asian Summer

Monsoon (Fig.9). The vegetation turned green while $ET_{wb}$ and PET increased in all nine basins with the significantly ascending air temperature during the period

1982-2011. The aridity index (PET/P) was found decrease in all basins except for the upper Yellow River basin above Jimai station and the upper Yalong River basin above

Yajiang station. Moreover, theboth the runoff coefficients and SWE (SWE) were decrease (decrease except for the Bayin River above Zelingou station and the upper

Yellow River above Tongren station) in the East Asian monsoon dominated basins.

< Figure 10, here please, thanks>

The hydrological cycles were also found intensified in the Indian monsoon-dominated basins such as Salween River and Brahmaputra River (Fig.11), which were in line with the strengthen (linear trend: 0.000601) of the Indian Summer monsoon (revealed by the Indian Ocean Dipole Mode Index) during the specific period 1982-2011 (Fig.9).

In the six basins, trends in P, Q and $ET_{wb}$ were all upward. For example, at

Jiayuqiao station, the annual streamflow showed slightly increasing trend which was consistent with that examined during 1980-2000 by Yao et al. (2012). The vegetation status, revealed by NDVI and LAI, turned better significantly with the ascending air temperature. The aridity index (PET/P) decreased in all basins except for the

Brahmaputra River above Tangjia station, which indicated that most basins in the

Indian monsoon-dominated regions turn wet over the period of 1982-2011. The runoff coefficient (Q/P) increased at Gongbujiangda and Nuxia while decreased at Jiayuqiao,

Pangduo, Tangji and Yangcun stations. Moreover, the basin-wide SWE declined in the upper Salween River and Brahmaputra River above Pangduo, Tangjia and

Gongbujiangda stations while increased in Brahmaputra River above Nuxia and

Yangcun stations.

     < Figure 11, here please, thanks>

**3.4 Uncertainties**

The results may unavoidably associate with several aspects of  uncertainty which mainly inherited from the multi-source datasets used. Although both GLEAM_E and VIC_E captured the  seasonal cycles of $ET_{wb}$ , they still have considerable uncertainties  at such as Numaitilangan, Gongbujiangda and Nuxia stations (Fig.5). With respect to the annual trend of $ET_{wb}$ (Table 4), most ET products (including the well-performed GLEAM_E and VIC_E in some basins) cannot detect the decreasing trends in 7 out of 18 basins (e.g., at Kulukelangan, Tongguziluoke, Xining, Tongren, Jimai, Nuxia and Gongbujiangda stations) due to their uncertainties inherited from different forcing data, algorithm used and varied spatial-temporal resolutions (Li et al., 2014; Liu W et al., 2016a). In particular, it is well known that land surface models have some difficulties (e.g., parameter tuning in boundary layer schemes) when applying to the TP (Xia et al., 2012; Bai et al., 2016). For example, Xue et al. (2013) indicated that GNoah_E underestimated the $ET_{wb}$ in the upper Yellow River and Yangtze River basins on the Tibetan Plateau mainly due to its negative-biased precipitation forcing. We thus only used $ET_{wb}$ in the trend detection of water budget components in Fig.8, Fig.10 and Fig.11 in this study.

 The VIC_IGSNRR simulated and GlobaSnow-2 snow water equivalents have also not been validated in the TP due to the lack of in situ observations. However, they showed similar seasonal cycles and annual trends in some basins such as Zelinggou and Numaitilangan, which revealed the applicability of the SWE products for these basins. Moreover, the interpolation of missing values of runoff with VIC_IGSNRR simulated runoff and the gridded precipitation data (which interpolated from limited gauged precipitation over the plateau) involved some uncertainties as well as. Finally, we obtained the contributions of glacier-melt to discharge in some basins from the literatures and took them as fixed numbers. It may inherit considerable uncertainty from varied studies using different approaches such as glacier mass-balance observation, isotope observation and hydrological modeling, and the contribution rates would also change under a warming climate. However, accurate quantification of the contribution of glacier-melt to discharge is technically difficult nowadays, especially for the data-sparse basins.

[revised manuscript text omitted]

Shen, M.G., Piao, S.L., Jeong, S., Zhou, L.M., Zeng, Z.Z., Ciais, P., Chen, D.L., Huang, M.T., Jin, C.S., Li, L.Z.X., Li, Y., Myneni, R.B., Yang, K., Zhang, G.X., Zhang, Y.J., and Yao, T.D.: Evporative cooling over the Tibetan Plateau induced by vegetation growth, Proc. Natl. Acad. Sci. U. S.A., 112(30), 9299-9304, 2015.

Shi, Y.F., Shen, Y.P., Li, D.L., Zhang, G.W., Ding, Y.J., Hu, R.J., and Kang, E.S.: Discussion on the present climate change from Warm2dry to Warm2wet in northwest China, Quat. Sci., 23(2), 152-164, 2003 (in Chinese).

Shepard, D.S.: Computer mapping: the SYMAP interpolation algorithm. Spatial Statistics and Models, G.L. Gaile and C.J. Willmott, Eds., D. Reidel, 133-145, 1984.

Sun, B., Mao, W., Feng, Y., Chang, T., Zhang, L., and Zhao, L.: Study on the change of air temperature, precipitation and runoff volume in the Yarkant River basin, Arid Zone Res., 23(2), 203-209, 2006 (in Chinese).

Takala, M., Luojus, K., Pulliainen, J., Derksen, C., Lemmetyinen, J., Kärnä, J.-P, Koskinen, J., and Bojkov, B.: Estimating northern hemisphere snow water equivalent for climate research through assimilation of spaceborne radiometer data and ground-based measurements, Remote Sens. .Environ., 115 (12), 3517-3529, 2011.

Tapley, B.D., Bettadpur, S., Watkins, M., and Rand eigber, C.: The gravity recovery and climate experiment: mission overview and early results, Geophys. Res. Lett., 31, L09607, 2004.

Tian, L., Yao, T., MacClune, K., White, J.W.C.., Schilla, A., Vaughn, B., Vachon, R., and Ichiyanagi, K.: Stable isotopic variations in west China: a consideration of moisture sources, J. Geophys. Res. Atmos., 112, D10112, 2007.

Tucker, C.J., Pinzon, J.E., Brown, M.E., Slayback, D., Pak, E.W., Mahoney, R., Vermote, E., and El Saleous, N.: An extended AVHRR 8 km NDVI data set compatible with MODIS and SPOT vegetation NDVI data, Int. J. Remote Sens., 26(20), 4485-4498, 2005.

von Storch, H.: Misuses of statistical analysis in climate research, In Analysis of Climate Variability: Applications of Statistical Techniques, Springer-Verlag: Berlin, 11-26, 1995.

Wang, A. and Zeng, X.:Evaluation of multireanalysis products within site observations over the Tibetan Plateau, J. Geophys. Res., 117, D05102, 2012.

Wang, L., Sun, L.T., Shrestha, M., Li, X.P., Liu, W.B., Zhou, J., Yang, K., Lu, H., and Chen, D.L.: Improving snow process modeling with satellite-based estimation of near-surface-air-temperature lapse rate, J. Geophys. Res. Atmos., 121, 12005-12030, 2016.

Xia, Y., Mitchell, K., Ek, M., Cosgrove, B., Sheffield, J., Luo, L., Alonge, C., Wei, H., Meng, J., Livneh, B., and Duang, Q.: Continental-scale water and energy flux analysis and validation for North American Land Data Assimilation System project phase 2 (NLDAS-2): 2. Validation of model-simulated streamflow, J. Geophys. Res. Atmos., 117(D3), D03110, 2012.

Xiang, L., Wang, H., Steffen, H., Wu, P., Jia, L., Jiang, L., and Shen, Q.: Groundwater storage changes in the Tibetan Plateau and adjacent areas revealed from GRACE satellite gravity data, Earth Planet. Sci. Lett., 449, 228-239, 2016.

Xu, L.: The land surface water and energy budgets over the Tibetan Plateau, Available from Nature Precedings < http://hdl.handle.net/10101/npre.2011.5587.1>, 2011.

Xue, B.L., Wang, L., Yang, K., Tian, L., Qin, J., Chen, Y., Zhao, L., Ma, Y., Koike, T., Hu, Z., and Li, X.P.: Modeling the land surface water and energy cycle of a mesoscale watershed in the central Tibetan Plateau with a distributed hydrological model, J. Geophys. Res. Atmos., 118, 8857-8868, 2013.

Yao, Z., Duan, R., and Liu, Z.: Changes in precipitation and air temperature and its impacts on runoff in the Nujiang River basins. Resour. Sci. 34(2), 202-210, 2012 (in Chinese)

Yang, K., Qin, J., Zhao, L., Chen, Y.Y., Tang, W.J., Han, M.L., Lazhu, Chen, Z.Q., Lv, N., Ding, B.H., Wu, H., and Lin, C.G.: A multi-scale soil moisture and freeze-thaw monitoring network on the third pole, Bull. Am. Meteorol. Soc., 94,1907-1916, 2013.

Yang, K., Wu, H., Qin, J., Lin, C.G., Tang, W.J., and Chen, Y.Y.: Recent climate changes over the Tibetan Plateau and their impacts on energy and water cycle: a review, Glob. Planet Change, 112, 79-91, 2014.

Yao, T.D., Thompson, L., Yang, W., Yu, W.S., Gao, Y., Guo, X.J., Yang, X.X., Duan, K.Q., Zhao, H.B., Xu, B.Q., Pu, J.C., Lu, A.X., Xiang, Y., Kattel, D.B., and Joswiak, D.: Different glacier status with atmospheric circulations in Tibetan Plateau and surroundings, Nat. Clim. Change, 2, 1-5, 2012.

Yao, Y.J., Zhao, S.H., Zhang, Y.H., Jia, K., and Liu, M.: Spatial and decadal variations in potential evapotranspiration of China based on reanalysis datasets during 1982-2010, Atmosphere, 5, 737-754, 2014.

Yin, G., Hu, Z.Y., Chen, X., and Tiyip, T.: Vegetation dynamics and its response to climate change in Central Asia, J. Arid Land, 8, 375, 2016.

Yu, J., Zhang, G., Yao, T., Xie, H., Zhang, H., Ke, C., and Yao, R.: Developing daily cloud-free snow composite products from MODIS Terra-Aqua and IMS for the Tibetan Plateau, IEEE Trans. Geosci. Remote Sens., 54(4), 2171-2180, 2015.

Yue, S., Pilon, P., Phinney, B., Cavadias, G.: The influence of autocorrelation on the ability to detect trend in hydrological series, Hydrol. Process., 16(9), 1807-1829, 2002.

[revised manuscript text omitted]

---

## Referee Comment (RC4) · Anonymous Referee #4 · 9 Feb 2017

**General Comments**

(This review is based on the revised manuscript uploaded by the authors on January 4[th] 2017.)

The authors present an analysis of the water balance of 18 catchments on the Tibetan Plateau using multiple datasets. The basis of their approach is to consider actual evapotranspiration (ET) as the main unknown in the water balance and estimate this using data on the other components. This work is interesting and potentially a useful contribution to our understanding of the hydrology of the Tibetan Plateau. However, the paper suffers from several major limitations in its current form, as described below.

1. Although uncertainty is briefly addressed in Section 3.4, I do not feel that this issue is considered in enough depth. As the authors acknowledge, there are large uncertainties associated with the datasets used in the study, as well as the method for estimating ET. I feel that these uncertainties should be incorporated into the analysis rather than addressed in general terms afterwards (Section 3.4). Otherwise the significance of the conclusions is unclear, i.e. whether the results are really an artefact of data limitations (see points below). This is my most significant concern and I feel that a fair amount of additional work and revision could be required to address this. (One variant of this approach could be to examine the errors from attempting to close the water balance with published data products, rather than forcing closure by considering ET as the residual term.)

2. It is not clear to me whether trend analysis is fully justified by the data. This is particularly the case for ET, as the authors use an approximate method to extend their ET estimates prior to the period of available GRACE data. As ET is also resolved as the residual in the water balance, any errors in the other components could be compensated for in ET estimates, which could presumably affect trend analysis. It may even be worth focusing on annual and seasonal water balance estimation in this paper and leaving the trend analysis for a more considered treatment in a separate manuscript.

3. Some of the main conclusions are stated in the abstract and summary as precipitation being the main contributor to runoff and snow water equivalent (SWE) being higher from late autumn to spring. These conclusions seem fairly basic and general. I think it may be possible to draw more substantial and specific conclusions from the analysis presented. This would stem from more focused discussion of results in Section 3.

4. The standard of English needs to be improved throughout the manuscript. While the meaning is usually (but not always) clear, there are a lot of grammatical errors (far too many to list). I suggest the authors enlist the help of someone with native-level proficiency to carefully revise the text.

**Specific Comments**

(Line numbers refer to the revised manuscript uploaded by the authors on January 4[th] 2017.)

Table 1 – It is not clear what data sources are behind all of the summary statistics for the catchments listed in this table – in particular for precipitation, temperature, NDVI, LAI and snow cover (imprecise reference). In addition, should the "Second Glacier Inventory of China" appear in Table 2 and/or the reference list? Perhaps it would make sense to introduce the catchment characteristics after the section on data rather than before.

Figure 1 – I wonder if the solid shading of catchments is really needed.

Section 2.1 – Would it be better to introduce the uncertainties associated with each dataset here? This should include observational uncertainty (e.g. precipitation undercatch (including snowfall) and discharge), as well as all model and remote sensing datasets.

Line 178-181 – What additional processing of the GRACE data products was done (regarding the glacier isostatic adjustment correction and destriping)?

Equation 2 – I am not sure that this equation is helpful. It may be better just to retain the statement in the preceding paragraph that glacier melt is a component of runoff. At the very least Q would need to

be defined differently from Equation 1 (i.e. introducing a term for non-glacial runoff), otherwise the two equations are not consistent (unless glacier melt equals zero). More generally in Section 2.2.1, I am not clear why the glacier melt contribution to catchment runoff needs to be estimated at all for the ET calculations (as observed runoff at the catchment outlets includes glacier melt).

Table 3 – As this is a very approximate estimation of glacier contribution to discharge, quoting percentages to two decimal places seems too precise.

Section 2.2.1 – I am struggling to fully understand all of the bias-correction procedure, particularly the rationale behind the second step. It would be useful if the authors could clarify this please. In addition, "m" and "a" are used in Equations 3 and 4 but I am not sure that they are defined anywhere. The uncertainties arising from extending the ET series back prior to the GRACE data period should also be considered.

Line 293-294 – The variable X should be defined.

Line 309-313 – It may be more useful to evaluate the VIC flow results in terms of biases and consistency of anomalies (monthly and annual) relative to observed discharge rather than Nash Sutcliff Efficiency (NSE), as the focus of the study is on water budgets. From Figure 2 it looks like peak flows are underestimated during "wetter" years. In addition, why does runoff simulated by VIC appear to drop to zero during the low flow season?

Line 313-317 – What data were used to force the VIC model (i.e. is there any circularity in this comparison of precipitation datasets)? The uncertainties in observation-derived and TRMM-estimated gridded precipitation products should also be acknowledged (i.e. consistency may be encouraging but neither represents "absolute truth").

Section 3.1 – In general, it would be useful to see how the different datasets look in a little more detail. For example, what magnitude of storage changes is present according to the GRACE data products? This would have big impacts on the ET calculations, so the authors need to demonstrate the credibility of GRACE data with reference to other studies (and with reference to our understanding of Tibetan Plateau hydrology to date).

Line 319-331 – Evaluating different ET products with reference to ET calculated as a residual of the water balance depends very heavily on the uncertainties/accuracy of the datasets underpinning the other terms of the water balance. The selection/rejection of ET products for further analysis in this section/paragraph is done without reference to uncertainties – so do we really know that these are the better products?

Figure 3 – This might need more clarity on what time scales are used in the analysis underpinning the figure.

Line 333-353 – Could more be made of the discussion of Figure 4? The differences in catchment properties and their relationships to climatic influences are interesting.

Figure 5 – It might be easier to interpret this figure if both the primary and secondary vertical axes used the same range (i.e. so precipitation can be compared with ET and runoff). Indeed displaying the data as a bar chart could be preferable (e.g. at its simplest, separate bars for precipitation, ET, runoff and implied storage change).

Figure 7 – Some of the VIC SWE estimates look unrealistic for some of the catchments and the range of scales on the secondary vertical axis complicates interpretation. I am not sure whether conclusions on SWE can really be drawn from this dataset.

Section 3.2 – I am not sure if full use is made of the figures and their underpinning analysis in this section generally. I think more focused discussion of the results of the water balance (annual and seasonal) should be possible.

Section 3.3 – Some of the discussion in this section seems speculative, particularly regarding the relationships of calculated trends with climate indicators. This is not a simple subject and I suggest

that this section should be worded more carefully and discuss the drivers of trends in less definitive terms.

Section 3.4 – As discussed above, I do not feel that this is a sufficient treatment of uncertainty (see general comments).

In addition, some of the references in the text appear to be misspelt (e.g. line 66 Immerzeel, line 99 Harris). I suggest that all references are carefully checked.

---

## Author Comment (AC6) · 15 Feb 2017

Review Comments (Anonymous Referee #4): General Comments (This review is based on the revised manuscript uploaded by the authors on January 4th 2017.)

The authors present an analysis of the water balance of 18 catchments on the Tibetan Plateau using multiple datasets. The basis of their approach is to consider actual evapotranspiration (ET) as the main unknown in the water balance and estimate this using data on the other components. This work is interesting and potentially a useful contribution to our understanding of the hydrology of the Tibetan Plateau. However, the paper suffers from several major limitations in its current form, as described below. Thank you very much for your invaluable comments/suggestions. We have revised

the manuscript accordingly (please see the point-to-point response below). In the Acknowledgement section [Line 587-588 in the new version]: "We thank the editors and reviewers for their invaluable comments and constructive suggestions".

1. Although uncertainty is briefly addressed in Section 3.4, I do not feel that this issue is considered in enough depth. As the authors acknowledge, there are large uncertainties associated with the datasets used in the study, as well as the method for estimating ET. I feel that these uncertainties should be incorporated into the analysis rather than addressed in general terms afterwards (Section 3.4). Otherwise the significance of the conclusions is unclear, i.e. whether the results are really an artefact of data limitations (see points below). This is my most significant concern and I feel that a fair amount of additional work and revision could be required to address this. (One variant of this approach could be to examine the errors from attempting to close the water balance with published data products, rather than forcing closure by considering ET as the residual term.)

The main objective of this study is to test whether the general hydrological regimes (seasonal cycles and trend of the water balance components) could be inferred from the perspective of multi-source datasets in the data-sparse Tibetan Plateau. We have also showed the potentials through the use of multi-source datasets (e.g., satellite retrievals, LSM simulations) to achieve this goal in this paper. The results obtained are generally in line with earlier studies (e.g., Liu et al., 1999; Dong et al., 2007; Fu et al., 2008; Zhu et al., 2011; Zhang et al., 2013; Cuo et al., 2014 ), thus the results are not artefact of data limitation.

This study may unavoidably associate with some uncertainty due to the use of multi-source datasets. We totally agree with the reviewer that the uncertainty should be quantified as well in the analysis. We have actually tried to consider the uncertainty in the analysis, for example, we compared the observed CMA precipitation with TRMM and IGSNRR_forcing data during 2000-2011. The water balance-based ETwb was also compared with other six global/regional ET products (including the mean and annual
trends) during the period 1982-2006. Moreover, to minimize the uncertainty in the analysis we only analyzed the well-performed ET products together with the observed runoff, precipitation and ETwb of basic water balance components during 1982-2011.

However, adequately quantification of uncertainty for each water budget component is difficult. When we focused on the analysis of one variable during the period 1982-2011, few datasets can be used together in the TP to quantify its uncertainty due to the data availability. For example, we have observed CMA precipitation from 1982-2011, but the TRMM precipitation is only available since 2000. We can also calculate ETwb for the period 1982-2011, but Zhang_E is only available from 1983-2006. Moreover, the global datasets for NDVI, LAI, SWE and water storage changes are also limited which, to some extent, restricted our attempts to quantify the uncertainties in the analysis using multi-source datasets.

In response, we have expanded the uncertainty section to discuss the uncertainty related issues as follows [Line 500-527 in the new version], "...In particular, it is well known that land surface models have some difficulties (e.g., parameter tuning in boundary layer schemes) when applying to the TP, even though they have good performances in different regimes (Xia et al., 2012; Bai et al., 2016). For example...There are also considerable uncertainties arising from empirical extending the ET series back prior to the GRACE era. Finally, we obtained the contributions of glacier-melt to discharge in some basins from the literatures and took them as constant numbers...With these caveats, we can interpret the general hydrological regimes and their responses to the changing climate in the TP basins from solely the perspective of multi-source datasets, which are comparable to the existing studies based on the in situ observations and complex hydrological modeling."

Reference: Liu, T.: Hydrological characteristics of Yalungzangbo River, Acta Geogr. Sin., 54 (Suppl.), 157-164, 1999 (in Chinese). Dong, X., Yao, Z., and Chen, C.: Runoff variation and responses to precipitation in the source regions of the Yellow River, Resour. Sci., 29(3), 67-73, 2007 (in Chinese). Fu, L., Chen, Y., Li, W., Xu, C., and He,

B.: Influence of climate change on runoff and water resources in the headwaters of the Tarim River, Arid Land Geogr., 31(2), 237-242, 2008 (in Chinese). Zhu, Y., Chen, J., Chen, G.: Runoff variation and its impacting factors in the headwaters of the Yangtze River in recent 32 years, J.Yangtze River Sci. Res. Inst., 28(6), 1-4, 2011 (in Chinese ). Zhang, L., Su, F., Yang, D., Hao, Z., and Tong, K.: Discharge regime and simulation for the upstream of major rivers over Tibetan Plateau, J. Geophys. Res. Atmos., 118(15), 8500-8518, 2013. Cuo, L., Zhang, Y.X., Zhu, F.X., and Liang, L.Q.: Characteristics and changes of streamflow on the Tibetan Plateau: A review, J. Hydrol. Reg. stud., 2, 49-68, 2014.

2. It is not clear to me whether trend analysis is fully justified by the data. This is particularly the case for ET, as the authors use an approximate method to extend their ET estimates prior to the period of available GRACE data. As ET is also resolved as the residual in the water balance, any errors in the other components could be compensated for in ET estimates, which could presumably affect trend analysis. It may even be worth focusing on annual and seasonal water balance estimation in this paper and leaving the trend analysis for a more considered treatment in a separate manuscript.

The closure of basin-scale water balance with published data products is actually difficult due to the lack of observation and the uncertainties inherited from multi-source data (remote sensing retrievals, reanalysis outputs and LSM simulations). Nowadays, there is no such a "best" approach to do this due to the data limitation, although some attempts have been published using the empirical and statistical methods. We also proposed a bias-correction based method which regards the basin-wide P minus Q as the biased ET and empirically correct the biased ET with the cumulative distribution of P minus Q and water storage changes (this method has been used in some papers such as Li et al., 2014 and Liu et al., 2016). The general hydrological regimes (e.g. the multiyear means and seasonal cyles of water balance components) concluded in this study were consistent with earlier studies (e.g., Liu et al., 1999; Dong et al., 2007; Fu

et al., 2008; Zhu et al., 2011; Zhang et al., 2013; Cuo et al., 2014 ). These all give us the confidences to believe the rationality of the further trend analysis.

Similar to the response of comment 1, we have expanded the uncertainty section to discuss the uncertainty related issues as follows [Line 500-527 in the new version], "...In particular, it is well known that land surface models have some difficulties (e.g., parameter tuning in boundary layer schemes) when applying to the TP, even though they have good performances in different regimes (Xia et al., 2012; Bai et al., 2016). For example...There are also considerable uncertainties arising from empirical extending the ET series back prior to the GRACE era. Finally, we obtained the contributions of glacier-melt to discharge in some basins from the literatures and took them as constant numbers...With these caveats, we can interpret the general hydrological regimes and their responses to the changing climate in the TP basins from solely the perspective of multi-source datasets, which are comparable to the existing studies based on the in situ observations and complex hydrological modeling."

Reference: Li, X.P., Wang, L., Chen, D.L., Yang, K., and Wang, A.H.: Seasonal evapotranspiration changes (1983-2006) of four large basins on the Tibetan Plateau, J. Geophys. Res., 119 (23), 13079-13095, 2014. Liu, W.B., Wang, L., Zhou, J., Li, Y.Z., Sun, F.B., Fu, G.B., Li, X.P., and Sang, Y-F.: A worldwide evaluation of basin-scale evapotranspiration estimates against the water balance method, J. Hydrol., 538, 82-95, 2016. Liu, T.: Hydrological characteristics of Yalungzangbo River, Acta Geogr. Sin., 54 (Suppl.), 157-164, 1999 (in Chinese). Dong, X., Yao, Z., and Chen, C.: Runoff variation and responses to precipitation in the source regions of the Yellow River, Resour. Sci., 29(3), 67-73, 2007 (in Chinese). Fu, L., Chen, Y., Li, W., Xu, C., and He, B.: Influence of climate change on runoff and water resources in the headwaters of the Tarim River, Arid Land Geogr., 31(2), 237-242, 2008 (in Chinese). Zhu, Y., Chen, J., Chen, G.: Runoff variation and its impacting factors in the headwaters of the Yangtze River in recent 32 years, J.Yangtze River Sci. Res. Inst., 28(6), 1-4, 2011 (in Chinese ). Zhang, L., Su, F., Yang, D., Hao, Z., and Tong, K.: Discharge regime and simulation for

the upstream of major rivers over Tibetan Plateau, J. Geophys. Res. Atmos., 118(15), 8500-8518, 2013. Cuo, L., Zhang, Y.X., Zhu, F.X., and Liang, L.Q.: Characteristics and changes of streamflow on the Tibetan Plateau: A review, J. Hydrol. Reg. stud., 2, 49-68, 2014.

3. Some of the main conclusions are stated in the abstract and summary as precipitation being the main contributor to runoff and snow water equivalent (SWE) being higher from late autumn to spring. These conclusions seem fairly basic and general. I think it may be possible to draw more substantial and specific conclusions from the analysis presented. This would stem from more focused discussion of results in Section 3. 4.

Similar to the responses of comment #1, the main objective of this study is to test whether the general hydrological regimes (seasonal cycles and trend of the water balance components) could be inferred from the perspective of multi-source datasets in the data-sparse Tibetan Plateau. The results obtained are also generally in line with those concluded from some basin-scale and observe-based studies. Certainly, as we discussed in the paper and responded in comment #1, this study may unavoidably associate with some uncertainty due to the use of multi-source datasets. We thus only draw some general results (e.g., seasonal cycles and trends of water balance components, some results have been obtained from previous basin-specific and observation-based studies but most have not been investigated) confidently considering the considerable uncertainties inherited in different datasets (not more substantial and specific conclusions). The reviewer also mentioned to stem more focused discussion from the results of Section 3.4, please also refer to the responses of comment #1. Thanks.

The standard of English needs to be improved throughout the manuscript. While the meaning is usually (but not always) clear, there are a lot of grammatical errors (far too many to list). I suggest the authors enlist the help of someone with native-level proficiency to carefully revise the text. We invited our colleague with sufficient English language training (Wee Ho Lim) to participate in this manuscript. We believe that the language structure and grammar is substantial improved in the revised version.

[Figure]

Specific Comments (Line numbers refer to the revised manuscript uploaded by the authors on January 4th 2017.) Table 1 – It is not clear what data sources are behind all of the summary statistics for the catchments listed in this table – in particular for precipitation, temperature, NDVI, LAI and snow cover (imprecise reference). In addition, should the "Second Glacier Inventory of China" appear in Table 2 and/or the reference list? Perhaps it would make sense to introduce the catchment characteristics after the section on data rather than before.

We agree. In the revised version, we have changed Table 1 to Table 2 (and correspondingly the former Table 2 has been changed to Table 1 in the new version). The introductions of catchment characteristics have been moved after the section on data as suggested by the reviewer. The "Second Glacier Inventory of China" and its reference have also added in the end of Table 1 (Table 2 in the old version). Further, some more descriptions have been added in the caption of Table 2 (Table 1 in the old version) to clearly indicate the data sources used for the summary statistics for the catchments listed in this table as follows [Line 824-827 in the new version],

"Table 2: Main features of the 18 TP river basins used in this study. The precipitation and temperature statistics for each basin were calculated from the observed CMA datasets while the NDVI and LAI statistics were extracted from the GIMMS NDVI dataset and GLASS LAI product. The GA% and SC% represented the percentages of multiyear-mean glacier cover and snow cover in each basin which were calculated from the Second Glacier Inventory Dataset of China and the daily TP snow cover dataset (2005-2013)"

Figure 1 – I wonder if the solid shading of catchments is really needed.

Good point. We substituted the solid shading of catchments (Figure 1) with catchment boundary (i.e., Figure R1 in this response).

Section 2.1 – Would it be better to introduce the uncertainties associated with each dataset here? This should include observational uncertainty (e.g. precipitation undercatch (including snowfall) and discharge), as well as all model and remote sensing datasets.

Thanks. We respect the reviewer's viewpoint. However, nowadays these global datasets (e.g., different ET products, SWE products, NDVI and LAI) has rarely been comprehensively assessed in the TP due to the lack of in situ observations. The advantages/disadvantages of different datasets are thus difficult to be fully described. Moreover, detailed validations of these products in the TP are beyond the scope of this study, which need be further investigated in the future works.

We have also tried to add more details associated with some datasets in the uncertainty Section through summarizing the limited studies available as follows [Line 498-527 in the new version]: "...due to their different forcing data; algorithm used as well as varied spatial-temporal resolutions (Xue et al., 2013; Li et al., 2014; Liu et al., 2016a). In particular, it is well known that land surface models have some difficulties (e.g., parameter tuning in boundary layer schemes) when applying to the TP, even though they have good performances in different regimes (Xia et al., 2012; Bai et al., 2016). For example, Xue et al. (2013) indicated that GNoah_E underestimated theãĂŰ ETãĂŮ_wb in the upper Yellow River and Yangtze River basins on the Tibetan Plateau mainly due to its negative-biased precipitation forcing. We thus only usedãĂŰ ETãĂŮ_wb in the trend detection of water budget components in Fig.8, Fig.10 and Fig.11 in this study. The two SWE products also showed large uncertainty with respect to both their seasonal cycles and trends. The VIC_IGSNRR simulated and GlobaSnow-2 SWEs have not been validated in the TP due to the lack of snow water equivalent observations, but in some basins (e.g., Zelingou and Numaitilangan) they showed similar seasonal cycles and annual trends.

Moreover, the interpolation of missing values of runoff with VIC_IGSNRR simulated runoff and the gridded precipitation data (which interpolated from limited gauged precipitation over the plateau) also introduced uncertainties. There are also considerable uncertainties arising from empirical extending the ET series back prior to the GRACE
era. Finally, we obtained the contributions of glacier-melt to discharge in some basins from the literatures and took them as constant numbers. It may inherit considerable uncertainty from varied studies using different approaches such as glacier mass-balance observation, isotope observation and hydrological modeling, and the contribution rates would also change under a warming climate. However, reliable quantification of the contribution of glacier-melt to discharge is technically challenging, especially for the data-sparse basins. With these caveats, we can interpret the general hydrological regimes and their responses to the changing climate in the TP basins from solely the perspective of multi-source datasets, which are comparable to the existing studies based on the in situ observations and complex hydrological modeling. "

Line 178-181 – What additional processing of the GRACE data products was done (regarding the glacier isostatic adjustment correction and destriping)?

The glacier isostatic adjustment correction and destriping have already been done by different data processing centers. We actually did not do any additional processing for the GRACE data in this study. We have deleted this description in this sentence as follows [Line 170-172 in the clean version], "To minimize the errors and uncertainty of extracted $\Delta S$, we averaged these GRACE retrievals (2002-2013) from different processing centers in this study."

Equation 2 – I am not sure that this equation is helpful. It may be better just to retain the statement in the preceding paragraph that glacier melt is a component of runoff. At the very least Q would need to be defined differently from Equation 1 (i.e. introducing a term for non-glacial runoff), otherwise the two equations are not consistent (unless glacier melt equals zero). More generally in Section 2.2.1, I am not clear why the glacier melt contribution to catchment runoff needs to be estimated at all for the ET calculations (as observed runoff at the catchment outlets includes glacier melt).

We respect the reviewer's viewpoint. From water balance perspective, Equation 1 and 2 are consistent to one and another. The difference is the composition of observed runoff

(Q). In the non-glacier basins, P=Q+ET+$\Delta$S (Equation 1). All precipitation is converted to Q, ET and water storage change. In the glacier-dominated basins, we consider that the observed Q includes additional the glacier melts which is not be reflected in $\Delta$S. With this conception, we adapted Equation 1 (P=Q+ET+$\Delta$S) into a form that is more feasible for basins. Thus we revised it into Equation 2 (P+M_G=Q+ET+$\Delta$S).

Table 3 – As this is a very approximate estimation of glacier contribution to discharge, quoting percentages to two decimal places seems too precise.

We agree. We have quoted percentages to only one decimal place in the revised version (Table R3). Thanks.

Section 2.2.1 – I am struggling to fully understand all of the bias-correction procedure, particularly the rationale behind the second step. It would be useful if the authors could clarify this please.

We have revised the descriptions of bias-correction procedure to make is more clearer, as follows [Line 268-301 in the new version],

"The terrestrial water storage ($\Delta$S) in Eq.(2) includes the surface, subsurface and ground water changes. It has been demonstrated cannot be neglected in water balance calculation over monthly and annual timescales due to snow cover change and anthropogenic interferences (e.g., reservoir operation, agricultural water withdrawal) (Liu W. et al., 2016a). For the period 2002-2011, we calculated basin-wide ET (ãĂŰETãĂŮ_wb) directly using the GRACE-derived $\Delta$S in Eq.(2). Since GRACE data is absent before 2002, we calculated the monthly ãĂŰETãĂŮ_wb using the following two-step bias-correction procedure (Li X. et al., 2014). We defined P+M_G-Q in Eq. (2) as biased ET (ãĂŰETãĂŮ_biased, available from 1982 to 2011) relative to the "true" ET (ãĂŰETãĂŮ_wb= P+M_G-Q-$\Delta$S, available during the period 2002-2011 when the GRACE data is available). Over the period 2002-2011, we first fitted ãĂŰETãĂŮ_biased and ãĂŰETãĂŮ_wb series separately using different gamma distributions, which has been evidenced as an proper method

for modeling the probability distribution of ET (Bouraoui et al., 1999). The monthly ãĂŰETãĂŮ_biased series (2002-2011) can then be bias-corrected through the inverse function (Fˆ(-1)) of the gamma cumulative distribution function (CDF,F) of ãĂŰETãĂŮ_wb by matching the cumulative probabilities between two CDFs as follow (Liu W. et al., 2016a), ãĂŰETãĂŮ_corrrected (m)=Fˆ(-1) (F(ãĂŰETãĂŮ_biased (m)|$\alpha$_biased,$\beta$_biased)|$\alpha$_wb,$\beta$_wb) (3) Here $\alpha$_biased, $\beta$_biased and $\alpha$_wb , $\beta$_wb are shape and scale parameters of gamma distributions for ãĂŰETãĂŮ_biased and ãĂŰETãĂŮ_wb. ãĂŰETãĂŮ_corrected (m) and ãĂŰETãĂŮ_biased (m) represent the monthly corrected and biased ET, respectively. The bias correction procedure can be flexibly applied to the period 1983-2011 by matching the CDF of ãĂŰETãĂŮ_biased (1983-2011) to that of ãĂŰETãĂŮ_corrected (2002-2011). The second step of bias correction is to eliminate the annual bias through the ratio of annual ãĂŰETãĂŮ_biased to annual ãĂŰETãĂŮ_corrected calculated in the first step using the following method, ãĂŰETãĂŮ_final (m)=(ãĂŰETãĂŮ_biased (a))/(ãĂŰETãĂŮ_corrected (a))×ãĂŰETãĂŮ_corrected (m) (4) where ãĂŰETãĂŮ_final (m) is the final monthly ET after bias correction. ãĂŰETãĂŮ_biased (a) and ãĂŰETãĂŮ_corrected (a) represent the annual biased and corrected ET while ãĂŰETãĂŮ_corrected (m) is the monthly corrected ET obtained from the first step. The procedure was then applied to correct the monthly ãĂŰETãĂŮ_biased series and calculated the monthly ãĂŰETãĂŮ_corrected during the period 1982-2001 for all TP basins. We take these results as sufficient representation of the "true" ET (ãĂŰETãĂŮ_wb) for evaluating multiple ET products and trend analysis. "

In addition, "m" and "a" are used in Equations 3 and 4 but I am not sure that they are defined anywhere.

We have explained "m" and "a" in the revised version as follows [Line 295-297 in the new version], "ãĂŰETãĂŮ_biased (a) and ãĂŰETãĂŮ_corrected (a) represent the annual biased and corrected ET while ãĂŰETãĂŮ_corrected (m) is the monthly corrected ET obtained from the first step."

The uncertainties arising from extending the ET series back prior to the GRACE data period should also be considered.

We respect the reviewer's viewpoint. However, the uncertainties are actually hard to quantify due to the lack of observed basin-scale ET. We have added more discussion in the Section 3.4 as follows [Line 515-517 in the new version], "...There are also considerable uncertainties arising from empirical extending the ET series back prior to the GRACE era..."

Line 293-294 – The variable X should be defined.

We have defined X in the revised version as follows [Line 307-308 in the new version], "...For example, X (X_1,X_2,...,X_n) is a time series data, it will be replaced by..."

Line 309-313 – It may be more useful to evaluate the VIC flow results in terms of biases and consistency of anomalies (monthly and annual) relative to observed discharge rather than Nash Sutcliff Efficiency (NSE), as the focus of the study is on water budgets.

In hydrological applications, NSE is commonly used for evaluation of simulated discharges against the observations and we intend to keep it. In the revised version, we supplemented Figure 2 (Figure R2 in this response) with relevant statistical information (e.g., bias. correlation coefficient). Thanks a lot.

From Figure 2 it looks like peak flows are underestimated during "wetter" years. In addition, why does runoff simulated by VIC appear to drop to zero during the low flow season?

The VIC simulated runoff was directly downloaded from the Land Surface Processes and Global Change Research Group in IGSNRR, CAS (Website http://hydro.igsnrr.ac.cn/public/vic_outputs.html). In some basins, the VIC-simulated runoff may drop to zero due to the uncertainties of meteorological forcing data. Similar results were also shown in Zhang et al. (2014).

Reference: Zhang, X., Tang, Q., Pan, M., and Tang, Y.: A long-term land surface hydrologic fluxes and states dataset for China, J. Hydrometeorol., 15, 2067-2084, 2014.

Line 313-317 – What data were used to force the VIC model (i.e. is there any circularity in this comparison of precipitation datasets)? The uncertainties in observation-derived and TRMM-estimated gridded precipitation products should also be acknowledged (i.e. consistency may be encouraging but neither represents "absolute truth").

The VIC model (Zhang et al., 2014) was mainly forced by static parameters (e.g., DEM-derived terrain parameters, satellite-derived NDVI and land cover map) and IGSNRR meteorological forcing data (0.5 degree, 3-hour), which was constructed solely from the CMA station-based meteorological data (756 stations). We have compared the basin-averaged IGSNRR_forcing precipitation (756 stations) and TRMM (2000-2011) against the CMA gridded precipitation (more than 2000 stations) for all 18 TP basins. However, it is difficult to draw all precipitation time series for 18 basins (only the results for the smallest basin as drawn in Figure 2c). We thus simply compared the annual means, correlation coefficient and RMSE in Figure 2d in the manuscript.

In response, we have acknowledged the uncertainties in observation-derived and TRMM-estimated gridded precipitation as follows [Line 332-333 in the new version], ". . ., although the observation-derived and TRMM-estimated precipitation also has uncertainties"

Reference: Zhang, X., Tang, Q., Pan, M., and Tang, Y.: A long-term land surface hydrologic fluxes and states dataset for China, J. Hydrometeorol., 15, 2067-2084, 2014.

Section 3.1 – In general, it would be useful to see how the different datasets look in a little more detail. For example, what magnitude of storage changes is present according to the GRACE data products? This would have big impacts on the ET calculations, so the authors need to demonstrate the credibility of GRACE data with reference to other studies (and with reference to our understanding of Tibetan Plateau hydrology to date).

Good idea. We have added a Table (Table 4 in the new version and Table R4 in this response) to describe the annual mean magnitude of GRACE-derived storage changes. Moreover, the uncertainties between the GRACE-derived water storage changes from different processing centers for 18 TP basins are also discussed as follows [Line 333-338 in the new version],

"...The magnitudes of GRACE-derived annual mean water storage change (△S) in 18 TP basins are relatively less than those for other water balance components such as annual P, Q and ET (Table 4). The uncertainties among GRACE-derived annual mean △S from different data processing centers (CSR, GFZ and JPL) are small for 18 basins except for in the basins controlled by Gadatan and Tangnaihai stations. .."

Line 319-331 – Evaluating different ET products with reference to ET calculated as a residual of the water balance depends very heavily on the uncertainties/accuracy of the datasets underpinning the other terms of the water balance. The selection/rejection of ET products for further analysis in this section/paragraph is done without reference to uncertainties–so do we really know that these are the better products?

The basin-scale ET is not measured. In hydrological applications, we often use the water balance-based ETwb (observed precipitation minus observed runoff and GRACE-observed water storage change) as the "true" reference and to assess different ET products against ETwb (Li et al., 2014; Liu et al., 2016) Thus, in this paper, the selection/rejection of ET products for further analysis is done with the reference of ETwb.

Reference: Li, X.P., Wang, L., Chen, D.L., Yang, K., and Wang, A.H.: Seasonal evapotranspiration changes (1983-2006) of four large basins on the Tibetan Plateau, J. Geophys. Res., 119 (23), 13079-13095, 2014. Liu, W.B., Wang, L., Zhou, J., Li, Y.Z., Sun, F.B., Fu, G.B., Li, X.P., and Sang, Y-F.: A worldwide evaluation of basin-scale evapotranspiration estimates against the water balance method, J. Hydrol., 538, 82-95, 2016.

Figure 3 – This might need more clarity on what time scales are used in the analysis

underpinning the figure.

Good point. We have clarified the time scales used in the caption of Figure 3 as follows [Line 912-914 in the new version],

"Figure 3. Comparison of different ET products against the calculated ET through the water balance (ETwb) at the monthly time scale for 18 river basins over the Tibetan Plateau during the period 1983-2006. The boxplot of monthly estimates of different ET products for 18 TP basins are shown in (a) while the correlation coefficients and root-mean -square-errors (RMSEs, mm/month) for each ET product relatively to ETwb are exhibited in (b). "

Line 333-353 – Could more be made of the discussion of Figure 4? The differences in catchment properties and their relationships to climatic influences are interesting.

Good idea. We have made more discussion in terms of Figure 4 in the revised version as follows [Line 371-376 in the new version],

"...The dominant climate systems are overall discrepant for the three TP regions with different water-energy characteristics and sources of water vapor. The westerlies-controlled basins are relatively colder than the Indian monsoon-dominated basins, thus they develop more glaciers (and thus have more snow melt contributions to total river streamflow) and have relatively less vegetation (and thus limit vegetation transpiration)..."

Figure 5 – It might be easier to interpret this figure if both the primary and secondary vertical axes used the same range (i.e. so precipitation can be compared with ET and runoff). Indeed displaying the data as a bar chart could be preferable (e.g. at its simplest, separate bars for precipitation, ET, runoff and implied storage change).

In the revised version, we have changed both the primary and secondary vertical axes in Figure 5 (Figure R5 in this response) using the same range as suggested by the reviewer. Thank you very much.

Figure 7 – Some of the VIC SWE estimates look unrealistic for some of the catchments and the range of scales on the secondary vertical axis complicates interpretation. I am not sure whether conclusions on SWE can really be drawn from this dataset.

The VIC_SWE may have certain uncertainties due to the limit considerations of snow physics in VIC simulation (Zhang et al., 2014). However, there are only few SWE datasets available for the Tibetan Plateau (e.g., VIC_SWE and GlobSnow-2 SWE, all have considerable uncertainties) which could be used to infer the seasonal cycles of SWE in TP conditions (as we discussed in the uncertainty section). The only available datasets may uncertain for some TP catchments, but sometimes they are useful for other catchments (especially in the data-sparse region). Moreover, the value ranges of VIC_SWE in some basins (e.g., Yangcun and Nuxia) are obviously larger than other basins based on the VIC model simulations, we thus cannot draw the secondary vertical axis in the same scale (the VIC_SWE in some basins with relative smaller SWE values will become nearly a straight line and the seasonal cycles will have to be differentiated), but we have tried to draw the first vertical axis using blue color to make them more interpretable in the revised version (Figure R7 in this response).

Reference: Zhang, X., Tang, Q., Pan, M., and Tang, Y.: A long-term land surface hydrologic fluxes and states dataset for China, J. Hydrometeorol., 15, 2067-2084, 2014.

Section 3.2 – I am not sure if full use is made of the figures and their underpinning analysis in this section generally. I think more focused discussion of the results of the water balance (annual and seasonal) should be possible.

Please refer to the responses of general comment #1 and #3. Thank you very much.

Section 3.3 – Some of the discussion in this section seems speculative, particularly regarding the relationships of calculated trends with climate indicators. This is not a simple subject and I suggest that this section should be worded more carefully and discuss the drivers of trends in less definitive terms.

We agree. In the revised manuscript, we replaced them with more appropriate terms [Line 432-488 in the new version],

Section 3.4 – As discussed above, I do not feel that this is a sufficient treatment of uncertainty (see general comments).

Please also refer to the responses of general comment #1.

In addition, some of the references in the text appear to be misspelt (e.g. line 66 Immerzeel, line 99 Harris). I suggest that all references are carefully checked.

We have double-checked all references and have revised the incorrect ones accordingly. Thank you very much.

Please also note the supplement to this comment:
http://www.hydrol-earth-syst-sci-discuss.net/hess-2016-624/hess-2016-624-AC6-supplement.pdf

---

## Author Comment (AC7) · 15 Feb 2017

**Seasonal cycles and trends of water budget components in 18 river basins across the Tibetan Plateau: a multiple datasets perspective**

Wenbin Liu[a], Fubao Sun[a,b*], Yanzhong Li[a], Guoqing Zhang[b,cc,d],

Yan-Fang Sang[a],

Wee Ho Lim[a,e], Jiahong Liu[d]Liu[f], Hong Wang[a] ,Peng Bai[a]

[a]Key Laboratory of Water Cycle and Related Land Surface Processes, Institute of Geographic Sciences and Natural Resources Research, Chinese Academy of Sciences, Beijing 100101, China

[b]Hexi University, Zhangye 734000, China

[b]Key [c]Key Laboratory of Tibetan Environmental Changes and Land Surface Processes, Institute of Tibetan Plateau Research, Chinese Academy of Sciences, Beijing 100101, China

[c]CAS [d]CAS Center for Excellent in Tibetan Plateau Earth Sciences, Beijing 100101, China

[e]Environmental Change Institute, Oxford University Centre for the Environment, School of Geography and the Environment, University of Oxford , Oxford OX1 3QY, UK

[d]Key [f]Key Laboratory of Simulation and Regulation of Water Cycle in River Basin, China Institute of Water Resources and Hydropower Research, Beijing 100038, China

**Submitted to**: Hydrology and Earth System Sciences

**Corresponding Author**: Dr. Fubao Sun (Sunfb@igsnrr.ac.cn), from the Key Laboratory of Water Cycle and Related Land Surface Processes, Institute of Geographic Sciences and Natural Resources Research, Chinese Academy of Sciences (No. A11, Datun Road, Chaoyang District, Beijing 100101, China)

**Email Addresses for other authors**: Wenbin Liu (liuwb@igsnrr.ac.cn), Yanzhong Li (liyz.14b@igsnrr.ac.cn), Guoqing Zhang (guoqing.zhang@itpcas.ac.cn), Yan-fang Sang (sangyf@igsnrr.ac.cn), Wee Ho Lim (limwh@igsnrr.ac.cn), Jiahong Liu (liujh@iwhr.com) , Hong Wang (wanghong@igsnrr.ac.cn) , Peng Bai (baip.11b@igsnrr.ac.cn)

   2017/2/15
* * *

无孤行控制，不对齐到网格，制
表位： 33.93 字符，左对齐

**Highlights**

- A water balance approach to quantify monthly evapotranspiration which accounts for the changes in glacier and water storage of Tibetan Plateau

- Evaluation of water budget components and trends for 18 river basin in Tibetan Plateau

- Discussion of uncertainties arise from multiple datasets used in Tibetan Plateau

**Abstract.** The dynamics of water budget over the Tibetan Plateau (TP) are not  well understood because of the lack of hydroclimatic observations. Based on multi-source datasets over a 30-year period (1982-2011), we investigate the seasonal cycles and trends of water budget components, e.g., precipitation (P), evapotranspiration (ET) and runoff (Q) of 18 river basins in TP. We apply a two-step bias-correction procedure to calculate the basin-scale ET considering the changes in glacier and water storage change. The results indicate that precipitation, which mainly concentrated during June-October (varied among different monsoons impacted basins), is the major contributor to the runoff in the TP basins. The basin-wide snow water equivalent (SWE) was relatively high from mid-autumn to spring for most of the river basins in TP. The water cycle intensified under  global warming in most of these basins; receded in the upper Yellow and Yalong sub-river basins due to the  weakening East Asian monsoon. Consistent with the  warming climate and moistening in the TP and western China, the aridity index (PET/P) in most of the river basins decreased. The results  demonstrate the usefulness of integrating the multi-source data (e.g., in situ observations, remote sensing products, reanalysis, land surface model simulations and climate model outputs) for hydrological applications in the data-sparse regions. More generally, such approach might offer helpful insights towards understanding the water/energy budgets and sustainability of water resource management practices of data-sparse regions in a changing environment.  and could be beneficial for understanding the water and energy budgets, sustainable management of water resources under a warming climate in the harsh and the data-sparse Tibetan Plateau.

**1  Introduction**

As the highest plateau in the globe (the average elevation is higher than 4000 meters above the sea level), the Tibetan Plateau (TP, also called "the roof of the world" or

"the third Pole") is regarded as one of the most vulnerable region under a warming climate and is subjected exposed to strong interactions among atmosphere, hydrosphere, biosphere and cryosphere in the earth system (Duan and Wu, 2006; Yao et al., 2012; Liu W. et al., 2016b). It also serves as the "Asian water tower" from which many some major Asian rivers such as Yellow riverRiver, Yangtze riverRiver,

Brahmaputra riverRiver, Mekong riverRiver, Indus riverRiver, etc., originate. It provides is a vital water resource to support the livelihood of hundreds of millions of people in China and the surrounding neighboring Asian countries (Immerzell

Immerzeel et al., 2010; Zhang et al., 2013). Knowledge about the water budgets and their responses to the changing environment is thus crucial for understanding the hydrological regimes and for sustainable water resources management as well as environmental protection in this special regionHence sound knowledge of water budget and hydrological regimes of TP and its response to changing environment would have practical relevance for achieving sustainable water resource management and environmental protection in this part of the world (Yang et al., 2014; Chen et al., 2015).

 Despite the importance of TP in this geographic region, advance in hydrological and land surfaces studies in this region has been limited by data scarcity (Zhang et al., 2007; Li F. et al., 2013; Liu X. et al., 2016). For instance, less than 80 observation stations (~10% of a total of ~750 observation station across China) have been established in TP by the  Chinese Meteorological Administration (CMA) since the mid-20$^{th}$ century (Wang and Zeng, 2012). These stations are generally sparse and unevenly distributed at relatively low elevation regions, focus only on the meteorological variables and lack of other land surface observations such as evapotranspiration, snow water equivalent and latent heat fluxes. In addition, long-term observations of river discharge, snow depth, lake depth and glacier melts in the TP are also absent (Akhta et al., 2009; Ma et al., 2016). Therefore, the water balance and hydrological regimes for each river basin of TP and their relation with monsoons are poorly understood (Cuo et al., 2014; Xu et al., 2016). Whilst this shortcoming could be resolved through

installation of in-situ monitoring systems (Yang et al., 2013; Zhou et al., 2013; Ma et al., 2015), the overall cost of running the operational sites would be substantial. Another workaround would be through modeling approach, i.e., feeding remote sensing information and meteorological forcing data into physically-based land surface model (LSM) to simulate the basin-wide water budget  (Bookhagen and Burbank, 2010; Xue et al., 2013; Zhang et al., 2013; Cuo et al., 2015; Zhou et al., 2015; Wang et al., 2016). However, such approach is not immune from the issue of data scarcity at multiple river basins (with varied sizes and/or terrain complexities) for supporting model calibration and validation purposes  (Li F. et al., 2014).

Most recently, several  global (or regional) datasets relevant to the calculation of water budget  have been released . They  include remote sensing-based retrievals (Tapley et al., 2004; Zhang et al., 2010; Long et al., 2014; Zhang Y. et al., 2016), land surface model (LSM) simulations (Rui, 2011), reanalysis outputs (Berrisford et al., 2011; Kobayashi et al., 2015) and gridded forcing data interpolated from the in situ observations (Harris et al., 2014). For example, there are many products  related to terrestrial evapotranspiration (ET) such as GLEAM_E (Global Land surface Evaporation: the Amsterdam Methodology, Miralles et al., 2011a), MTE_E (a product integrated the point-wise ET observation at

FLUXNET sites with geospatial information extracted from surface meteorological observations and remote sensing in a machine-leaning algorithm, Jung et al., 2010 ),

LSM-simulated ETs from Global Land Data Assimilation System version 2

(GLDAS-2) with different land surface schemes (Rodell et al., 2004), ETs from

Japanese 55-year reanalysis (JRA55_E), the ERA-Interim global atmospheric reanalysis dataset (ERAI_E) and the National Aeronautic and Space Administration (NASA) Modern Era Retrosphective-analysis for Research and Application (MERRA)

reanalysis data (Lucchesi, 2012). Moreover, there are also several global or regional

LSM-based runoff simulations from GLDAS and the Variable Infiltration Capacity (VIC) model (Zhang et al., 2014). A few attempts have been made to validate multiple datasets for certain water budget components and to explore their possible hydrological implications. , fFor example, Li X. et al. (2014) and Liu W. et al. (2016a)

evaluated multiple ET estimates against the water balance method at annual and monthly time scales. Bai et al. (2016) assessed streamflow simulations of GLDAS

LSMs in five major rivers over the TP based on the discharge observations. Although there are certain uncertainties might exist among different datasets with various spatial and temporal resolutions and calculated through using different algorithms (Xia et al., 2012), they offer an opportunity do provide a great chance for us to quantify examine the general basin-wide water budgets and their uncertainties in gauge-sparse regions such as the TP considered in this study.

From the multiple datasets perspective, this study aims to investigate the water budget in 18 TP river basins distributed across the Tibetan Plateau; and evaluate seasonal cycles and annual trends of these water budget components. The objectives of this study are (1) to investigate the general water budgets in 18 river basins across the

Tibetan Plateau from the perspective of multiple datasets, and (2) to evaluate the seasonal cycles and annual trends of water budget components for 18 TP basins. The paper is organized as follows: the datasets and methods applied in this study are described in Sect.2. The results of season cycles and annual trends of water budget components for the river basins are presented and discussed in Sect.3. The uncertainties arise from employing multiple datasets are also discussed in the same section. In  Sect.4, we generalize our findings  which would be helpful for understanding the water balances of the river basins under constant influence of interplay between westerlies and monsoons (e.g., Indian monsoon, East Asian monsoon) in the Tibetan Plateau.

**2  Data and methods**

**2.1 Multiple datasets used**
* * *
~~Eighteen river basins over the TP (Fig.1) with the drainage area ranging from 2832 to 191235 km² (Table 1) are chosen in this study due to the availability of runoff data during the period 1982-2011. They mainly locate at the northwestern, southeastern and eastern parts of the plateau with multiyear-mean and basin-averaged temperature and precipitation ranging from -5.68 to 0.97 °C and 128 to 717 mm, which are solely or combined controlled by the westerlies, the Indian Summer monsoon and the Easter Asian monsoon (Yao et al., 2012). The glacier and snow covers are relatively more for the westerlies-dominant basins such as Yerqiang, Yulongkashi and Keliya (10.86-23.27% and 29.16-35.95%, respectively) whereas are less for the East Asian~~

**2.1. 1 Runoff,  precipitation and  terrestrial storage**

**change**

We obtained the Observed daily runoff (Q) of the study period   from the National Hydrology Almanac of China (Table 1). There are < 30% missing data in some gauging stations such as Yajiang, Tongren, Gandatan and Zelingou. Therefore, the VIC Retrospective Land Surface Dataset over China (1952~2012, VIC_IGSNRR simulated) with a spatial resolution of 0.25 degree and a daily temporal resolution from the Geographic Sciences and Natural Resources Research (IGSNRR), Chinese Academy of Sciences, is also used. This dataset is derived from the VIC model forced by the gridded daily observed forcing (IGSNRR_forcing) (Zhang et al., 2014). A degree-day scheme was used in the model to  account for the influences of snow and glacier on hydrological processes.

In terms of precipitation (P), we used the gridded monthly precipitation dataset available at CMA (spatial resolution of 0.5 degree; 1961-2011; interpolated drom observations of 2372 national meteorological stations using the Thin Plate Spline method)    (Table 1). Since the reliability of this dataset might be restricted by the relatively sparse stations and complex terrain conditions of TP, we make an attempt to incorporate two other precipitation datasets ((IGSNRR_forcing and Tropical Rainfall Measuring Mission TRMM 3B43 V7).  The precipitation from IGSNRR forcing datasets (0.25 degree) was derived by interpolating gauged daily precipitation from 756 CMA stations based on the synergraphic mapping system algorithm (Shepard, 1984; Zhang et al., 2014) and was further bias-corrected using the CMA gridded precipitation.

<Table 1, here please, thanks>

To get the change in terrestrial storage (ΔS), we used of three latest global terrestrial water storage anomaly and water storage change  datasets (available on the GRACE Tellus website: http://grace.jpl.nasa.gov/) that were retrieved from the Gravity Recovery and Climate Experiment (GRACE, Tapley et al., 2004; Landerer and Swenson, 2012; Long et al., 2014). Briefly, they  were processed separately at the Jet Propulsion Laboratory (JPL), the GeoForschungsZentrum (GFZ) and the Center for Space Research at the University of Texas (CSR). To minimize the errors and uncertainty of extracted ΔS. we averaged these GRACE retrievals (2002-2013)  from different processing centers in this study.

**2.1. 2 Temperature, potential evaporation and ET**

We obtained the monthly gridded temperature dataset (0.5 degree) from CMA; and potential evaporation (PET) dataset (0.5 degree, Harris et al., 2013) from

Climatic Research Unit (CRU),  University of East Anglia.  Moreover, we used six  global /regional ET products (four diagnostic products and two LSMs simulations, Table 2), namely (1) GLEAM_E (Miralles et al., 2010, 2011), which  consist of three sources of ET (transpiration, soil evaporation and interception) for bare soil, short vegetation and vegetation with a tall canopy  calculated using a set of algorithm (www.gleam.eu), (2) GNoah_E simulated  using GLDAS-2 with the Catchment Noah scheme (http://disc.sci.gsfc.nasa.gove/hydrology/data-holdings) (Rodell et al., 2004), (3) Zhang_E (Zhang et al., 2010), which is estimated using the modified Penman-Monteith  equation forced with MODIS data, satellite-based vegetation parameters and meteorological observations (http://www.ntsg.umt.edu/project/et), (4) MET_E (Jung et al., 2010) (https://www.bgc-jena.mpg.de/geodb/projects/Home.phs), (5) VIC_E (Zhang et al., 2014) from VIC_IGSNRR simulations (http://hydro.igsnrr.ac.cn/public/vic_outputs.html) and (6) PML_E (Zhang Y. et al., 2016) computed from global observation-driven Penman-Monteith-Leuning (PML) model (https://data.csiro.au/dap/landingpage?pid=csiro:17375&v=2&d=true).

**2.1. 3 Vegetation and snow/glacier parameters**

To quantify the dynamics of vegetation of each river basin, we applied the Normalized Difference Vegetation Index (NDVI) and the Leaf Area Index (LAI)  (Table 1). Briefly, the NDVI data was obtained from the Global Inventory Modeling and Mapping Studies (GIMMS) (Turker et al., 2005)

(https://nex.nasa.gov/nex/projects/1349/wiki/general_data_description_and_access/)

while the LAI data was collected from the Global Land Surface Satellite (GLASS)

products (http://www.glcf.umd.edu/data/lai/) (Liang and Xiao, 2012). Whist the change in seasonal snow cover and glacier has significant impact on the water and energy budgets of TP, it remains a technical challenge to get reliable observations due to harsh environment (especially at the basin scale). However, recently available satellite-based/LSM-simulated products might provide adequate characterization of the variation of snow cover and glacier.

To quantify the change in snow cover at each basin, we applied the  daily cloud free snow composite product from MODIS Terra-Aqua and the Interactive

Multisensor Snow and Ice Mapping System for the Tibetan Plateau (Zhang et al., 2012; Yu et al., 2015), in conjunction with the snow water equivalent (SWE) retrieved from Global Snow

Monitoring for Climate Research product (GlobSnow-2, http://www.globsnow.info/)

and the VIC_IGSNRR simulations  (Takala et al., 2011;

Zhang et al., 2014). We extracted general distribution of glacier of TP from the Second Glacier Inventory Dataset of China (Guo et al., 2014). All gridded datasets used were first uniformly interpolated to a spatial resolution of 0.5 degree based on the bilinear interpolation to make their inter-comparison possible. The datasets were then extracted for each of TP basins.

**2.1. 4 Monsoon indices**

In general, the TP climate is  under the influence of the westerlies, Indian summer monsoon and East Asian summer monsoon (Yao et al.,

2012). To investigate the changes of monsoon systems and their potential impact on the water budget in the TP basins, we used three monsoon indices, namely

Asian Zonal Circulation Index (AZCI), Indian Ocean Dipole Mode Index (IODMI)

and East Asian Summer Monsoon Index (EASMI). Briefly, t~~

T~~he IODMI is an indicator of the east-west temperature gradient across the tropical

Indian Ocean  (Saji et al. 1999), which can be downloaded from the following website:

http://www.jamstec.go.jp/frcgc/research/d1/iod/HTML/Dipole%20Mode%20Index.ht ml. The EASMI and AZCI ($60^{o}$-$150^{o}$E) reflect the dynamics of East Asian summer monsoon (Li and Zeng, 2002) and the westerlies (represented by Asian Zonal

Circulation index), which can be obtained from Beijing Normal University (

http://ljp.gcess.cn/dct/page/65577) and the National Climate Center of China (http://ncc.cma.gov.cn/Website/index.php?ChannelID=43WCHID=5), respectively.

**2.1.5 Study basins**

In this study, we selected 18 river basins of varied sizes (range: 2832-191235 $km^{2}$; see Table 1 for details) with adequate runoff data over a 30-year period (1982-2011).

They are distributed in  the northwestern, southeastern and eastern parts of the plateau with multiyear-mean and basin-averaged temperature and precipitation ranging from -5.68 to 0.97 $^{o}$C and 128 to 717 mm, which are solely dominated or under the influences of  the westerlies, the Indian Summer monsoon and the East Asian monsoon (Yao et al., 2012). There are more  glacier and snow covers  in the westerlies-dominant basins such as Yerqiang, Yulongkashi and Keliya (10.86~23.27% and 29.16~35.95%, respectively); less for  the East Asian monsoon-dominated basins such as Yellow, Yangtze and Bayin (0~0.96% and 9.42~20.05%, respectively) (Table 2).

<Figure 1, here please, thanks>

<Table 2, here please, thanks>

**2.2 Methods**

**2.2.1 Water balance-based ET estimation**

The basin-wide water balance at the monthly and annual timescales could be written as the principle of mass conservation (also known as the continuity equation, Oliveira et al., 2014) of basin-wide precipitation (P, mm), evapotranspiration ($ET_{wb}$, mm), runoff (Q, mm) as well as terrestrial water storage change ($\Delta S$, mm),

$$ET_{wb} = P - Q - \Delta S \qquad (1)$$

In most TP basins, glacier melt ($M_G$, mm) contributes to river discharge together with precipitation (liquid precipitation and snow). The monthly and annual water balance in these basins can thus be revised as,

$$ET_{wb} = P + M_G - Q - \Delta S \qquad (2)$$

Several attempts have been made for separating glacier contributions to river discharge through site-scale isotopic observations, remote sensing as well as land-surface hydrological modeling for some individual TP basins (Zhang et al., 2013;

Zhou et al., 2014; Neckel et al., 2014; Xiang et al., 2016). However, accurate quantification of $M_G$ is difficult in the data-sparse TP, especially for multiple basins.

In this study, we simply use the percentages of glacier melt to river discharge for some TP basins derived from the literatures (Chen, 1988; Mansur and Ajnis, 2005;

Zhang et al., 2013; Liu J. et al., 2016) and the empirical relations between the glacier area ratio (%) and glacier melt in basins mentioned above (Table 3).

<Table 3, here please, thanks>

The terrestrial water storage (ΔS) in Eq. (2) includes the surface, subsurface and ground water changes. It has been demonstrated  cannot be neglected in water balance calculation over monthly and annual timescales due to snow

 cover change and  anthropogenic interferences  (e.g., reservoir operation, agricultural water withdrawal)

 (Liu  et al., 2016a). For the period 2002-2011, we calculated basin-wide  ET ($ET_{wb}$)

directly using the GRACE-derived

ΔS in Eq. (2). Since GRACE data is absent before 2002, we calculated the monthly $ET_{wb}$ using the following two-step bias-correction procedure

 (Li X. et al., 2014)

. We defined $P + M_G − Q$ in Eq. (2) as biased ET

($ET_{biased}$, available from 1982- to 2011) relative to the "true" ET (−$ET_{wb}$  $P +$

$M_G − Q − ΔS$, available  during the period 2002-2011 when the GRACE data is available) . Over the period 2002-2011, we first fitted

 $ET_{biased}$ and $ET_{wb}$ series separately

 using  different gamma distribution, which has been evidenced as an proper method for modeling the probability distribution of ET (Bouraoui et al., 1999). The  monthly $ET_{biased}$ series (2002-2011) can then be bias-corrected through the inverse function $(F^{-1})$ of the gamma cumulative distribution function (CDF, F) of $ET_{wb}$ by matching the cumulative probabilities between two CDFs as follow (Liu  et al., 2016a),

$$ET_{corrrected}(m) = F^{-1}(F(ET_{biased}(m)|\alpha_{biased}, \beta_{biased})|\alpha_{wb}, \beta_{wb}) \qquad (3)$$

Here $\alpha_{biased}$, $\beta_{biased}$ and $\alpha_{wb}$, $\beta_{wb}$ are shape and scale parameters of gamma distributions for $ET_{biased}$ and $ET_{wb}$. $ET_{corrected}(m)$ and $ET_{biased}(m)$ represent the monthly corrected and biased ET, respectively. The bias correction procedure can be flexibly applied to the period 1983-2011 by matching the CDF of $ET_{biased}$ (1983-2011) to that of $ET_{corrected}$ (2002-2011). The second step of bias correction is to eliminate the annual bias through the ratio of annual $ET_{biased}$ to annual $ET_{corrected}$ calculated in the first step using the following method,

$$ET_{final}(m) = \frac{ET_{biased}(a)}{ET_{corrected}(a)} \times ET_{corrected}(m) \qquad (4)$$

where $ET_{final}(m)$ is the final monthly ET after bias correction. $ET_{biased}(a)$ and $ET_{corrected}(a)$ represent the annual biased and corrected ET while $ET_{corrected}(m)$ is the monthly corrected ET obtained from the first step. The procedure was then applied to correct the monthly $ET_{biased}$ series and calculated the monthly $ET_{corrected}$ during the period 1982-2001 for all TP basins. We take these results as sufficient representation of the "true" ET ($ET_{wb}$) for evaluating multiple ET products and trend analysis."

$$(m) = (F((m)|,)|,) \qquad (3)$$

**2.2.2 Modified Mann-Kendall test method**

The Mann-Kendall (MK) test is a rank-based nonparametric approach which is less sensitive to outlier relative to other parametric statistics, but it is sometimes influenced by the serial correlation of time series. Pre-whitening is often used to eliminate the influence of lag-1 autocorrelation before the use of MK test. for example, $X (X_1, X_2, ..., X_n)$ is a time series data, it

 will be replaced by $(X_2 - cX_1, X_3 - cX_2, ..., X_{n+1} - cX_n)$ in pre-whitening if the lag-1 autocorrelation coefficient (c) is larger than 0.1 (von Storch,

1995). However, significant lag-i autocorrelation may still be detected after pre-whitening because only the lag-1 autocorrelation is considered in pre-whitening (Zhang et al., 2013). Moreover, it sometimes underestimate the trend for a given time series (Yue et al., 2002). Hamed and Rao (1998) proposed a modified version of MK

test (MMK) to consider the lag-i autocorrelation and related robustness of the autocorrelation through the use of equivalent sample size, which has been widely used in previous studies during the last five decades (McVicar et al., 2012; Zhang et al.,

2013; Liu and Sun, 2016). In the MMK approach, if the lag-i autocorrelation coefficients are significantly distinct from zero, the original variance of MK statistics will be replaced by the modified one. In this study, we used the MMK approach to quantify the trends of water budget components in18 TP basins and the significance of trend was tested at the >95% confidence level.

**3   Results and Discussion**

**3.1 ET evaluation and General hydrological characteristics of 18 TP basins**

In this study, wWe first assessed the VIC_IGSNRR simulated runoff against the observations for each basin (for example, at Tangnaihai and Pangduo stations in

Fig.2). If the Nash Efficiency coefficient (NSE) between the observation and simulation is above 0.65, tThe VIC_IGSNRR simulated runoff is acceptable and could be used to replace the missing values for a given basin, if the Nash Efficiency coefficient (NSE) between the observation and simulation is above 0.65. Moreover, the CMA precipitation is consistent with TRMM (Corr = 0.86, RMSE = 8.34

mm/month) and IGSNRR forcing (Corr = 0.94, RMSE = 7.15mm/month)

precipitation for multiple basins (and also for the smallest basin above Tongren station,

Fig.2), , although the observation-derived and TRMM-estimated precipitation also has uncertainties which reveals the applicably of CMA precipitation under the TP

conditions. The magnitudes of GRACE-derived annual mean water storage change ($\Delta S$) in 18 TP basins are relatively less than those for other water balance components such as annual P, Q and ET (Table 4). The uncertainties among GRACE-derived annual mean $\Delta S$ from different data processing centers (CSR, GFZ and JPL) are small for 18 basins except for in the basins controlled by Gadatan and Tangnaihai stations.

< Figure 2, here please, thanks>

< Table 4, here please, thanks>

We then evaluated six ET products in 18 TP basins against our calculated $ET_{wb}$ at a monthly basis during the period 1983-2006 (Fig. 3). The ranges of monthly averaged ET among different basins (approximately 4−39 mm month$^{-1}$) are very close for all products compare  to that calculated from the $ET_{wb}$(6−42 mm month$^{-1}$). However, GLEAM_E (correlation coefficient: Corr = 0.85 and root-mean-square-error: RMSE = 5.69 mm month$^{-1}$) and VIC_E (Corr = 0.82 and RMSE = 6.16 mm month$^{-1}$) perform relatively better than others. Although Zhang_E and GNoah_E were found closely correlated to monthly $ET_{wb}$ in the upper Yellow River, the upper Yangtze River, Qiangtang and Qaidam basins (Li X. et al., 2014), they did not exhibit overall good performances (Corr = 0.61, RMSE = 7.97 mm month$^{-1}$ for Zhang_E and Corr = 0.42, RMSE = 10.16 mm month$^{-1}$ for GNoah_E) for 18 TP basin used in this study. We thus use GLEAM_E and VIC_E together with $ET_{wb}$ to calculate the seasonal cycles and trends of ET in 18 TP basins in the following sections.

< Figure 3, here please, thanks>

To investigate the general hydroclimatic characteristics of rivers over the TP, we classify 18 basins into three categories, namely westerlies-dominated basins (Yerqiang, Yulongkashi and Kelia), Indian monsoon-dominated basins (Brahmaputra and Salween), and East Asian monsoon-dominated basins (Yellow, Yalong and Yangtze) referred to Tian et al. (2007), Yao et al. (2012) and Dong et al. (2016). Interestingly, they are clustered into three groups under Budyko framework (Budyko, 1974; Zhang D. et al., 2016) with relatively lower evaporative index  in Indian monsoon-dominant basins and higher aridity index  in westerlies-dominant basins, which reveal various long-term hydroclimatologic conditions (Fig. 4). Overall, the annual mean air temperature increases (-5.68 ~0.97 $^{o}$C) while multiyear mean glacier area (and thus the glacier melt normalized by precipitation) decreases (23.27 ~ 0%)

gradually from the westerlies-dominant, Indian monsoon-dominant to East Asian monsoon-dominant basins. The vegetation status (NDVI range: 0.05~0.43; LAI range:

0.03~0.83) tends to be better and ET increases (and thus runoff coefficient gradually decreases) from cold to warm basins (Fig. 4 and Table 1). The $R^2$ between basin-averaged NDVI and ET is 0.76 which shows a clear vegetation control on ET in

TP basins. The results is are in line with Shen et al. (2015), which indicated that the spatial pattern of ET trend was significantly and positively correlated with NDVI

trend over the TP. The dominant climate systems are overall discrepant for the three

TP regions with different water-energy characteristics and sources of water vapor. The westerlies-controlled basins are relatively colder than the Indian monsoon-dominated basins, thus they develop more glaciers (and thus have more snow melt contributions to total river streamflow) and have relatively less vegetation (and thus limit vegetation transpiration). It is a general picture of hydrological regime in high-altitude and cold regions (Zhang et al., 2013; Cuo et al., 2014), which could be interpreted from the perspective of multi-source datasets in the data-sparse TP.

< Figure 4, here please, thanks>

**3.2 Seasonal cycles of basin-wide water budget components for the TP basins**

The multi-year means of water budget components (i.e., P, Q, ET, snow cover and

SWE) and vegetation parameters (i.e., NDVI and LAI) were are calculated for each calendar month and for 18 TP river basins using multi-source datasets available from to 2011. Overall, the seasonal variations of P, Q, ET, air temperature and vegetation parameters are similar in all TP basins with peak values occurred in May to

September (Fig.5 and Fig.6). The seasonal cycles of snow cover and SWE are generally time consistent among the as well for 18 TP basins (the peak values mainly occur from October to next April, Fig.7). With the ascending air temperature from cold to warm months, the basin-wide precipitation increases and vegetation cover expandsturns green gradually (the basin-wide ET also increase). Meanwhile, snow cover and glaciers retreat and snow melt or vanish gradually with the melt water supplying the river discharge together with precipitation. The inter-basin variations of hydrological regime are to a large extent linked to the climate systems that prevail over the TP.

< Figure 5, here please, thanks>

Although the temporal patterns of hydrological components are generally analogous, they varied vary among the parameters, climate zones and even basins (Zhou et al.,

2005). For example, relative to air temperature, the seasonal pattern of variation of runoff is more similar to precipitation which reveals that runoff is mainly controlled by precipitation in most TP basins. It is in agreement with that summarized by Cuo et al. (2014). In the westerlies-dominated basins, the peak values of precipitation and runoff mainly concentrate in June-August, which contribute approximately 68-82%

and 67-78% of annual totals, respectively. During this period, the runoff always exceeds precipitation which indicates large contributions of glacier/snow-melt water to streamflow. It is consistent with the existing findings in Tarim River (Yerqiang,

Yulongkashi and Keliya rivers are the major tributaries of Tarim River), which indicated that the melt water accounted for about half of the annual total streamflow (Fu et al., 2008). The ET (vegetation cover) in three westerlies-dominated basins are relatively less (scarcer) than that in other TP basins while the percentages of glacier and seasonal snow cover are higher in these basins which contribute more melt water to river discharge (Fig.6 and Fig.7). Overall, the SWE in Yerqiang, Yulongkashi and

Keliya rivers are relatively higher in winter than other seasons, but they vary with basins and products which  reflect considerable uncertainties in SWE

estimations.

< Figure 6, here please, thanks>

In the Indian monsoon and East Asian monsoon-dominated basins, the runoff concentrates during June-September (or June- October) with precipitation being the dominant contributor of annual total runoff. For example, the peak values of precipitation and runoff occur during June-September at Zhimenda station (contributing about 80% and 74% of the annual totals) while those occur during

June-October at Tangnaihai station (contributing about 78% and 71% of the annual totals, respectively). The results are quite similar to the related studies in eastern and southern TP such as Liu (1999), Dong et al. (2007), Zhu et al. (2011), Zhang et al.

(2013), Cuo et al. (2014). The vegetation cover (ET) in most basins is denser (higher) than that in the westerlies-dominant basins. Moreover, the seasonal snow mainly covers from mid-autumn to spring and correspondingly the SWE is relatively higher in these months in all basins except for Yellow River above Xining station, Salwee River above Jiayuqiao station and Brahmaputra River above Nuxia and Yangcun stations.

< Figure 7, here please, thanks>

**3.3 Trends of basin-wide water budget components for the TP basins**

The hydrological cycle has intensified in the westerlies-dominated basins with Q, P

and $ET_{wb}$ all ascended  under regional warming (Fig.8), especially in the Keliya

River basin (Numaitilangan station). The aridity index (PET/P), which is an indicator for the degree of dryness, slightly declined in all basins in northwestern TP. Although both P and PET were found both increase since the 1980s (Shi et al., 2003; Yao et al., 2014), the declined PET/P is, to some extent, attributed to the ascending P exceed the increase in PET for in these basins (except for the Yulongkashi basin). –The climate moistening (Shi et al., 2003) in the headwaters of these inland rivers would be beneficial to the water resources and oasis agro-ecosystems in the middle and lower basins. The increase in streamflow was also found in most tributaries of the Tarim River (Sun et al., 2006; Fu et al., 2010; Mamat et al., 2010).

Moreover, the westerlies, revealed by the Asian Zonal Circulation Index ($60^{o}$-$150^{o}$ E), slightly enhanced (linear trend: 0.21) over the period of 1982-2011 (Fig.9). With the strengthening westerlies, moreMore water vapor was may be transported and fell as precipitation or snow in northwestern TP (e.g., the eastern Pamir region) with the strengthening westerlies. Both SWE products (VIC_IGSNRR simulated and GlobaSnow-2 product) showed slightly  increase for across all basins with the incrementalrising seasonal snow covers and advanced glaciers (Yao et al., 2012). More precipitation was transformed into snow or glacier and the runoff coefficient (Q/P) exhibited decrease although precipitation obviously increased (Fig.8). In addition, the transpiration in these basins may might decrease with vegetation degradation as revealed by the NDVI and LAI (Yin et al., 2016) but the atmospheric evaporative demand indicated by CRU PET increased (significantly increase in the Yulongkashi and Keliya rivers) during the period 1982-2011.

< Figure 8, here please, thanks>

< Figure 9, here please, thanks>

In the East Asian monsoon-dominated basins, there are two types of change for basin-wide water budget components. For example, P and Q slightly decreased in the upper Yellow River (Tangnihai, Huangheyan and Jimai stations) and Yalong River (Yajiang station) but increased in other basins (Zelingou, Gandatan, Xining, Tongren and Zhimenda stations) over the period of 1982-2011 (Fig.10). The decline in Q and P

for the upper Yellow and Yalong Rivers (locate at the eastern Tibetan Plateau) were consistent with that found by Cuo et al. (2013, 2014) as well as Yang et al. (2014), and were in line with the weakening (linear slope: -0.01) of the East Asian Summer

Monsoon (Fig.9). The vegetation turned green while $ET_{wb}$ and PET increased in all nine basins with the significantly ascending air temperature during the period

1982-2011. The aridity index (PET/P) was found decreased in all basins except for the upper Yellow River basin above Jimai station and the upper Yalong River basin above

Yajiang station. Moreover, both the runoff coefficients and SWE were decreased except for the Bayin River above Zelingou station and the upper Yellow River above

Tongren station in the East Asian monsoon dominated basins.

< Figure 10, here please, thanks>

The hydrological cycles were also found intensified in the Indian monsoon-dominated basins such as Salween River and Brahmaputra River (Fig.11), which were in line with the strengthening (linear trend: 0.01) of the Indian Summersummer monsoon (revealed by the Indian Ocean Dipole Mode Index) during the specific period

1982-2011 (Fig.9). In the six basins, trends in P, Q and $ET_{wb}$ were are all upward.

For example, at Jiayuqiao station, the annual streamflow showed slightly increasing trend which was consistent with that examined during 1980-2000 by Yao et al. (2012).

The vegetation status, revealed by NDVI and LAI, turned better significantly with the ascending air temperature. The aridity index (PET/P) decreased in all basins except for the Brahmaputra River above Tangjia station, which indicated that most basins in the Indian monsoon-dominated regions turn wet over the period of 1982-2011. The runoff coefficient (Q/P) increased at Gongbujiangda and Nuxia while decreased at

Jiayuqiao, Pangduo, Tangji and Yangcun stations. Moreover, the basin-wide SWE

declined in the upper Salween River and Brahmaputra River above Pangduo, Tangjia and Gongbujiangda stations while increased in Brahmaputra River above Nuxia and

Yangcun stations.

< Figure 11, here please, thanks>

**3.4 Uncertainties**

The results may unavoidably associate with several aspects of uncertainties which mainly inherited from the multi-source datasets used. For example, although the seasonal cycles of $ET_{wb}$ can be captured by GLEAM_E and VIC_E, they still have considerable uncertainties such as at some stations (e.g., Numaitilangan,

Gongbujiangda and Nuxia stations) (Fig.5). With respect toCompared to the the annual trend of $ET_{wb}$ (Table 4), most ET products (including the well-performed

GLEAM_E and VIC_E in some basins) cannot detect the decreasing trends in 7 out of

18 basins (at Kulukelangan, Tongguziluoke, Xining, Tongren, Jimai, Nuxia and

Gongbujiangda stations) due to their different forcing data; algorithm used as well as varied spatial-temporal resolutions (Xue et al., 2013; Li et al., 2014; Liu W et al.,

2016a). In particular, it is well known that land surface models have some difficulties (e.g., parameter tuning in boundary layer schemes) when applying to the TP, even though they have good performances in different regimes (Xia et al., 2012; Bai et al.,

2016). For example, Xue et al. (2013) indicated that GNoah_E underestimated the $ET_{wb}$ in the upper Yellow River and Yangtze River basins on the Tibetan Plateau mainly due to its negative-biased precipitation forcing. We thus only used $ET_{wb}$ in the trend detection of water budget components in Fig.8, Fig.10 and Fig.11 in this study.

The two SWE products also showed large uncertainty with respect to both their seasonal cycles and trends. The VIC_IGSNRR simulated and GlobaSnow-2 SWEs have not been validated in the TP due to the lack of snow water equivalent observations, but in some basins (e.g., Zelingou and Numaitilangan) they showed similar seasonal cycles and annual trends.

Moreover, the interpolation of missing values of runoff with VIC_IGSNRR simulated runoff and the gridded precipitation data (which interpolated from limited gauged precipitation over the plateau) also involved introduced some uncertainties as well as.

There are also considerable uncertainties arising from empirical extending the ET

series back prior to the GRACE era. Finally, we obtained the contributions of glacier-melt to discharge in some basins from the literatures and took them as fixed constant numbers. It may inherit considerable uncertainty from varied studies using different approaches such as glacier mass-balance observation, isotope observation and hydrological modeling, and the contribution rates would also change under a warming climate. However, accurate reliable quantification of the contribution of glacier-melt to discharge is technically difficult nowadayschallenging, especially for the data-sparse basins. With these caveats, we can interpret the general hydrological regimes and their responses to the changing climate in the TP basins from solely the perspective of multi-source datasets, which are comparable to the existing studies based on the in situ observations and complex hydrological modeling.

<Table 45, here please, thanks>

**4 Summary**

In this study, we investigated the seasonal cycles and trends of water budget
components in 18 TP basins during the period 1982-2011, which is not well
understood so far due to the lack of adequate observations in the harsh environment,
through integrating the multi-source global/regional datasets such as gauge data,
satellite remote sensing and land surface model simulations. By using a two-step bias
correction procedure, we calculated the annual basin-wide $ET_{wb}$ was calculated
through the water balance approach considering the impacts of glacier and water
storage change. The We found that the GLEAM_E and VIC_E were found perform
better relative to other products against the calculated $ET_{wb}$.

From the Budyko framework perspective, tThe general water and energy budgets
were are different in the westerlies-dominated (with higher aridity index, runoff
coefficient and glacier cover), the Indian monsoon-dominated and the East Asian
monsoon-dominated (with higher air temperature, vegetation cover and
evapotranspiration) basins under the perspective of Budyko framework. In the 18 TP
basins, precipitation is the major contributor to the river runoff, which concentrates
mainly during June-October (June-August for the westerlies-dominated basins,
June-September or June to October for the Indian monsoon-dominated and the East
Asian monsoon-dominated basins). The basin-wide SWE is relatively higher from
mid-autumn to spring for all 18 TP basins except for Keliya River and Brahmaputra
River above the Nuxia and Yangcun stations. The vegetation cover is relatively less
whereas snow/glacier cover is more in the westerlies-dominant basins compared with
to other basins. In the period 1982-2011, we found that hydrologic cycle intensified across most of the basins in Tibetan Plateau; receded at some tributaries located at the upper Yellow River and Yalong River (due to weakening East Asian monsoon).  The aridity index (PET/P) exhibited decrease in most TP basins which corresponded to the warming and moistening climate in the TP and western China. Moreover, the runoff coefficient (Q/P) declined in most basins which may be, to some extent, due to ET increase induced by vegetation greening and the influences of snow and glacier changes. Although there are considerable uncertainties inherited from multi-source data used, the general hydrological regimes in the TP basins could be revealed, which are consistent to the existing results obtained from in situ observations and complex land surface modeling. It indicated the usefulness of integrating the multiple datasets  (e.g.,  in situ observations, remote sensing-based products, reanalysis outputs, land surface model simulations and climate model outputs) for hydrological applications. The generalization here  could be helpful for understanding the hydrological cycle and supporting sustainable  water resources management and eco-environment protection  in the  Tibetan Plateau under global warming.

*Author contributions*. Wenbin Liu and Fubao Sun developed the idea to see the general water budgets in the TP basins from the perspective of multisource datasets. Wenbin Liu collected and processed the multiple datasets with the help of Yanzhong Li, Guoqing Zhang, Wee Ho Lim, Hong Wang as well as Peng Bai, and prepared the manuscript. The results were extensively commented and discussed by Fubao Sun,

Jiahong Liu and Yan-Fang Sang.

*Acknowledgements*. This study was supported by the National Key Research and

Development Program of China (2016YFC0401401 and 2016YFA0602402),National

Natural Science Foundation of China (41401037 and 41330529), the Open Research

Fund of State Key Laboratory of Desert and Oasis Ecology in Xinjiang Institute of

Ecology and Geography, Chinese Academy of Sciences (CAS), the CAS Pioneer

Hundred Talents Program (Fubao Sun), the Initial Founding of Scientific Research (Y5V50019YE) and, the CAS President's International Fellowship Initiative (2017PC0068) and the program for the "Bingwei" Excellent Talents from the Institute of Geographic Sciences and Natural Resources Research, CAS. We are grateful to the

NASA MEaSUREs Program (Sean Swenson) for providing the GRACE land data processing algorithm. The basin-wide water budget series in the TP Rivers used in this study are available from the authors upon request (liuwb@igsnrr.ac.cn). We wish to thank the editors and reviewers for their invaluable comments and constructive suggestions to improve the quality of the manuscript.

**Table 2**: Overview of multi-source datasets applied in this study

| Data category | Data source | Spatial resolution | Temporal resolution | Available period used | Reference |
|---|---|---|---|---|---|
| Runoff (Q) | Observed, National Hydrology Almanac of China | — | Daily | 1982-2011 | — |
| | VIC_IGSNRR simulated | 0.25$^o$ | Daily | 1982-2011 | Zhang et al. (2014) |
| Precipitation (P) | Observed, CMA | 0.5$^o$ | Monthly | 1982-2011 | — |
| | TRMM 3B43 V7 | 0.25$^o$ | Monthly | 2000-2011 | Huffman et al. (2012) |
| | IGSNRR forcing | 0.25$^o$ | Daily | 1982-2011 | Zhang et al. (2014) |
| Temperature (Temp.) | Observed, CMA | 0.5$^o$ | Monthly | 2000-2011 | — |
| Terrestrial storage change (ΔS) | GRACE-CSR | Approx.300-400 km | Monthly | 2002-2011 | Tapley et al. (2004) |
| | GRACE-GFZ | Approx.300-400 km | Monthly | 2002-2011 | Tapley et al. (2004) |
| | GRACE-JPL | Approx.300-400 km | Monthly | 2002-2011 | Tapley et al. (2004) |
| Potential evaporation (PET) | CRU | 0.5$^o$ | Monthly | 1982-2011 | Harris et al. (2013) |
| Actual evaporation (ET) | MTE_E | 0.5$^o$ | Monthly | 1982-2011 | Jung et al. (2010) |
| | VIC_E | 0.25$^o$ | Daily | 1982-2011 | Zhang et al. (2014) |
| | GLEAM_E | 0.25$^o$ | Daily | 1982-2011 | Miralles et al. (2011) |
| | PML_E | 0.5$^o$ | Monthly | 1982-2011 | Zhang Y et al. (2016) |
| | Zhang_E | 8 km | Monthly | 1983-2006 | Zhang et al. (2010) |
| | GNoah_E | 1.0$^o$ | 3 hourly | 1982-2011 | Rui (2011) |

| | | | | | |
|---|---|---|---|---|---|
| NDVI | GIMMS NDVI dataset | 8 km | 15 daily | 1982-2011 | Tucker et al. (2005) |
| LAI | GLASS LAI Product | 0.05$^o$ | 8 daily | 1982-2011 | Liang and Xiao (2012) |
| Snow Cover | TP Snow composite Products | 500 m | Daily | 2005-2013 | Zhang et al. (2012) |
| SWE | VIC_IGSNRR simulated | 0.25$^o$ | Daily | 1982-2011 | Zhang et al. (2014) |
| | GlobSnow-2 Product | 25 km | Daily | 1982-2011 | Takala et al. (2011) |
| Glacier Area | the Second Glacier Inventory Dataset of China | _ | _ | 2005 | Guo et al. (2014) |

**Table 12**: Main features of the 18 used TP river basins used in this study. The precipitation and temperature statistics for each basin were calculated from the observed CMA datasets while the NDVI and LAI statistics were extracted from the GIMMS NDVI dataset and GLASS LAI product. The GA% and SC% represented the percentages of multiyear-mean glacier cover and snow cover in each basin which were calculated . The glacier and snow cover data are extracted, respectively, from the Second Glacier Inventory Dataset of China and the daily TP snow cover dataset (2005-2013).

| No. | Station | Altitude (m) | River name | Drainage area (km$^2$) | Multiyear-mean (1982-2011) and basin-averaged parameters | | | | | | |
|---|---|---|---|---|---|---|---|---|---|---|---|
| | | | | | Q (mm/yr) | Prec. (mm/yr) | Temp.($^o$C/yr) | NDVI | LAI | GA% | SC% |
| 01 | Kulukelangan | 2000 | Yerqiang | 32880.00 | 158.60 | 128.34 | -5.68 | 0.05 | 0.03 | 10.97 | 35.03 |
| 02 | Tongguziluoke | 1650 | Yulongkashi | 14575.00 | 151.56 | 134.04 | -4.07 | 0.06 | 0.04 | 23.27 | 35.95 |
| 03 | Numaitilangan | 1880 | Keliya | 7358.00 | 103.18 | 137.14 | -4.78 | 0.06 | 0.03 | 10.86 | 29.16 |
| 04 | Zelingou | 4282 | Bayin | 5544.00 | 41.42 | 340.68 | -4.98 | 0.13 | 0.09 | 0.09 | 21.22 |
| 05 | Gadatan | 3823 | Yellow | 7893.00 | 200.95 | 566.01 | -4.60 | 0.34 | 0.54 | 0.13 | 14.94 |
| 06 | Xining | 3225 | Yellow | 9022.00 | 99.90 | 503.74 | 0.97 | 0.36 | 0.70 | 0.00 | 10.06 |
| 07 | Tongren | 3697 | Yellow | 2832.00 | 149.36 | 533.25 | -1.37 | 0.39 | 0.83 | 0.00 | 9.42 |
| 08 | Tainaihai | 2632 | Yellow | 121972.00 | 159.48 | 540.32 | -2.40 | 0.34 | 0.72 | 0.09 | 15.89 |
| 09 | Huangheyan | 4491 | Yellow | 20930.00 | 31.18 | 386.42 | -4.81 | 0.23 | 0.61 | 0.00 | 17.25 |
| 10 | Jimai | 4450 | Yellow | 45015.00 | 85.50 | 441.48 | -4.16 | 0.26 | 0.52 | 0.00 | 20.05 |

| 11 | Yajiang | 2599 | Yalong | 67514.00 | 237.66 | 717.05 | -0.23 | 0.43 | 0.80 | 0.15 | 18.36 |
| 12 | Zhimenda | 3540 | Yangtze | 137704.00 | 96.23 | 405.66 | -4.83 | 0.20 | 0.26 | 0.96 | 17.87 |
| 13 | Jiaoyuqiao | 3000 | Salween | 72844.00 | 364.26 | 620.88 | -1.89 | 0.29 | 0.44 | 2.02 | 23.73 |
| 14 | Pangduo | 5015 | Brahmaputra | 16459.00 | 348.31 | 544.59 | -1.53 | 0.27 | 0.33 | 1.66 | 23.33 |
| 15 | Tangjia | 4982 | Brahmaputra | 20143.00 | 350.61 | 555.17 | -1.89 | 0.27 | 0.34 | 1.39 | 21.83 |
| 16 | Gongbujiangda | 4927 | Brahmaputra | 6417.00 | 586.96 | 692.06 | -4.24 | 0.27 | 0.36 | 4.12 | 25.99 |
| 17 | Nuxia | 2910 | Brahmaputra | 191235.00 | 307.38 | 401.35 | -0.73 | 0.22 | 0.25 | 1.90 | 13.50 |
| 18 | Yangcun | 3600 | Brahmaputra | 152701.00 | 163.25 | 349.91 | -0.87 | 0.19 | 0.18 | 1.28 | 10.52 |

**Table 33**: Contribution of glacier-melt to discharge in eighteen 18 TP basins ("—" shows no glacier influences, "—*" shows the percentage is empirically estimated through the relation between lacier area ratio and glacier melt for basins in which the glacier melt contribution has been reported in the literatures)

| Basin | Contributions of glacier-melt to discharge (%) | Reference |
|---|---|---|
| Kulukelangan | 62.73 | Mansur and Ajnisa (2005) |
| Tongguziluoke | 64.90 | Liu J et al. (2016) |
| Numaitilangan | 71.0 | Chen (1988) |
| Zelingou | — | — |
| Gadatan | — | — |
| Xining | — | — |
| Tongren | — | — |
| Tainaihai | 0.80 | Zhang et al. (2013) |
| Huangheyan | — | — |
| Jimai | — | — |

| Basin | | |
|-------|-----|------|
| Yajiang | 1.40 | —* |
| Zhimenda | 6.50 | Zhang et al. (2013) |
| Jiaoyuqiao | 4.80 | Zhang et al. (2013) |
| Nuxia | 11.60 | Zhang et al. (2013) |
| Pangduo | 10.13 | —* |
| Tangjia | 8.549 | —* |
| Gongbujiangda | 25.2.15 | —* |
| Yangcun | 7.81 | —* |

**Table 4**: Annual-averaged water storage changes (ΔS) in 18 TP basins derived from GRACE retrievals (2002-2013) from three different processing centers (CSR, GFZ and JPL)

[revised manuscript text omitted]